# Learning General Causal Structures with Hidden Dynamic Process for Climate Analysis

## Abstract

Understanding climate dynamics requires going beyond correlations in observational data to uncover their underlying causal process. Latent drivers, such as atmospheric processes, play a critical role in temporal dynamics, while direct causal influences also exist among geographically proximate observed variables. Traditional Causal Representation Learning (CRL) typically focuses on latent factors but overlooks such observable-to-observable causal relations, limiting its applicability to climate analysis. In this paper, we introduce a unified framework that jointly uncovers (i) causal relations among observed variables and (ii) latent driving forces together with their interactions. We establish conditions under which both the hidden dynamic processes and the causal structure among observed variables are simultaneously identifiable from time-series data. Remarkably, our guarantees hold even in the nonparametric setting, leveraging contextual information to recover latent variables and causal relations. Building on these insights, we propose CaDRe (**Ca**usal **D**iscovery and **Re**presentation learning), a time-series generative model with structural constraints that integrates CRL and causal discovery. Experiments on synthetic datasets validate our theoretical results. On real-world climate datasets, CaDRe not only delivers competitive forecasting accuracy but also recovers visualized causal graphs aligned with domain expertise, thereby offering interpretable insights into climate systems.

## 1 Introduction

Understanding the causal structure of climate systems is fundamental not only to scientific reasoning (Runge et al., 2019a), but also to reliable modeling and prediction. Given the observed data with $d_x$ variables: $\mathbf{x}_t = [x_{t,1}, \ldots, x_{t,d_x}]$, our goal is twofold: (i) to discover the underlying latent variables $\mathbf{z}_t = [z_{t,1}, \ldots, z_{t,d_z}]$ and their temporal interactions, and (ii) to identify causal relations among observed variables. To better understand this problem, we describe it using a causal modeling perspective. As depicted in Figure 1, latent drivers $\mathbf{z}_t$, such as pressure and precipitation (Chen & Wang, 1995), are not directly measured but significantly influence the observed dynamics. These latent processes evolve jointly and stochastically, exhibiting both *instantaneous* and *time-lagged* causal dependencies (Lucarini et al., 2014; Rolnick et al., 2022). They govern observable quantities $\mathbf{x}_t$, such as temperature, which reflect underlying dynamics and also exhibit spatial interactions through emergent weather patterns, like wind circulation systems.

Identifying these underlying hidden variables and temporal relations is the central objective of Causal Representation Learning (CRL) (Schölkopf et al., 2021). Recent advances in identifiability theory and practical algorithm design fall under the framework of nonlinear Independent Component Analysis (ICA). These approaches typically rely on auxiliary variables (Hyvarinen & Morioka, 2016; 2017; Hyvärinen et al., 2023; Yao et al., 2022), sparsity (Lachapelle et al., 2022; Zheng et al., 2022; Zheng & Zhang, 2023; Lachapelle et al., 2024; Brouillard et al., 2024), or restricted generative functions (Gresele et al., 2021), and generally assume a *noise-free* and *invertible* generation from $\mathbf{z}_t$ to $\mathbf{x}_t$, in order to *directly* recover latent space. However, climatic measurements exhibit both observational dependencies and stochastic noise, violating these assumptions and limiting the applicability of existing CRL approaches.

This problem can also be cast as the problem of Causal Discovery (CD) (Spirtes et al., 2001; Pearl, 2009) in the presence of latent processes. CD often relies on parametric models, such as linear non-

Figure 1: From climate system to causal graph. $\mathbf{x}_t$ represent observed data and $\mathbf{z}_t$ denotes unobserved variables behind $\mathbf{x}_t$, $\epsilon_t^z$ denotes the stochasticity in latent causal process, and $s_t$ denotes the noise variable varying with $\mathbf{z}_t$, $e.g.$, human activities (Chen & Wang, 1995).

Gaussianity (Shimizu et al., 2006), nonlinear additive (Hoyer et al., 2008; Lachapelle et al., 2019), post-nonlinear models (Zhang & Hyvarinen, 2012), as well as nonparametric methods with (Huang et al., 2020; Reizinger et al., 2023; Monti et al., 2020) or without auxiliary variables (Spirtes & Glymour, 1991; Zheng et al., 2023; Zhang et al., 2012). However, generally speaking, they cannot identify latent variables, their interrelations, and their causal influence on observed variables. For example, Fast Causal Inference (FCI) algorithm (Spirtes & Glymour, 1991) produces asymptotically correct results in the presence of latent confounders by exploiting conditional independence relations, but its result is often not informative enough, $e.g.$, it cannot recover causally-related latent variables.

The above underscores the need for a unified framework capable of modeling both the causal relations among the observed variables and latent dynamic processes inherent to real-world climate systems. We understand the climate system through a causal lens and establish the identifiability guarantees for jointly recovering latent dynamics and causal graphs among observations. Intuitively, the temporal structure enables leveraging contextual observable information to identify latent factors, while the inferred latent dynamics, in turn, modulate how that observable-level causal graphs evolve. We instantiate this insight in a state-space Variational AutoEncoder (VAE), which can conduct nonparametric **Ca**usal **D**iscovery and **Re**presentation learning (**CaDRe**) simultaneously.

CaDRe employs parallel flow-based priors to *learn independent components* to reflect structural dependencies, and introduces gradient-based structural penalties on both latent transitions and decoders to ensure identifiability. Extensive synthetic experiments on the identification of latent representation learning and causal discovery validate our theoretical guarantees. On real-world climate data, CaDRe achieves competitive forecasting accuracy, indicating the effectiveness of the learned temporal process. Moreover, the visualized causal graphs and latent variables provide physical interpretations aligned with known scientific phenomena, and they further reveal structural patterns that may inspire new hypotheses in climate science.

## 2 PROBLEM SETUP

**Technical Notations.** We formalize the climate system using the following variable/function notations. We observe a time-series of observed variables $\mathbf{X} = [\mathbf{x}_1, \mathbf{x}_2, \cdots, \mathbf{x}_T]$, whereas their underlying factors $\mathbf{Z} = [\mathbf{z}_1, \mathbf{z}_2, \cdots, \mathbf{z}_T]$ are unobservable. Regarding the system in one time-step, as depicted in Figure 1, it consists of observed variables $\mathbf{x}_t := [x_{t,i}]_{i \in \mathcal{I}}$ with index set $\mathcal{I} = \{1, 2, \ldots, d_x\}$, and latent variables $\mathbf{z}_t := [z_{t,j}]_{j \in \mathcal{J}}$ indexed by $\mathcal{J} = \{1, 2, \ldots, d_z\}$. Let $\mathbf{pa}(\cdot)$ denote the parent variables, $\mathbf{pa}_O(\cdot)$ refer to observable parents, and $\mathbf{pa}_L(\cdot)$ indicate the latent parents. In particular, $\mathbf{pa}_L(\cdot)$ comprises latent variables from both the current and previous time step. Throughout the paper, the hat notation, $e.g.$, $\hat{\mathbf{x}}_t$, denotes estimated variables or functions.

**Data Generating Process.** We translate how the complex and dynamic climate system evolves to the following Structural Equation Model (SEM) (Pearl, 2009) at each discrete time step:

$$x_{t,i} = \underbrace{g_i(\mathbf{pa}_O(x_{t,i}), \mathbf{pa}_L(x_{t,i}), s_{t,i})}_{\text{effects from } \mathbf{x}_t \text{ and } \mathbf{z}_t}, \quad z_{t,j} = \underbrace{f_j(\mathbf{pa}_L(z_{t,j}), \epsilon_{t,j}^z)}_{\text{effects from } \mathbf{z}_{t-1} \text{ and } \mathbf{z}_t}, \quad s_{t,i} = \underbrace{g_{s_i}(\mathbf{z}_t, \epsilon_{t,i}^x)}_{\text{noise conditioned on } \mathbf{z}_t}, \quad (1)$$

Table 1: Key notations, explanations, and structures used in the CaDRe.

| Symbol | Explanation | Symbol | Explanation | Symbol | Explanation |
|---|---|---|---|---|---|
| $\mathbf{x}_t$ | observed variables at time $t$ | $\mathbf{z}_t$ | latent variables at time $t$ | $\mathbf{s}_t$ | observation noise dependent on $\mathbf{z}_t$ |
| $\epsilon_{\mathbf{x}_t}$ | independent noise of observations | $\epsilon_{\mathbf{z}_t}$ | independent latent noise | $\mathcal{X}_t$ | support of observed variables |
| $\mathcal{Z}_t$ | support of latent variables | $g(\cdot)$ | generating function from $(\mathbf{z}_t, \mathbf{s}_t)$ to $\mathbf{x}_t$ | $r(\cdot)$ | latent transition function from $\mathbf{z}_{t-1}$ to $\mathbf{z}_t$ |
| $m(\cdot)$ | mixing function from $(\mathbf{z}_t, \mathbf{s}_t)$ to $\mathbf{x}_t$ | $h_z(\cdot)$ | invertible transformation from $\mathbf{z}_t$ to $\hat{\mathbf{z}}_t$ | $\mathbf{J}_g(\mathbf{x}_t)$ | Jacobian for observed causal DAG |
| $\mathbf{J}_r(\mathbf{z}_t)$ | Jacobian for latent instantaneous DAG | $\mathbf{J}_m(\mathbf{s}_t)$ | Jacobian for mixing structure | $\mathrm{supp}(\cdot)$ | support matrix of Jacobian |

where $g_i$ and $f_j$ are differentiable functions, and noise terms $\epsilon_{t,j}^z \sim p_{\epsilon_{z_j}}, \epsilon_{t,i}^x \sim p_{\epsilon_{x_i}}$ are mutually independent for $\mathcal{I}$ and $\mathcal{J}$. As discussed in the introduction, the observed variable $x_{t,i}$ may be influenced by other observed components $\mathbf{x}_{t,\backslash i}$ and the latent variables $\mathbf{z}_t$. For example, temperature in a specific region may be governed by latent drivers such as solar radiation, and also be affected by neighboring regions through heat transfer. The endogenous mediator or nonstationary noise $s_{t,i}$, depending on latent driving forces $\mathbf{z}_t$, is designed to capture dynamic uncertainties, *e.g.*, perturbations introduced by $CO_2$ (Stips et al., 2016). The latent variable $z_{t,j}$ evolves according to both instantaneous interactions with other components $\mathbf{z}_{t,\backslash j}$ and time-lagged dependencies from the previous step $\mathbf{z}_{t-1}$. Aiming at reliably discovering causal graphs, we adopt an assumption in Spirtes et al. (2001):

**Assumption 1.** *The distribution of* $(\mathbf{X}, \mathbf{Z})$ *is Markov and faithful to a Directed Acyclic Graph (DAG).*

## 3 IDENTIFICATION THEORY

**Overview of Theory.** Given the above definitions and goals, we establish the identifiability of the latent space in Theorem 1, and then identify the latent causal process in Theorem A.3 under sparsity within the latent temporal process. To leverage those recovered latent driving, in Theorem 2, we draw a connection between the SEM and nonlinear ICA with latent variables, which are shown to describe the same data generating process. It nourishes a *functional equivalence* in Theorem 2 for computing the causal graphs through the mixing structure of ICA. Then, leveraging the cross-derivative condition (Lin, 1997), Theorem 3 subsequently identifies causal graphs over observed variables in the spirit of identifiable nonlinear ICA with internal temporal shift and the DAG constraints.

### 3.1 LATENT SPACE RECOVERY AND LATENT VARIABLES IDENTIFICATION

In this section, we aim to characterize the relationships between ground truth $\mathbf{z}_t$ and estimated $\hat{\mathbf{z}}_t$. We consider, without loss of generality, a first-order Markov structure, in which three consecutive observations $\{\mathbf{x}_{t-1}, \mathbf{x}_t, \mathbf{x}_{t+1}\}$ are used as contextual information. The generalization to higher-order Markov structures is discussed in Appendix E.3. To formalize the stochastic generation process, we introduce an operator $L$ (Dunford & Schwartz, 1971) to represent distribution-level transformations, that is, how one probability distribution is pushed forward to another. Given two random variables $a$ and $b$ with supports $\mathcal{A}$ and $\mathcal{B}$ respectively, the transformation $p_a \mapsto p_b$ is formalized as:

$$p_b = L_{b|a} \circ p_a, \text{ where } L_{b|a} \circ p_a \coloneqq \int_{\mathcal{A}} p_{b|a}(\cdot \mid a) p_a(a) da. \tag{2}$$

For example, operators $L_{\mathbf{x}_{t+1}|\mathbf{z}_t}$ and $L_{\mathbf{x}_{t-1}|\mathbf{x}_{t+1}}$ represent the distributional transformations $p_{\mathbf{z}_t} \mapsto p_{\mathbf{x}_{t+1}}$ and $p_{\mathbf{x}_{t+1}} \mapsto p_{\mathbf{x}_{t-1}}$, respectively. With this operator in hand, we turn to addressing the problem of "causally-related" observed variables in recovering the latent space. When the causal graph is a DAG, information propagates along causal pathways without being trapped in a *self-loop*. In other words, causal influence can be traced back to its source by following the reverse direction of the DAG through the "short reaction lag" (Fisher, 1970). Consequently, the distributional transformation from sources $\mathbf{s}_t$ to observations $\mathbf{x}_t$ in the generative process must be *injective*. We formalize it as:

**Lemma 1.** *(Injective DAG Operator) Under Assumption 1, $L_{\mathbf{x}_t|\mathbf{s}_t}$ is injective for all $t \in \mathcal{T}$.*

This result implies that the nonlinear causal DAG over $\mathbf{x}_t$ does not disturb the recovery of the latent space distribution. Building on this, we now address a more fundamental challenge: the latent variable $\mathbf{z}_t$ cannot be recovered from a single noisy observation $\mathbf{x}_t$, as the stochasticity makes the value-level mapping ill-posed. Instead of identifying values directly, we first formulate identifiability at the distributional level, and then seek the value-level identifiability from it for the subsequent component-wise interpretability. We found that, adjacent observations $\mathbf{x}_{t-1}$ and $\mathbf{x}_{t+1}$ contain non-trivial information about $\mathbf{z}_t$ if they exhibit *minimal distributional changes*. To instantiate it, we

provide *nonparametric* identifiability of the latent submanifold based on distributional variations captured by contextual measurements.

**Theorem 1.** *(Identifiability of Latent Space)* *Suppose observed variables and hidden variables follow the data generating process in Eq. (1), and estimated observations $\{\hat{\mathbf{x}}_{t-1}, \hat{\mathbf{x}}_t, \hat{\mathbf{x}}_{t+1}\}$ match the true joint distribution of $\{\mathbf{x}_{t-1}, \mathbf{x}_t, \mathbf{x}_{t+1}\}$. The following assumptions are imposed:*

*A1 (Computable Probability:) The joint, marginal, and conditional distributions of $(\mathbf{x}_t, \mathbf{z}_t)$ are all bounded and continuous.*

*A2 (Contextual Variability:) The operators $L_{\mathbf{x}_{t+1}|\mathbf{z}_t}$ and $L_{\mathbf{x}_{t-1}|\mathbf{x}_{t+1}}$ are injective and bounded.*

*A3 (Latent Drift:) For any $\mathbf{z}_t^{(1)}, \mathbf{z}_t^{(2)} \in \mathcal{Z}_t$ where $\mathbf{z}_t^{(1)} \neq \mathbf{z}_t^{(2)}$, we have $p(\mathbf{x}_t|\mathbf{z}_t^{(1)}) \neq p(\mathbf{x}_t|\mathbf{z}_t^{(2)})$.*

*A4 (Differentiability:) There exists a functional $F$ such that $F\left[p_{\mathbf{x}_t|\mathbf{z}_t}(\cdot \mid \mathbf{z}_t)\right] = h_z(\mathbf{z}_t)$ for all $\mathbf{z}_t \in \mathcal{Z}_t$, where $h_z$ is differentiable.*

*Then we have $\hat{\mathbf{z}}_t = h_z(\mathbf{z}_t)$, where $h_z : \mathbb{R}^{d_z} \to \mathbb{R}^{d_z}$ is an invertible and differentiable function.*

**Discussion on Assumptions.** As presented, A1 is a moderate condition for computable density functions. A2 introduces sufficient distributional variability, formalized via injectivity at the density level. A3 ensures that distinct values of $\mathbf{z}_t$ induce distinct conditionals $p(\mathbf{x}_t \mid \mathbf{z}_t)$, which is violated only when two values of $\mathbf{z}_t$ yield identical distributions. A4 requires that the mapping from $\mathbf{z}_t$ to $p(\mathbf{x}_t \mid \mathbf{z}_t)$ is differentiable, a condition naturally satisfied by models based on differentiable neural networks, such as VAEs. Please refer to Appendix A.2 for a detailed elaboration of the assumptions.

**Proof Sketch and Contributions.** The complete proof is deferred to Appendix A. Prior work on nonparametric identifiability (Hu & Schennach, 2008; Carroll et al., 2010) relies on partially knowing the function form of $g$, and yields only distribution-level identifiability, *i.e.*, $p_{\hat{\mathbf{z}}_t} = p_{\mathbf{z}_t}$. In contrast, our approach requires no such prior knowledge and achieves identifiability at the value level, a more informative result for analyzing the latent components. We begin by proving the uniqueness of the posterior collection $\{p(\mathbf{x}_t \mid \hat{\mathbf{z}}_t)\}_{\hat{\mathcal{Z}}_t}$, where the unordered set unveils the existence of a relabeling function $h_z$ on the conditioning variables. A3 then ensures a one-to-one correspondence between $\mathbf{z}_t$ and the posteriors $p(\mathbf{x}_t \mid \hat{\mathbf{z}}_t)$, thereby ruling out degenerate mappings from posteriors to values.

$$\boxed{\{p_{\mathbf{x}_t|\mathbf{z}_t}(\cdot \mid \mathbf{z}_t)\}_{\mathcal{Z}_t} = \{p_{\mathbf{x}_t|\hat{\mathbf{z}}_t}(\cdot \mid \hat{\mathbf{z}}_t)\}_{\hat{\mathcal{Z}}_t}} \Rightarrow \boxed{p_{\mathbf{x}_t|\mathbf{z}_t}(\mathbf{x}_t \mid h_z(\mathbf{z}_t)) = p_{\mathbf{x}_t|\hat{\mathbf{z}}_t}(\mathbf{x}_t \mid \hat{\mathbf{z}}_t)} \Rightarrow \boxed{\hat{\mathbf{z}}_t = h_z(\mathbf{z}_t)}$$

Finally, A4 pins down the $h_z$ to be differentiable. Unlike nonlinear ICA, the nonparametric formulation accommodates *sparse/noisy* observations, thereby improving its applicability on climate data. To enhance interpretability by ensuring that each latent component corresponds to a distinct physical variable, in Appendix A.3, we introduce a sparsity assumption on the latent dynamics to achieve ***component-wise identifiability of latent variables and causal process***, which is motivated by that physical climate factors, *e.g.*, solar radiation and atmospheric, exhibit localized sparse influences.

### 3.2 NONPARAMETRIC CAUSAL DISCOVERY WITH HIDDEN DYNAMIC PROCESS

Building upon these results, our goal is to identify nonparametric causal graphs over $\mathbf{x}_t$, even if they are modulated by a hidden dynamic process. This setting is notably more general than the identifiability guarantees in prior (standard) CD frameworks, as summarized in Table 2. Recent works (Monti et al., 2020; Reizinger et al., 2023) extend the ICA-based CD (Shimizu et al., 2006) to nonparametric settings via nonlinear ICA (Hyvarinen et al., 2019). However, they are not applicable in the presence of dynamic latent confounders. To overcome this limitation, we start by establishing a connection between SEMs and nonlinear ICA in the presence of a hidden process.

**Lemma 2.** *(Nonlinear SEM ⇔ Nonlinear ICA) There exists a function $m_i$, which is differentiable w.r.t. $s_{t,i}$ and $\mathbf{x}_t$, for any fixed $s_{t,i}$ and $\mathbf{z}_t$, such that the following two representations,*

$$x_{t,i} = g_i(\mathbf{pa}_O(x_{t,i}), \mathbf{pa}_L(x_{t,i}), s_{t,i}) \quad and \quad x_{t,i} = m_i(\mathbf{z}_t, \mathbf{s}_t) \tag{3}$$

*describe the same data generating process. That is, both expressions yield the same value of $x_{t,i}$.*

After building this equivalence, we proceed to perform CD via the nonlinear ICA with latent variables. We begin by introducing the Jacobian matrices on this data generating process, as they serve as proxies for the (nonlinear) adjacency matrix. For all $(i,j) \in \mathcal{I} \times \mathcal{I}$, we define $[\mathbf{J}_m(\mathbf{s}_t)]_{i,j} = \frac{\partial x_{t,i}}{\partial s_{t,j}}$, $[\mathbf{J}_g(\mathbf{x}_t)]_{i,j} = \frac{\partial x_{t,i}}{\partial x_{t,j}}$, $\mathbf{D}_m(\mathbf{s}_t) = \text{diag}(\frac{\partial x_{t,1}}{\partial s_{t,1}}, \frac{\partial x_{t,2}}{\partial s_{t,2}}, \dots, \frac{\partial x_{t,d_x}}{\partial s_{t,d_x}})$, and $\mathbf{I}_{d_x}$ is the identity matrix in $\mathbb{R}^{d_x \times d_x}$. Especially, $\mathbf{J}_m(\mathbf{s}_t)$ corresponds to the mixing procedure of nonlinear ICA, as described on the R.H.S. of Eq. (3), and $\mathbf{J}_g(\mathbf{x}_t)$ signifies the causal graph over observations in the nonlinear SEM, the L.H.S. of Eq. (3), provided the faithfulness assumption outlined below holds.

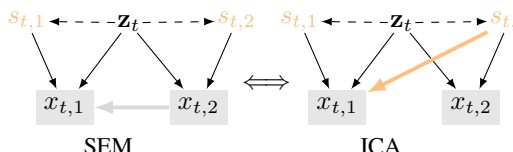

Figure 2: **Equivalent SEM and ICA.** The gray line in SEM denotes the influence $x_{t,2} \rightarrow x_{t,1}$ through the observation causal relation, which is equivalently represented as an indirect effect (the orange line): $s_{t,2} \dashrightarrow x_{t,1}$ in ICA, which can be decomposed into $s_{t,2} \rightarrow x_{t,2}$ and $x_{t,2} \rightarrow x_{t,1}$.

**Assumption 2** (Functional Faithfulness). *The causal adjacency structure among observed variables is given by the support of the Jacobian matrix $\mathbf{J}_g(\mathbf{x}_t)$.*

This assumption implies *edge minimality* in causal graphs, analogous to the structural minimality discussed in Remark 6.6 of Peters et al. (2017) and minimality in Zhang (2013), which enables us to establish an equivalence between the causal graph and the ICA mixing structure as follows.

**Theorem 2.** *(**Functional Equivalence**) Consider the two types of data generating process described in Eq. (3), the following equation always holds:*

$$\mathbf{J}_g(\mathbf{x}_t)\mathbf{J}_m(\mathbf{s}_t) = \mathbf{J}_m(\mathbf{s}_t) - \mathbf{D}_m(\mathbf{s}_t). \tag{4}$$

**Proof Sketch.** Following the depiction of the SEM, the flow of information can be traced starting from the observed variables $\mathbf{x}_t$. The DAG structure ensures that the sources are the latent variables and the independent noise, implying that the data generation process conforms to a specific form of nonlinear ICA: $[\mathbf{z}_t, \boldsymbol{\epsilon}_{x_t}] \Rightarrow \mathbf{x}_t$, where $\mathbf{z}_t$ is characterized as a conditional prior (Refer to Appendix A.4 for a detailed proof). From this formulation, we derive two corollaries: (i) the DAG structure removes the need for the invertibility assumption in nonlinear ICA, and (ii) the SEM–ICA correspondence enables efficient CD through transforming the mixing procedure, an *instant inference* yielding causal graphs.

**Corollary 2.1.** *Under Assumption 1, given any $\mathbf{z}_t \in \mathcal{Z}_t$, $\mathbf{J}_m(\mathbf{s}_t)$ is an invertible matrix.*

**Corollary 2.2.** *Causal graphs over observations are represented by $\mathbf{J}_g(\mathbf{x}_t) = \mathbf{I}_{d_x} - \mathbf{D}_m(\mathbf{s}_t)\mathbf{J}_m^{-1}(\mathbf{s}_t)$.*

Building upon these SEM–ICA connections, we derive sufficient conditions under which the causal graph over observed variables becomes identifiable, as it benefits from the recovered latent dynamics.

**Theorem 3.** *(**Identifiability of Causal Graph over Observed Variables**) Let $\mathbf{A}_{t,k} = \log p(\mathbf{s}_{t,k}|\mathbf{z}_t)$, assume that $\mathbf{A}_{t,k}$ is twice differentiable in $s_{t,k}$ and is differentiable in $z_{t,l}$, where $l = 1, 2, ..., d_z$. Suppose Assumption 1, 2 holds true, and*

*A5 (Generation Variability). For any estimated $\hat{g}_m$ that makes $\mathbf{x}_t = \hat{\mathbf{x}}_t = \hat{m}(\hat{\mathbf{z}}_t, \hat{\mathbf{s}}_t)$, let*

$$\mathbf{V}(t,k) := \left[\frac{\partial^2 \mathbf{A}_{t,k}}{\partial s_{t,k}\partial z_{t,1}}, \dots, \frac{\partial^2 \mathbf{A}_{t,k}}{\partial s_{t,k}\partial z_{t,d_z}}\right], \mathbf{U}(t,k) := \left[\frac{\partial^3 \mathbf{A}_{t,k}}{\partial s_{t,k}\partial^2 z_{t,1}}, \dots, \frac{\partial^3 \mathbf{A}_{t,k}}{\partial s_{t,k}\partial^2 z_{t,d_z}}\right]^T,$$

*where for $k = 1, 2, \dots, d_x$, $2d_x$ vector functions $\mathbf{V}(t,1), \dots, \mathbf{V}(t,d_x), \mathbf{U}(t,1), \dots, \mathbf{U}(t,d_x)$ are linearly independent. Then we attain ordered component-wise identifiability (Definition 4), and the structure of the causal graph is identifiable, i.e., $supp(\mathbf{J}_g(\mathbf{x}_t)) = supp(\mathbf{J}_{\hat{g}}(\hat{\mathbf{x}}_t))$.*

**Proof Sketch.** The core idea of the proof extends the notion of component-wise identifiability. From Theorem 1, we know that block-level information about $\mathbf{z}_t$ is identifiable, which can be treated as a continuous conditional prior. Recall from data generating process 1 that satisfies $s_{t,i} \perp\!\!\!\perp s_{t,j} \mid \mathbf{z}_t$ for all $i \neq j$, by introducing variability as specified in A5, we can recover the components up to permutation. To eliminate this permutation ambiguity, we further exploit the structural constraints encoded by the DAG over observed variables. The full proof is provided in Appendix A.7.

$$\boxed{\hat{\mathbf{z}}_t = h_z(\mathbf{z}_t)} \Rightarrow \boxed{\hat{s}_{t,i} = h_s(s_{t,\pi(i)})} \Rightarrow \boxed{\text{supp}(\mathbf{J}_{\hat{m}}) = \text{supp}(\mathbf{J}_m)} \Rightarrow \boxed{\text{supp}(\mathbf{J}_{\hat{g}}) = \text{supp}(\mathbf{J}_g)}$$

Generally speaking, Theorem 1 establishes latent space recovery through nonparametric identification, Theorem 2 demonstrates the functional equivalence between SEM and ICA, and Theorem 3 builds upon these results—first applying Theorem 1 to identify a nonlinear ICA model and then using Theorem 2's equivalence to identify the corresponding nonlinear SEM.

## 4 ESTIMATION METHODOLOGY

Our theoretical insights shed light on the practical implementations. As shown in Figure 3, we instantiate them into an estimation framework for doing **Ca**usal **D**iscovery and causal **Re**presentation learning (**CaDRe**) in the nonparametric setting, enabling accurate inference of causal structures.

Table 2: Attributes table of CD methods as applied to time-series data. A check mark indicates that the method possesses the attribute or achieves the result, while a cross mark indicates the opposite.

| Method | Nonparametric | Latent Variables | Latent Causal Graph | Causal Graph over Observations | No Equivalence Classes |
|---|---|---|---|---|---|
| NESSM (Huang et al., 2019) | ✗ | ✗ | ✗ | ✓ | ✓ |
| CD-NOD (Huang et al., 2020) | ✓ | ✗ | ✗ | ✓ | ✗ |
| FCI (Spirtes & Glymour, 1991) | ✓ | ✓ | ✗ | ✓ | ✗ |
| LPCMCI (Gerhardus & Runge, 2020) | ✓ | ✓ | ✗ | ✓ | ✗ |
| CDSD (Brouillard et al., 2024) | ✓ | ✓ | ✓ | ✗ | ✓ |
| **CaDRe** | ✓ | ✓ | ✓ | ✓ | ✓ |

**Overall Architecture.** The proposed architecture is built upon the variational autoencoder (Kingma, 2013). In light of the data generating process described in Eq. (1), we can establish the Evidence Lower BOund (ELBO) as follows:

$$\mathcal{L}_{ELBO} = \mathbb{E}_{q(\mathbf{s}_{1:T}|\mathbf{x}_{1:T})}\left[\log p(\mathbf{x}_{1:T} \mid \mathbf{s}_{1:T}, \mathbf{z}_{1:T})\right] - \\ \lambda_1 D_{\text{KL}}\left(q(\mathbf{s}_{1:T} \mid \mathbf{x}_{1:T}) \,\|\, p(\mathbf{s}_{1:T} \mid \mathbf{z}_{1:T})\right) - \lambda_2 D_{\text{KL}}\left(q(\mathbf{z}_{1:T} \mid \mathbf{x}_{1:T}) \,\|\, p(\mathbf{z}_{1:T})\right), \quad (5)$$

where $\lambda_1$ and $\lambda_2$ are hyperparameters, and $\mathcal{D}_{KL}$ represents the Kullback-Leibler (KL) divergence. In practice, we set $\lambda_1 = 4 \times 10^{-3}$ and $\lambda_2 = 1.0 \times 10^{-2}$ to ensure stable training and reliable convergence. In Figure 3, the z-encoder, s-encoder, and decoder are implemented by Multi-Layer Perceptrons (MLPs) as

$$\hat{\mathbf{z}}_{1:T} = \phi(\mathbf{x}_{1:T}), \ \hat{\mathbf{s}}_{1:T} = \eta(\mathbf{x}_{1:T}), \ \hat{\mathbf{x}}_{1:T} = \psi(\hat{\mathbf{z}}_{1:T}, \hat{\mathbf{s}}_{1:T}), \quad (6)$$

where z-encoder $\phi$ extracts the latent variables from observations, s-encoder $\psi$ and decoder $\eta$ approximate functions for encoding $\mathbf{s}_t$ and reconstructing observations $\mathbf{x}_t$, respectively.

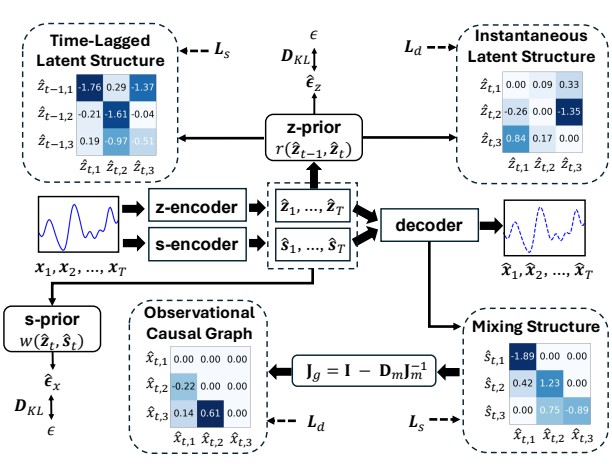

Figure 3: **The estimation procedure of CaDRe.** The model framework includes two encoders: z-encoder for extracting latent variables $\mathbf{z}_t$, and s-encoder for extracting nonstationary noise $\mathbf{s}_t$. A decoder reconstructs $\mathbf{x}_t$ from them. Additionally, prior networks estimate the prior distribution using a normalizing flow, focusing on learning the causal structures based on the Jacobian matrix. $\mathcal{L}_s$ imposes a sparsity constraint and $\mathcal{L}_d$ enforces the DAG structure on Jacobian matrix. $\mathcal{D}_{KL}$ enforces independence of the estimated noise by minimizing its KL divergence w.r.t. $\mathcal{N}(0, \mathbf{I})$. In summary, this method learns independent noise to inversely infer the causal structures.

**Estimating $\mathbf{z}_t$ with $\mathbf{s}_t$ using Flow Networks.** To capture temporal dependencies with emerging latent causal structures, we minimize the KL divergence between the approximate posterior from $\mathbf{x}_t$ and a learned prior of $\mathbf{z}_t$ from $\mathbf{z}_{t-1}$. We use a z-encoder to compute the posterior, and use a z-prior network, a normalizing flow conditioned on selected inputs, to compute the prior. Such input selection is guided by learnable inverse transition functions $r_i$ for $i$-th latent variables, where $\hat{\epsilon}^z_{t,i} = r_i(\hat{\mathbf{z}}_{t-1}, \hat{\mathbf{z}}_t)$ to identify which latent variables that causally influence $\hat{z}_{t,i}$. In detail, for the transformation $\kappa := \{\hat{\mathbf{z}}_{t-1}, \hat{\mathbf{z}}_t\} \to \{\hat{\mathbf{z}}_{t-1}, \hat{\epsilon}^z_t\}$, corresponded Jacobian matrix $\mathbf{J}_\kappa = \begin{pmatrix} \mathbf{I} & 0 \\ \mathbf{J}_r(\hat{\mathbf{z}}_{t-1}) & \mathbf{J}_r(\hat{\mathbf{z}}_t) \end{pmatrix}$ encodes these causal relations. Derived from normalizing flow, we have a prior estimation: $\log p(\hat{\mathbf{z}}_t, \hat{\mathbf{z}}_{t-1}) = \log p(\hat{\mathbf{z}}_{t-1}, \hat{\epsilon}^z_t) + \log |\frac{\partial r_i}{\partial \hat{z}_{t,i}}|$. Since the noise $\epsilon^z_{t,i}$ is independent of $\mathbf{z}_{t-1}$, the prior can be decomposed in terms as:

$$\log p(\hat{\mathbf{z}}_{1:T} \mid \hat{\mathbf{z}}_1) = \prod_{\tau=2}^{T}\left(\sum_{i=1}^{d_z} \log p(\hat{\epsilon}^z_{\tau,i}) + \sum_{i=1}^{d_z} \log |\frac{\partial r_i}{\partial \hat{z}_{\tau,i}}|\right), \quad (7)$$

where we assume $p(\hat{\epsilon}^z_{\tau,i}) \sim \mathcal{N}(0, \mathbf{I})$. After that, we can access $\frac{\partial r_i}{\partial \hat{z}_{t,i}}$ or $\mathbf{J}_r(\hat{\mathbf{z}}_t)$ which captures latent instantaneous structure and $\frac{\partial r_i}{\partial \hat{z}_{t-1,i}}$ or $\mathbf{J}_r(\hat{\mathbf{z}}_{t-1})$ which captures latent transition effects.

Figure 4: **Comparison with Constraint-Based CD.** We report the mean/standard deviation for all experiments. **Top:** Results with $d_x = 6$, $d_z = 3$, varying $n = \{200, 1000, 5000, 10000, 20000\}$. **Bottom:** Results with the sample size $n = 10000$ while selecting $d_x = \{3, 6, 8, 10\}$.

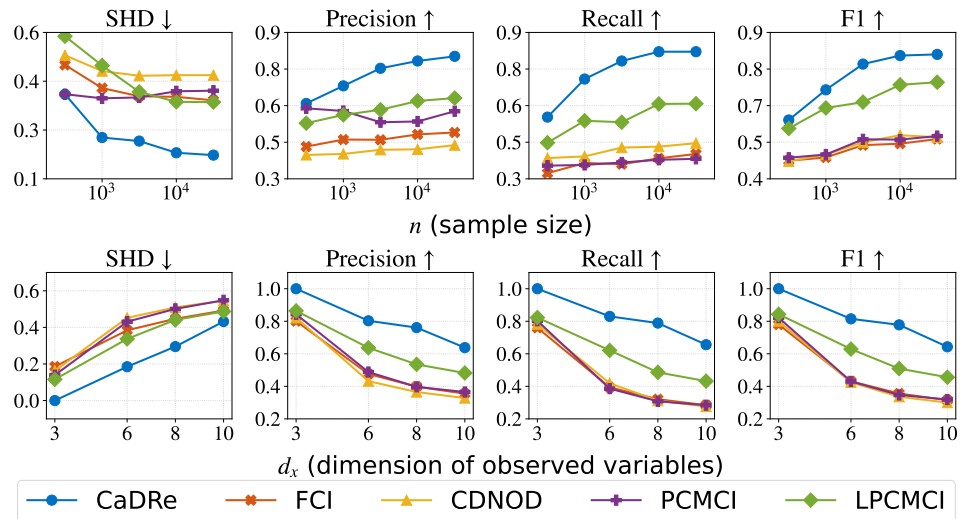

Using the same estimation backbone, we minimize the KL divergence between the prior of $\mathbf{s}_t$, parameterized through $\hat{\epsilon}^x_{t,i} = w_i(\hat{\mathbf{z}}_t, \hat{\mathbf{s}}_t)$ via a s-prior network, and the posterior obtained from the s-encoder. The transformation between $\hat{\mathbf{s}}_t$ and $\hat{\mathbf{z}}_t$ is then modeled as follows:

$$\log p\left(\hat{\mathbf{s}}_{1:T} \mid \hat{\mathbf{z}}_{1:T}\right) = \prod_{\tau=1}^{T} \left( \sum_{i=1}^{d_x} \log p\left(\hat{\epsilon}^x_{\tau,i}\right) + \sum_{i=1}^{d_x} \log \left| \frac{\partial w_i}{\partial \hat{s}_{\tau,i}} \right| \right). \tag{8}$$

Specifically, to ensure the conditional independence of $\hat{\mathbf{z}}_t$ and $\hat{\mathbf{s}}_t$, we use $\mathcal{D}_{KL}$ to minimize the KL divergence from the distributions of $\hat{\epsilon}^x_t$ and $\hat{\epsilon}^z_t$ to the distribution $\mathcal{N}(\mathbf{0}, \mathbf{I})$.

**Structure Learning.** Considering the causal graph over observed variables, we compute $\mathbf{J}_{\hat{m}}(\hat{\mathbf{s}}_t)$ from the decoder, and instantly obtain the causal graph over observations $\mathbf{J}_{\hat{g}}(\hat{\mathbf{x}}_t)$ via Corollary 2.2. Notably, the entries of $\mathbf{J}_{\hat{g}}(\hat{\mathbf{x}}_t)$ vary with other variables such as $\hat{\mathbf{z}}_t$, resulting in a DAG that could change over time. For the latent structure, we directly compute $\mathbf{J}_r(\hat{\mathbf{z}}_{t-1})$ and $\mathbf{J}_r(\hat{\mathbf{z}}_t)$ from z-prior network as the time-lagged structure and instantaneous structure in latent space, respectively. To prevent redundant edges and cycles, a sparsity penalty $\mathcal{L}_s$ is imposed on each learned structure, and DAG constraints $\mathcal{L}_d$ are imposed on the causal graph over observed variables and instantaneous latent causal DAG. Specifically, the Markov network structure for latent variables is derived as $\mathcal{M}(\mathbf{J}) = (\mathbf{I} + \mathbf{J})^\top (\mathbf{I} + \mathbf{J}) - \mathbf{I}$. Formally, we define these penalties:

$$\mathcal{L}_s = \|\mathcal{M}(\mathbf{J}_r(\hat{\mathbf{z}}_t))\|_1 + \|\mathcal{M}(\mathbf{J}_d(\hat{\mathbf{z}}_{t-1}))\|_1 + \|\mathbf{J}_{\hat{g}}(\hat{\mathbf{x}}_t)\|_1, \quad \mathcal{L}_d = \mathcal{D}_g(\mathbf{J}_{\hat{g}}(\hat{\mathbf{x}}_t)) + \mathcal{D}_g(\mathbf{J}_r(\hat{\mathbf{z}}_t)). \tag{9}$$

where $\mathcal{D}_g(A) = \text{tr}\left[(I + \frac{1}{d}A \circ A)^d\right] - d$ is the DAG constraint from Yu et al. (2019), with $A$ being an $d$-dimensionality matrix, and $\|\cdot\|_1$ denotes the matrix $l_1$ norm. In summary, the overall loss function of the CaDRe model integrates ELBO and penalties for structural constraints as follows:

$$\mathcal{L}_{ALL} = \mathcal{L}_{ELBO} + \alpha \mathcal{L}_s + \beta \mathcal{L}_d, \tag{10}$$

with setting $\alpha = 1.0 \times 10^{-4}$ and $\beta = 5.0 \times 10^{-5}$ are hyperparameters to yield the best performance.

## 5 EXPERIMENT

Based on the proposed framework, we conduct extensive experiments on both synthetic and real-world climate data to examine the identifiability of the latent process and causal graphs over observations, as well as climate forecasting and scientific interpretability in realistic climate systems.

### 5.1 ON SYNTHETIC CLIMATE DATA

**Baselines.** For CD, we compare CaDRe with several constraint-based methods suited for nonparametric settings. Specifically, we include PC (Spirtes et al., 2001), FCI (Spirtes & Glymour, 1991),

Table 3: **Comparison with Temporal CRL.** We set the dimensions as $d_z = 3$ and $d_x = 10$, considering: (1) *Independent*: $z_{t,i}$ and $z_{t,j}$ are conditionally independent given $\mathbf{z}_{t-1}$; (2) *Sparse*: $z_{t,i}$ and $z_{t,j}$ are dependent given $\mathbf{z}_{t-1}$, but the latent Markov network and time-lagged latent structure are sparse; (3) *Dense*: No sparsity restrictions on latent causal graph. **Bold** numbers indicate the best.

| Setting | Metric | CaDRe | iCITRIS | G-CaRL | CaRiNG | TDRL | LEAP | SlowVAE | PCL | i-VAE | TCL |
|---|---|---|---|---|---|---|---|---|---|---|---|
| **Independent** | MCC | **0.9811** | 0.6649 | 0.8023 | 0.8543 | _0.9106_ | 0.8942 | 0.4312 | 0.6507 | 0.6738 | 0.5916 |
| | $R^2$ | **0.9626** | 0.7341 | _0.9012_ | 0.8355 | 0.8649 | 0.7795 | 0.4270 | 0.4528 | 0.5917 | 0.3516 |
| **Sparse** | MCC | **0.9306** | 0.4531 | _0.7701_ | 0.4924 | 0.6628 | 0.6453 | 0.3675 | 0.5275 | 0.4561 | 0.2629 |
| | $R^2$ | **0.9102** | 0.6326 | 0.5443 | 0.2897 | 0.6953 | 0.4637 | 0.2781 | 0.1852 | 0.2119 | 0.3028 |
| **Dense** | MCC | **0.6750** | 0.3274 | _0.6714_ | 0.4893 | 0.3547 | 0.5842 | 0.1196 | 0.3865 | 0.2647 | 0.1324 |
| | $R^2$ | **0.9204** | 0.6875 | _0.8032_ | 0.4925 | 0.7809 | 0.7723 | 0.5485 | 0.6302 | 0.1525 | 0.2060 |

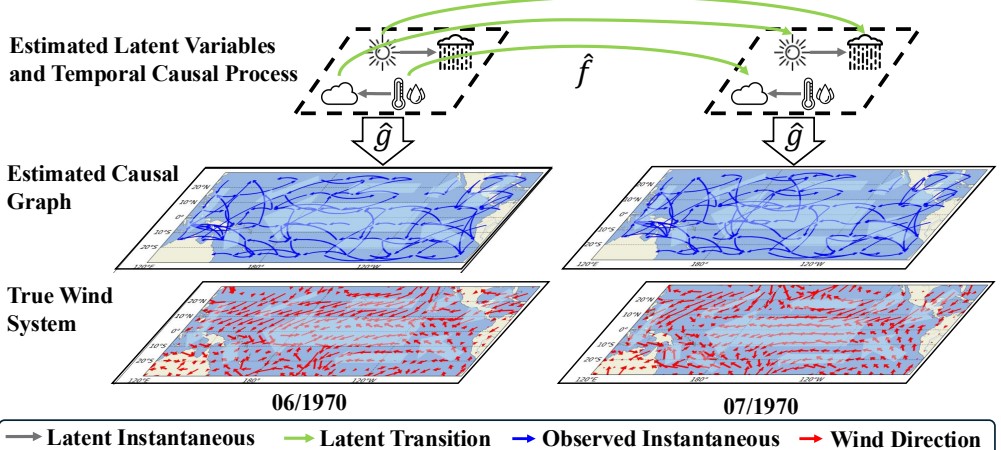

Figure 5: Estimated latent variables, latent causal process, and causal graph over the observed climate grids from CESM2, together with the wind field from (Rasp et al., 2020), where longer red arrows indicate stronger winds. We also draw the overall wind trend in each map to show the consistency. Latent causal structures are consistent across time steps. Unrotated version of climate visualization is shown in A10.

and CD-NOD (Huang et al., 2020), which handle latent confounders, time-series methods including PCMCI (Runge et al., 2019b) and LPCMCI (Gerhardus & Runge, 2020), which account for instantaneous and lagged effects with latent confounding. In CRL, we benchmark against CaRiNG (Chen et al., 2024), TDRL (Yao et al., 2022), LEAP (Yao et al., 2021), SlowVAE (Klindt et al., 2020), PCL (Hyvarinen & Morioka, 2017), i-VAE (Khemakhem et al., 2020), TCL (Hyvarinen & Morioka, 2016), and models that can handle instantaneous effects, including iCITRIS (Lippe et al., 2022) and G-CaRL (Morioka & Hyvärinen, 2023). We use SHD, TPR, and precision as metrics for assessing structure learning, and MCC for evaluating the recovery of latent representations (sources).

**Comparison with Constraint-Based CD.** Figure 4 shows that CaDRe outperforms all baselines across different sample sizes and variable dimensions, while FCI performs poorly when latent confounders are dependent, often leading to low recall. CD-NOD relies on pseudo-causal sufficiency, assuming that latent variables are functions of surrogate variables, which does not hold in general latent settings. PCMCI ignores latent dynamics altogether, while LPCMCI assumes no causal relations among latent confounders, limiting its applicability in complex systems. These comparisons highlight the effectiveness of CaDRe in addressing the limitations of these constraint-based methods.

**Comparison with Temporal CRL.** As presented in Table 3, the MCC and $R^2$ results for the *independent* and *sparse* settings demonstrate that our model achieves component-wise identifiability (Theorem A.3). In contrast, other considered methods fail to recover latent variables, as they cannot properly address cases where the observed variables are causally-related. For the *dense* setting, our approach can recover the latent space (Theorem 1) with the highest $R^2$, while other methods exhibit significant degradation because they are not specifically tailored to handle scenarios involving general noise in the generating function. These outcomes are consistent with our theoretical analysis.

Table 4: **Quantitative Comparison of CD methods on CESM2.** Lower WSHD is better, higher WTPR is better. **Bold** / Underlined numbers represent the best / second-best performance.

| Metric | CaDRe | PC | FCI | CDNOD | PCMCI | LPCMCI | TCDF | TDRL | IDOL |
|---|---|---|---|---|---|---|---|---|---|
| WSHD ↓ | **0.012** | 0.043 | 0.028 | 0.031 | 0.024 | 0.019 | 0.021 | 0.084 | 0.035 |
| WTPR ↑ | **0.532** | 0.202 | 0.302 | 0.251 | 0.198 | 0.274 | 0.327 | 0.246 | 0.150 |
| Latency (ms) ↓ | 1.095$_{\pm 0.203}$ | 2230.72$_{\pm 27.94}$ | 999.09$_{\pm 16.27}$ | 2242.88$_{\pm 27.14}$ | 3391.35$_{\pm 76.64}$ | 3508.50$_{\pm 123.36}$ | 2.327$_{\pm 0.723}$ | **0.974**$_{\pm 0.126}$ | 1.536$_{\pm 0.251}$ |

Table 5: **Results on Temperature Forecasting.** Lower MSE/MAE is better. **Bold** numbers represent the best performance among the models, while underlined numbers denote the second-best.

| Dataset | Length | CaDRe | | TDRL | | CARD | | FITS | | MICN | | iTransformer | | TimesNet | | Autoformer | | Timer-XL | |
|---|---|---|---|---|---|---|---|---|---|---|---|---|---|---|---|---|---|---|---|
| | | MSE ↓ | MAE ↓ | MSE ↓ | MAE ↓ | MSE ↓ | MAE ↓ | MSE ↓ | MAE ↓ | MSE ↓ | MAE ↓ | MSE ↓ | MAE ↓ | MSE ↓ | MAE ↓ | MSE ↓ | MAE ↓ | MSE ↓ | MAE ↓ |
| CESM2 | 96 | 0.410 | 0.483 | 0.439 | 0.507 | **0.409** | 0.484 | 0.439 | 0.508 | 0.417 | 0.486 | 0.422 | 0.491 | 0.415 | 0.486 | 0.959 | 0.735 | 0.433 | **0.425** |
| CESM2 | 192 | **0.412** | **0.487** | 0.440 | 0.508 | 0.422 | 0.493 | 0.447 | 0.515 | 1.559 | 0.984 | 0.425 | 0.495 | 0.417 | 0.497 | 1.574 | 0.972 | 0.454 | 0.524 |
| CESM2 | 336 | **0.413** | **0.485** | 0.441 | 0.505 | 0.421 | 0.497 | 0.482 | 0.536 | 2.091 | 1.173 | 0.426 | 0.494 | 0.423 | 0.499 | 1.845 | 1.078 | 0.527 | 0.565 |
| Weather | 96 | **0.157** | **0.203** | 0.442 | 0.511 | 0.423 | 0.497 | 0.172 | 0.221 | 0.199 | 0.256 | 0.168 | 0.214 | 0.180 | 0.231 | 0.225 | 0.259 | 0.367 | 0.252 |
| Weather | 192 | 0.207 | 0.248 | 0.492 | 0.545 | 0.482 | 0.544 | 0.216 | 0.260 | 0.238 | 0.298 | **0.193** | **0.241** | 0.212 | 0.265 | 0.354 | 0.348 | 0.434 | 0.298 |
| Weather | 336 | **0.270** | **0.314** | 0.536 | 0.612 | 0.525 | 0.596 | 0.386 | 0.439 | 0.316 | 0.496 | 0.426 | 0.494 | 0.423 | 0.499 | 0.354 | 0.348 | 0.527 | 0.565 |
| ERSST | 96 | **0.145** | **0.268** | 0.187 | 0.268 | 0.197 | 0.273 | 0.539 | 0.297 | 0.726 | 0.765 | 0.247 | 0.264 | 0.432 | 0.508 | 0.953 | 0.272 | 0.163 | **0.259** |
| ERSST | 192 | **0.208** | 0.307 | 0.214 | **0.293** | 0.233 | 0.375 | 0.226 | 0.752 | 1.263 | 0.892 | 0.251 | 0.535 | 0.452 | 0.585 | 1.024 | 0.908 | 0.210 | 0.294 |
| ERSST | 336 | **0.305** | 0.361 | 0.462 | 0.388 | 0.487 | 0.484 | 0.439 | 0.535 | 1.173 | 1.172 | 0.305 | 0.659 | 0.581 | 0.607 | 1.387 | 1.353 | 0.352 | **0.337** |

## 5.2 ON REAL-WORLD CLIMATE DATA

**Baselines.** We consider the following state-of-the-art time series forecasting models including Autoformer (Wu et al., 2021), TimesNet (Wu et al., 2022), Timer-XL (Liu et al., 2024c), TimeMixer (Wang et al., 2024a), TimeXer (Wang et al., 2024b), xLSTM-Mixer (Kraus et al., 2024), and MICN (Wang et al., 2022). Moreover, we consider several latest methods for time series analysis like CARD (Wang et al., 2023), FITS (Xu et al., 2024), and iTransformer (Liu et al., 2023). Finally, we consider the temporal CRL methods including TDRL (Yao et al., 2022) and IDOL (Li et al., 2025) to compare the causal structure learning. We publish the average performance over 3 random seeds.

**Scientific Discovery using the Estimated Causal Graphs.** As the ground-truth latent variables and causal graphs are inaccessible in real climate data, we adopt the contemporaneous wind field (Rasp et al., 2020) as a surrogate. As shown in Figure 5, CaDRe recovers causal graphs that align closely with physical wind patterns, providing scientific evidence for the validity of its results. Additionally, large-scale physical patterns produced from CaDRe can unveil new scientific findings, *e.g.*, westward flows in equatorial oceans and a southwestward propagation near Central America, while revealing structurally complex zones along coastal boundaries. These dense, irregular edges may reflect coupled land–atmosphere dynamics or anthropogenic influences (Vautard et al., 2019; Boé & Terray, 2014).

**Quantitative Results on CD.** To quantitatively evaluate CaDRe on real-world CD, we introduce two metrics incorporating wind direction priors: Wind-SHD (WSHD) and Wind-TPR (WTPR). We further compare against neural-based methods, including TCDF (Nauta et al., 2019), as well as TDRL and IDOL, where latent variables are treated as observed. As shown in Table 4, the causal graphs recovered by CaDRe align most closely with physical wind patterns, highlighting its effectiveness on real-world climate data. Moreover, it achieves low latency since it produces causal graphs through one-step generation, demonstrating its efficiency in realistic high-dimensional settings.

**Physical Interpretability of Latent Space.** We endow the learned latent variables with physical interpretability by aligning them with ground-truth physical variables from Rasp et al. (2020) and quantifying component-wise information retrieval using the Subset Mean Correlation Coefficient (SMCC). We visualize the alignment results using icons in Figure 5. Our analysis shows that, during the mid-1970s, precipitation was largely driven by other climate variables such as solar radiation and cloud cover, while humidity influenced cloud cover with a delayed rather than instantaneous effect. These observations are consistent with established scientific evidence (Betts et al., 2014).

**Weather Prediction.** We evaluate our method on three climate datasets for real-world weather forecasting. As summarized in Table 5, our approach outperforms existing time-series forecasting models in precision, due to existing models struggling with causally-related observations and non-contaminated generation, restricting their usability in real-world climate data.

**Additional Experiments.** The appendix provides extended experimental materials as follows: the simulation processes are detailed in Appendix C.1.1, and the climate datasets are described in Appendix C.2.1. Evaluation metrics are presented in Appendix C.1.2, with further results on temperature forecasting given in Table A20. Additional analyses include studies on different dimensions $d_x$

(Table A9) and $d_z$ (Table A11), hyperparameter sensitivity (Table A16), computation comparison (Figure A9), assumption violations (Table A13), and higher-order Markov structures (Table A14).

## 6 CONCLUSION, LIMITATION, AND FUTURE WORK

We focused on advancing the causal understanding of climate science by proposing a causal model that integrates latent processes with directly causally related observed variables. We establish identifiability results and develop an estimation procedure to uncover latent causal variables, latent causal processes, and causal structures among observed variables in the climate system, aiming to provide insight into the underlying "why" questions of climate phenomena. Our simulation experiments substantiate the theoretical results, while analyses on real-world climate data provide a detailed causal interpretation for climate science. In **limitation**, the current implementation has been evaluated only on climate data; extending the framework to other domains remains an important direction for future research. **Future work** will extend this framework to account for time-lagged causal relations in the observed space with sparse transition and/or generation processes, thereby more accurately characterizing how latent variables influence observations.

## ETHICS STATEMENT

Our research does not involve human subjects, personally identifiable data, or sensitive demographic information. All datasets used in this work are publicly available benchmark datasets that do not contain private or restricted content. The proposed methodology is designed for scientific and academic purposes, and we are not aware of foreseeable harmful applications. Therefore, to the best of our knowledge, this study does not raise ethical concerns.

## REPRODUCIBILITY STATEMENT

We have taken several steps to ensure the reproducibility of our results. A complete description of the simulated dataset and preprocessing steps is included in Appendix C.1. The implementation details for real-world experiments, including real-world datasets, model architectures, training hyperparameters, and optimization settings, are provided in Appendix C.2. Extended experiment results are reported in Appendix D.

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

*Supplement to*

# "Learning General Causal Structures with Hidden Dynamic Process for Climate Analysis"

Appendix organization:

# A    THEOREM PROOFS

## A.1    NOTATION LIST

This section collects the notations used in the theorem proofs for clarity and consistency.

Table A6: List of notations, explanations, and corresponding values.

| Index | Explanation | Support |
|---|---|---|
| $d_x$ | number of observed variables | $d_x \in \mathbb{N}^+$ |
| $d_z$ | number of latent variables | $d_z \in \mathbb{N}^+$ and $d_z \leq d_x$ |
| $t$ | time index | $t \in \mathbb{N}^+$ and $t \geq 3$ |
| $\mathcal{I}$ | index set of observed variables | $\mathcal{I} = \{1, 2, \ldots, d_x\}$ |
| $\mathcal{J}$ | index set of latent variables | $\mathcal{J} = \{1, 2, \ldots, d_z\}$ |
| **Variable** | | |
| $\mathcal{X}_t$ | support of observed variables in time-index $t$ | $\mathcal{X}_t \subseteq \mathbb{R}^{d_x}$ |
| $\mathcal{Z}_t$ | support of latent variables | $\mathcal{Z}_t \subseteq \mathbb{R}^{d_z}$ |
| $\mathbf{x}_t$ | observed variables in time-index $t$ | $\mathbf{x}_t \in \mathcal{X}_t$ |
| $\mathbf{z}_t$ | latent variables in time-index $t$ | $\mathbf{z}_t \in \mathcal{Z}_t$ |
| $\mathbf{s}_t$ | dependent noise of observations in time-index $t$ | $\mathbf{s}_t \in \mathbb{R}^{d_x}$ |
| $\boldsymbol{\epsilon}_{\mathbf{x}_t}$ | independent noise for generating $\mathbf{s}_t$ in time-index $t$ | $\boldsymbol{\epsilon}_{\mathbf{x}_t} \sim p_{\epsilon_x}$ |
| $\boldsymbol{\epsilon}_{\mathbf{z}_t}$ | independent noise of latent variables in time-index $t$ | $\boldsymbol{\epsilon}_{\mathbf{z}_t} \sim p_{\epsilon_z}$ |
| $\mathbf{z}_{t \setminus [i,j]}$ | latent variables except for $z_{t,i}$ and $z_{t,j}$ in time-index $t$ | / |
| **Function** | | |
| $p_{a\mid b}(\cdot \mid b)$ | density function of $a$ given $b$ | / |
| $p_{a,b\mid c}(a, \cdot \mid c)$ | joint density function of $(a, b)$ given $a$ and $c$ | / |
| $\mathbf{pa}(\cdot)$ | variable's parents | / |
| $\mathbf{pa}_O(\cdot)$ | variable's parents in observed space | / |
| $\mathbf{pa}_L(\cdot)$ | variable's parents in latent space | / |
| $g(\cdot)$ | generating function of SEM from $(\mathbf{z}_t, \mathbf{s}_t, \mathbf{x}_t)$ to $\mathbf{x}_t$ | $\mathbb{R}^{d_z+2d_x} \to \mathbb{R}^{d_x}$ |
| $m(\cdot)$ | mixing function of ICA from $(\mathbf{z}_t, \mathbf{s}_t)$ to $\mathbf{x}_t$ | $\mathbb{R}^{d_z+d_x} \to \mathbb{R}^{d_x}$ |
| $h_z(\cdot)$ | invertible transformation from $\mathbf{z}_t$ to $\hat{\mathbf{z}}_t$ | $\mathbb{R}^{d_z} \to \mathbb{R}^{d_z}$ |
| $\pi(\cdot)$ | permutation function | $\mathbb{R}^{d_x} \to \mathbb{R}^{d_x}$ |
| $\mathrm{supp}(\cdot)$ | support matrix of Jacobian matrix | $\mathbb{R}^{d_x \times d_x} \to \{0,1\}^{d_x \times d_x}$ |
| **Symbol** | | |
| $A \to B$ | $A$ causes $B$ directly | / |
| $A \dashrightarrow B$ | $A$ causes $B$ indirectly | / |
| $\mathbf{J}_g(\mathbf{x}_t)$ | Jacobian matrix representing observed causal DAG | $\mathbf{J}_g(\mathbf{x}_t) \in \mathbb{R}^{d_x \times d_x}$ |
| $\mathbf{J}_g(\mathbf{x}_t, \mathbf{s}_t)$ | Jacobian matrix representing mixing structure from $(\mathbf{x}_t, \mathbf{s}_t)$ to $\mathbf{x}_t$ | $\mathbf{J}_g(\mathbf{x}_t, \mathbf{s}_t) \in \mathbb{R}^{d_x \times d_x}$ |
| $\mathbf{J}_m(\mathbf{s}_t)$ | Jacobian matrix representing mixing structure from $\mathbf{s}_t$ to $\mathbf{x}_t$ | $\mathbf{J}_m(\mathbf{s}_t) \in \mathbb{R}^{d_x \times d_x}$ |
| $\mathbf{J}_r(\mathbf{z}_{t-1})$ | Jacobian matrix representing latent time-lagged structure | $\mathbf{J}_r(\mathbf{z}_{t-1}) \in \mathbb{R}^{d_z \times d_z}$ |
| $\mathbf{J}_r(\mathbf{z}_t)$ | Jacobian matrix representing instantaneous latent causal graph | $\mathbf{J}_r(\mathbf{z}_t) \in \mathbb{R}^{d_z \times d_z}$ |

## A.2    PROOF OF THEOREM 1

We first introduce another operator to represent the point-wise distributional multiplication. To maintain generality, we denote two variables as $a$ and $b$, with respective support sets $\mathcal{A}$ and $\mathcal{B}$.

**Definition 1.** *(Diagonal Operator) Consider two random variable $a$ and $b$, density functions $p_a$ and $p_b$ are defined on some support $\mathcal{A}$ and $\mathcal{B}$, respectively. The diagonal operator $D_{b\mid a}$ maps the density function $p_a$ to another density function $D_{b\mid a} \circ p_a$ defined by the pointwise multiplication of the function $p_{b\mid a}$ at a fixed point $b$:*

$$p_{b\mid a}(b \mid \cdot)p_a = D_{b\mid a} \circ p_a, \text{where } D_{b\mid a} = p_{b\mid a}(b \mid \cdot). \tag{A1}$$

*Proof.* $\mathbf{x}_{t-1}, \mathbf{x}_t, \mathbf{x}_{t+1}$ are conditional independent given $\mathbf{z}_t$, which implies two equations:

$$p(\mathbf{x}_{t-1} \mid \mathbf{x}_t, \mathbf{z}_t) = p(\mathbf{x}_{t-1} \mid \mathbf{z}_t), \quad p(\mathbf{x}_{t+1} \mid \mathbf{x}_t, \mathbf{x}_{t-1}, \mathbf{z}_t) = p(\mathbf{x}_{t+1} \mid \mathbf{z}_t). \tag{A2}$$

We can obtain $p(\mathbf{x}_{t+1}, \mathbf{x}_t \mid \mathbf{x}_{t-1})$ directly from the observations, $p(\mathbf{x}_{t-1})$ and $p(\mathbf{x}_{t+1}, \mathbf{x}_t, \mathbf{x}_{t-1})$, and then the transformation in density function are established by

$$p(\mathbf{x}_{t+1}, \mathbf{x}_t \mid \mathbf{x}_{t-1}) = \underbrace{\int_{\mathcal{Z}_t} p(\mathbf{x}_{t+1}, \mathbf{x}_t, \mathbf{z}_t \mid \mathbf{x}_{t-1}) d\mathbf{z}_t}_{\text{integration over } \mathcal{Z}_t} = \underbrace{\int_{\mathcal{Z}_t} p(\mathbf{x}_{t+1} \mid \mathbf{x}_t, \mathbf{z}_t, \mathbf{x}_{t-1}) p(\mathbf{x}_t, \mathbf{z}_t \mid \mathbf{x}_{t-1}) d\mathbf{z}_t}_{\text{factorization of joint conditional probability}}$$

$$= \underbrace{\int_{\mathcal{Z}_t} p(\mathbf{x}_{t+1} \mid \mathbf{z}_t) p(\mathbf{x}_t, \mathbf{z}_t \mid \mathbf{x}_{t-1}) d\mathbf{z}_t}_{\text{by } p(\mathbf{x}_{t+1}|\mathbf{x}_t,\mathbf{x}_{t-1},\mathbf{z}_t)=p(\mathbf{x}_{t+1}|\mathbf{z}_t)} = \underbrace{\int_{\mathcal{Z}_t} p(\mathbf{x}_{t+1} \mid \mathbf{z}_t) p(\mathbf{x}_t \mid \mathbf{z}_t) p(\mathbf{z}_t \mid \mathbf{x}_{t-1}) d\mathbf{z}_t}_{\text{by } p(\mathbf{x}_{t-1}|\mathbf{x}_t,\mathbf{z}_t)=p(\mathbf{x}_{t-1}|\mathbf{z}_t)} .$$

(A3)

Then we show how to transform the Eq. (A3) to the form of spectral decomposition:

$$\implies \int_{\mathcal{X}_{t-1}} p(\mathbf{x}_{t+1}, \mathbf{x}_t \mid \mathbf{x}_{t-1}) p(\mathbf{x}_{t-1}) d\mathbf{x}_{t-1} =$$

$$\int_{\mathcal{X}_{t-1}} \int_{\mathcal{Z}_t} p(\mathbf{x}_{t+1} \mid \mathbf{z}_t) p(\mathbf{x}_t \mid \mathbf{z}_t) p(\mathbf{z}_t \mid \mathbf{x}_{t-1}) p(\mathbf{x}_{t-1}) d\mathbf{z}_t d\mathbf{x}_{t-1} \qquad \text{(A4)}$$

$$\implies [L_{\mathbf{x}_t;\mathbf{x}_{t+1}|\mathbf{x}_{t-1}} p](\mathbf{x}_{t+1}) = [L_{\mathbf{x}_{t+1}|\mathbf{z}_t} D_{\mathbf{x}_t|\mathbf{z}_t} L_{\mathbf{z}_t|\mathbf{x}_{t-1}} p](\mathbf{x}_{t+1}), \qquad \text{(A5)}$$

$$\implies L_{\mathbf{x}_t;\mathbf{x}_{t+1}|\mathbf{x}_{t-1}} = L_{\mathbf{x}_{t+1}|\mathbf{z}_t} D_{\mathbf{x}_t|\mathbf{z}_t} L_{\mathbf{z}_t|\mathbf{x}_{t-1}} \qquad \text{(A6)}$$

$$\implies \int_{\mathbf{x}_t \in \mathcal{X}_t} L_{\mathbf{x}_t;\mathbf{x}_{t+1}|\mathbf{x}_{t-1}} d\mathbf{x}_t = \int_{\mathbf{x}_t \in \mathcal{X}_t} L_{\mathbf{x}_{t+1}|\mathbf{z}_t} D_{\mathbf{x}_t|\mathbf{z}_t} L_{\mathbf{z}_t|\mathbf{x}_{t-1}} d\mathbf{x}_t \qquad \text{(A7)}$$

$$\implies L_{\mathbf{x}_{t+1}|\mathbf{x}_{t-1}} = L_{\mathbf{x}_{t+1}|\mathbf{z}_t} L_{\mathbf{z}_t|\mathbf{x}_{t-1}} \qquad \text{(A8)}$$

$$\implies L_{\mathbf{x}_{t+1}|\mathbf{z}_t}^{-1} L_{\mathbf{x}_{t+1}|\mathbf{x}_{t-1}} = L_{\mathbf{z}_t|\mathbf{x}_{t-1}} \qquad \text{(A9)}$$

$$\implies L_{\mathbf{x}_t;\mathbf{x}_{t+1}|\mathbf{x}_{t-1}} = L_{\mathbf{x}_{t+1}|\mathbf{z}_t} D_{\mathbf{x}_t|\mathbf{z}_t} L_{\mathbf{x}_{t+1}|\mathbf{z}_t}^{-1} L_{\mathbf{x}_{t+1}|\mathbf{x}_{t-1}} \qquad \text{(A10)}$$

$$\implies L_{\mathbf{x}_t;\mathbf{x}_{t+1}|\mathbf{x}_{t-1}} L_{\mathbf{x}_{t+1}|\mathbf{x}_{t-1}}^{-1} = L_{\mathbf{x}_{t+1}|\mathbf{z}_t} D_{\mathbf{x}_t|\mathbf{z}_t} L_{\mathbf{x}_{t+1}|\mathbf{z}_t}^{-1} . \qquad \text{(A11)}$$

$$\implies L_{\mathbf{x}_{t+1}|\mathbf{z}_t} D_{\mathbf{x}_t|\mathbf{z}_t} L_{\mathbf{x}_{t+1}|\mathbf{z}_t}^{-1} = (C L_{\mathbf{x}_{t+1}|\mathbf{z}_t} P)(P^{-1} D_{\mathbf{x}_t|\mathbf{z}_t} P)(P^{-1} L_{\mathbf{x}_{t+1}|\mathbf{z}_t}^{-1} C^{-1})$$

(A12)

$$\implies L_{\mathbf{x}_{t+1}|\mathbf{z}_t} = C L_{\mathbf{x}_{t+1}|\hat{\mathbf{z}}_t} P, \quad D_{\mathbf{x}_t|\mathbf{z}_t} = P^{-1} D_{\mathbf{x}_t|\hat{\mathbf{z}}_t} P \qquad \text{(A13)}$$

where

- in Eq. (A4), we add the integration over $\mathcal{X}_{t-1}$ in both sides of Eq. (A3). s
- in Eq. (A5), we replace the probability with operators by using Eq. (2) and Definition 1. Specifically, we have: $L_{\mathbf{x}_t;\mathbf{x}_{t+1}|\mathbf{x}_{t-1}} = \int_{\mathcal{X}_{t-1}} p_{\mathbf{x}_{t+1}}(\mathbf{x}_t, \cdot \mid \mathbf{x}_{t-1}) p(\mathbf{x}_{t-1}) d\mathbf{x}_{t-1}$.
- in Eq. (A9), the operator $L_{\mathbf{x}_{t+1}|\mathbf{z}_t}$ is injective by Assumption 1
- in Eq. (A10), the $L_{\mathbf{z}_t|\mathbf{x}_{t-1}}$ in Eq. (A6) is substituted by Eq. (A9):
- in Eq. (A11), if $L_{\mathbf{x}_{t-1}|\mathbf{x}_{t+1}}$ is injective, then $L_{\mathbf{x}_{t+1}|\mathbf{x}_{t-1}}^{-1}$ exists and is densely defined over $\mathcal{F}(\mathcal{X}_{t+1})$.
- in Eq. (A13), Assumption 1 ensures that $L_{\mathbf{x}_t;\mathbf{x}_{t+1}|\mathbf{x}_{t-1}} L_{\mathbf{x}_{t+1}|\mathbf{x}_{t-1}}^{-1}$ is bounded; by the uniqueness of spectral decomposition (see e.g., (Conway, 1994) Ch. VII and (Dunford & Schwartz, 1971) Theorem XV 4.5), $L_{\mathbf{x}_{t+1}|\mathbf{z}_t} D_{\mathbf{x}_t|\mathbf{z}_t} L_{\mathbf{x}_{t+1}|\mathbf{z}_t}^{-1}$ admits a unique spectral decomposition in which the eigenvalues, *i.e.*, $D_{\mathbf{x}_t|\mathbf{z}_t}$, which are precisely the entries of $\{p_{\mathbf{x}_t|\mathbf{z}_t}(\mathbf{x}_t \mid \mathbf{z}_t)\}$, and eigenfunctions, *i.e.*, $D_{\mathbf{x}_t|\mathbf{z}_t}$, which columns are $\{p_{\mathbf{x}_{t+1}|\mathbf{z}_t}(\cdot \mid \mathbf{z}_t)\}$, up to standard indeterminacies. $C$ is an nonzero scalar rescaling eigenvalues, and $P$ is a operator permuting the eigenvalues and eigenfunctions.

We obtain a unique spectral decomposition in Eq. (A13) with permutation and scaling indeterminacies. In the following, we will show how these indeterminacies can be resolved—if not, what informative results can still be inferred.

First, considering the arbitrary scaling $C$, since the normalizing condition

$$\int_{\mathcal{X}_{t+1}} p_{\mathbf{x}_{t+1}|\hat{\mathbf{z}}_t} d\mathbf{x}_{t+1} = 1 \qquad \text{(A14)}$$

must hold for every $\hat{\mathbf{z}}_t$, one only solution of $\int_{\mathcal{X}_{t+1}} C p_{\mathbf{x}_{t+1}|\mathbf{z}_t} d\mathbf{x}_{t+1} = 1$ is to set $C = 1$.

Second, regarding the permutation indeterminacy, we start from $D_{\mathbf{x}_t|\mathbf{z}_t} = P^{-1}D_{\mathbf{x}_t|\hat{\mathbf{z}}_t}P$. The operator, $D_{\mathbf{x}_t|\mathbf{z}_t}$, corresponding to the set $\{p_{\mathbf{x}_t|\mathbf{z}_t}(\mathbf{x}_t \mid \mathbf{z}_t)\}$ for fixed $\mathbf{x}_t$ and all $\mathbf{z}_t$, admits a unique solution ($P$ only change the entry position):

$$\{p_{\mathbf{x}_t|\mathbf{z}_t}(\mathbf{x}_t \mid \mathbf{z}_t)\} = \{p_{\mathbf{x}_t|\hat{\mathbf{z}}_t}(\mathbf{x}_t \mid \hat{\mathbf{z}}_t)\}, \quad \text{for all } \mathbf{z}_t, \hat{\mathbf{z}}_t \tag{A15}$$

Due to the set is unorder, the only way to match the R.H.S. with the L.H.S. in a consistent order is to exchange the conditioning variables, that is,

$$\{p_{\mathbf{x}_t|\mathbf{z}_t}(\mathbf{x}_t \mid \mathbf{z}_t^{(1)}), p_{\mathbf{x}_t|\mathbf{z}_t}(\mathbf{x}_t \mid \mathbf{z}_t^{(2)}), \ldots\} = \{p_{\mathbf{x}_t|\hat{\mathbf{z}}_t}(\mathbf{x}_t \mid \hat{\mathbf{z}}_t^{(1)}), p_{\mathbf{x}_t|\hat{\mathbf{z}}_t}(\mathbf{x}_t \mid \hat{\mathbf{z}}_t^{(2)}), \ldots\} \tag{A16}$$

$$\implies \quad [p_{\mathbf{x}_t|\mathbf{z}_t}(\mathbf{x}_t \mid \mathbf{z}_t^{(\pi(1))}), p_{\mathbf{x}_t|\mathbf{z}_t}(\mathbf{x}_t \mid \mathbf{z}_t^{(\pi(2))}), \ldots] = [p_{\mathbf{x}_t|\hat{\mathbf{z}}_t}(\mathbf{x}_t \mid \hat{\mathbf{z}}_t^{(\pi(1))}), p_{\mathbf{x}_t|\hat{\mathbf{z}}_t}(\mathbf{x}_t \mid \hat{\mathbf{z}}_t^{(\pi(2))}), \ldots] \tag{A17}$$

where superscript $(\cdot)$ denotes the index of a conditioning variable, and $\pi$ is reindexing the conditioning variables. We use a relabeling map $h$ to represent its corresponding value mapping:

$$p_{\mathbf{x}_t|\mathbf{z}_t}(\mathbf{x}_t \mid h(\mathbf{z}_t)) = p_{\mathbf{x}_t|\hat{\mathbf{z}}_t}(\mathbf{x}_t \mid \hat{\mathbf{z}}_t), \quad \text{for all } \mathbf{z}_t, \hat{\mathbf{z}}_t. \tag{A18}$$

By Assumption 1, different $\mathbf{z}_t$ corresponds to different $p_{\mathbf{x}_t|\mathbf{z}_t}(\mathbf{x}_t \mid \mathbf{z}_t)$, there is no repeated element in $\{p_{\mathbf{x}_t|\mathbf{z}_t}(\mathbf{x}_t \mid \mathbf{z}_t)\}$ (and $\{p_{\mathbf{x}_t|\hat{\mathbf{z}}_t}(\mathbf{x}_t \mid \hat{\mathbf{z}}_t)\}$). Hence, the relabelling map $h$ is one-to-one (invertible). Furthermore, Assumption 4 implies that $p_{\mathbf{x}_t|\mathbf{z}_t}(\mathbf{x}_t \mid h(\mathbf{z}_t))$ determines a unique $h(\mathbf{z}_t)$. The same holds for the $p_{\mathbf{x}_t|\hat{\mathbf{z}}_t}(\mathbf{x}_t \mid \hat{\mathbf{z}}_t)$, implying that

$$p_{\mathbf{x}_t|\mathbf{z}_t}(\mathbf{x}_t \mid h(\mathbf{z}_t)) = p_{\mathbf{x}_t|\hat{\mathbf{z}}_t}(\mathbf{x}_t \mid \hat{\mathbf{z}}_t) \implies \hat{\mathbf{z}}_t = h(\mathbf{z}_t). \tag{A19}$$

Next, Assumption 1 implies that the function $h$ must be differentiable. Since the VAE is differentiable, we can learn a differentiable function $h$ that satisfies Assumption 1. Consider $\hat{\mathbf{z}}_t$ related to $\mathbf{z}_t$ via $\hat{\mathbf{z}}_t = h(\mathbf{z}_t)$. Then, we have

$$F\left[p_{\mathbf{x}_t|\hat{\mathbf{z}}_t}(\cdot \mid \mathbf{z}_t)\right] = F\left[p_{\mathbf{x}_t|\mathbf{z}_t}(\cdot \mid h(\mathbf{z}_t))\right] = h(\mathbf{z}_t), \tag{A20}$$

which is equal to $\hat{\mathbf{z}}_t$ only if $h$ is differentiable.

To ensure the latent dimension $d_z$ is also identifiable, we analyze two scenarios :

i. $d_{\hat{z}} > d_z$: $d_z$ latent components in $\hat{\mathbf{z}}_t$ are sufficient to explain $\mathbf{x}_t$, *i.e.*,

$$p(\mathbf{x}_t \mid \mathbf{z}_{t,:d_{\hat{z}}-d_z}, \mathbf{z}_{t,d_{\hat{z}}-d_z:}^{(1)}) = p(\mathbf{x}_t \mid \mathbf{z}_{t,:d_{\hat{z}}-d_z}, \mathbf{z}_{t,d_{\hat{z}}-d_z:}^{(2)}), \tag{A21}$$

which contradicts the Assumption 1.

ii. $d_{\hat{z}} < d_z$: This suggests that only $d_{\hat{z}}$ dimensions are sufficient to reconstruct $\mathbf{x}_t$, leaving $d_z - d_{\hat{z}}$ components constant, which violates that there are $d_z$ latent *variables*.

$\square$

**More Discussions of Assumption 1** The injectivity of the operator enables us to take inverses of certain operators, which is commonly made in nonparametric identification (Hu & Schennach, 2008; Carroll et al., 2010; Hu & Shum, 2012). Intuitively, different input distribution correpsonds to different output distribution. In the context of the climate system, it represent the necessity of temporal variability. However, it is difficult to formalize it in terms of functions. We give some examples in terms of $p_a \Rightarrow p_b$ to make it understandable:

**Example 1.** *$b = g(a)$, where $g$ is an invertible function.*

**Example 2.** *$b = a + \epsilon$, where $p(\epsilon)$ must not vanish everywhere after the Fourier transform (Theorem 2.1 in Mattner (1993)).*

**Example 3.** *$b = g(a) + \epsilon$, where the same conditions from Examples 1 and 2 are required.*

**Example 4.** *$b = g_1(g_2(a) + \epsilon)$, a post-nonlinear model with invertible nonlinear functions $g_1, g_2$, combining the assumptions in **Examples 1-3**.*

**Example 5.** *$b = g(a, \epsilon)$, where the joint distribution $p(a, b)$ follows an exponential family.*

**Example 6.** *$b = g(a, \epsilon)$, a general nonlinear formulation. Certain deviations from the nonlinear additive model (**Example 3**), e.g., polynomial perturbations, can still be tractable.*

### A.3 COMPONENT-WISE IDENTIFIABILITY OF LATENT VARIABLES

**Theorem A1.** *(Component-Wise Identifiability of Latent Variables (Li et al., 2025)) Let $\mathbf{c}_t = \{\mathbf{z}_{t-1}, \mathbf{z}_t\}$ and $\mathcal{M}_{\mathbf{c}_t}$ be the variable set of two consecutive timestamps and the corresponding Markov network, respectively. Suppose the following assumptions hold:*

    *i.* *(Smooth and Positive Density): The probability function of the latent variables $\mathbf{c}_t$ is smooth and positive, i.e., $p_{\mathbf{c}_t}$ is third-order differentiable and $p_{\mathbf{c}_t} > 0$ over $\mathbb{R}^{2n}$.*

    *ii.* *(Sufficient Variability)s: Denote $|\mathcal{M}_{\mathbf{c}_t}|$ as the number of edges in Markov network $\mathcal{M}_{\mathbf{c}_t}$. Let*

$$w(m) = \left( \frac{\partial^3 \log p(\mathbf{c}_t|\mathbf{z}_{t-2})}{\partial c_{t,1}^2 \partial z_{t-2,m}}, \dots, \frac{\partial^3 \log p(\mathbf{c}_t|\mathbf{z}_{t-2})}{\partial c_{t,2n}^2 \partial z_{t-2,m}} \right) \oplus$$

$$\left( \frac{\partial^2 \log p(\mathbf{c}_t|\mathbf{z}_{t-2})}{\partial c_{t,1} \partial z_{t-2,m}}, \dots, \frac{\partial^2 \log p(\mathbf{c}_t|\mathbf{z}_{t-2})}{\partial c_{t,2n} \partial z_{t-2,m}} \right) \oplus \left( \frac{\partial^3 \log p(\mathbf{c}_t|\mathbf{z}_{t-2})}{\partial c_{t,i} \partial c_{t,j} \partial z_{t-2,m}} \right)_{(i,j) \in \mathcal{E}(\mathcal{M}_{\mathbf{c}_t})},$$
$$\tag{A22}$$

*where $\oplus$ denotes the concatenation operation and $(i, j) \in \mathcal{E}(\mathcal{M}_{\mathbf{c}_t})$ denotes all pairwise indices such that $c_{t,i}, c_{t,j}$ are adjacent in $\mathcal{M}_{\mathbf{c}_t}$. For $m \in \{1, \dots, n\}$, there exist $4n + 2|\mathcal{M}_{\mathbf{c}_t}|$ different values of $\mathbf{z}_{t-2,m}$ as the $4n + 2|\mathcal{M}_{\mathbf{c}_t}|$ values of vector functions $w(m)$ are linearly independent.*

    *iii.* *(Sparse Latent Process): For any $z_{t,i} \in \mathbf{z}_t$, the intimate neighbor set of $z_{t,i}$ is an empty set, where the intimate neighbor set is defined as*

        **Definition 2.** *(Intimate Neighbor Set) Consider a Markov network $\mathcal{M}_Z$ over variables set $Z$, and the intimate neighbor set of variable $z_{t,i}$ is*

$$\Psi_{\mathcal{M}_{\mathbf{c}_t}}(c_{t,i}) = \left\{ c_{t,j} \, \middle| \, \begin{array}{l} c_{t,j} \text{ is adjacent to } c_{t,i} \text{ and also adjacent} \\ \text{to all other neighbors of } c_{t,i}, \ c_{t,j} \in \mathbf{c}_t \setminus \{c_{t,i}\} \end{array} \right\} \tag{A23}$$

    *iv.* *(Transition Variability): For any pair of adjacent latent variables $z_{t,i}, z_{t,j}$ at time step $t$, their time-delayed parents are not identical, i.e., $\mathbf{pa}(z_{t,i}) \neq \mathbf{pa}(z_{t,j})$.*

*Then for any two different entries $\hat{c}_{t,k}, \hat{c}_{t,l} \in \hat{\mathbf{c}}_t$ that are **not adjacent** in the Markov network $\mathcal{M}_{\hat{\mathbf{c}}_t}$ over estimated $\hat{\mathbf{c}}_t$,*

    *i. The estimated Markov network $\mathcal{M}_{\hat{\mathbf{c}}_t}$ is isomorphic to the ground-truth Markov network $\mathcal{M}_{\mathbf{c}_t}$.*

    *ii. There exists a permutation $\pi$ of the estimated latent variables, such that $z_{t,i}$ and $\hat{z}_{t,\pi(i)}$ is one-to-one corresponding, i.e., $z_{t,i}$ is component-wise identifiable.*

    *iii. The causal graph of the latent causal process is identifiable.*

**Proof Sketch and Discussions.** Once latent space is recovered by Theorem 1, *i.e.*, *(i)* $\hat{\mathbf{z}}_t = h_z(\mathbf{z}_t)$ is established, leveraging two properties of latent space—namely, *(ii)* the sparsity in the latent Markov network, and *(iii)* $z_{t,i} \perp\!\!\!\perp z_{t,j} \mid \mathbf{z}_{t-1}, \mathbf{z}_{t/[i,j]}$ if $z_{t,i}, z_{t,j}$ $(i \neq j)$ are not adjacent in Markov network, we can obtain the component-wise identifiability of latent variables under *sufficient variability* assumption. In contrast to prior work on CRL with component-wise identifiability (Zhang et al., 2024; Li et al., 2025), which assumes multiple distributions or temporal steps while only using a single measurement for the latent space recovery, typically requiring the full invertible mapping $g$ to recover the $\hat{\mathbf{z}}$ in a block-wise manner, our approach fully exploits the temporally adjacent measurements, thereby avoiding the need for the strong assumptions of invertible/deterministic $g$.

### A.4 PROOF OF LEMMA 2

**Definition 3.** *(Causal Order) $x_{t,i}$ is in the $\tau$-th causal order if only observed variables in the $(\tau - 1)$-th causal order directly influence it. We specify $\mathbf{z}_t$ is in the $0$-th causal order.*

For an observed variable $x_{t,i}$, we define the set $\mathcal{P}$ to include all variables in $\mathbf{x}_t$ involved in generating $x_{t,i}$, initialized as $\mathcal{P} = \mathbf{pa}_O(x_{t,i})$. The upper bound of the cardinality of $\mathcal{P}$ is given by $\mathcal{U}(|\mathcal{P}|)$, which satisfies $\mathcal{U}(|\mathcal{P}|) = d_x - 1$ initially. Let $\mathcal{Q}$ denote the set of latent variables, and define the separated set as $\mathcal{S}$, where $g_{s_i}(\mathbf{pa}_L(x_{t,i}), \epsilon_{x_{t,i}})$ is denoted by $s_{t,i}$. Initially, $\mathcal{S} = \{s_{t,i}\}$. We express $x_{t,i}$ as $x_{t,i} = g_i(\mathcal{P}, \mathcal{S}, \mathcal{Q})$, and traverse all $x_{t,j} \in \mathbf{x}_t$ in descending causal order $\tau_j$, performing the following operations:

    i. Remove $x_{t,j}$ from $\mathcal{P}$ and apply Eq. (1) to obtain

$$x_{t,i} = f_1\left(\mathcal{P} \setminus \{x_{t,j}\}, \mathcal{S}, \mathcal{Q}, \mathbf{pa}_O(x_{t,j}), \mathbf{pa}_L(x_{t,j}), s_{t,j}\right). \tag{A24}$$

    Then, update $\mathcal{P} \leftarrow (\mathcal{P} \setminus \{x_{t,j}\}) \cup \mathbf{pa}_O(x_{t,j})$ and $\mathcal{Q} \leftarrow \mathcal{Q} \cup \mathbf{pa}_L(x_{t,j})$. By Assumption 1, $x_{t,j}$ cannot reappear in the set of its ancestors, resulting in $\mathcal{U}(|\mathcal{P}|) \leftarrow \mathcal{U}(|\mathcal{P}|) - 1$.

ii. Assumption 1 also ensures that a variable with a lower causal order does not appear in the generation of its descendants. Hence, $x_{t,j}$ cannot appear in the generation of its descendants, since their causal orders are larger than $\tau_j$. Similarly, $s_{t,j}$, which is involved in generating $x_{t,j}$, does not appear in the generation of its descendants. Thus, $s_{t,j} \notin \mathcal{S}$. Define the new separated set as $\mathcal{S} \leftarrow \mathcal{S} \cup \{s_{t,j}\}$, giving

$$x_{t,i} = f_2\left(\mathcal{P}, \mathcal{S}, \mathcal{Q}\right), \tag{A25}$$

where the new cardinality is updated as $|\mathcal{S}| \leftarrow |\mathcal{S}| + 1$.

Given that $\mathcal{U}(|\mathcal{P}|) \geq |\mathcal{P}|, \mathcal{U}(|\mathcal{P}|)$ ensures that this iterative process can be performed until $|\mathcal{P}| = 0$. According to the definition of data generating process, all the aforementioned functions are partially differentiable w.r.t. $\mathbf{s}_t$ and $\mathbf{x}_t$, or they are compositions of such functions. As a result, $\mathcal{Q} = \mathbf{an}_{\mathbf{z}_t}(x_{t,i})$, and there exists a function $g_{m_i}$ such that

$$x_{t,i} = g_{m_i}(\mathbf{an}_{\mathbf{z}_t}(x_{t,i}), \mathbf{s}_t).$$

Moreover, we observe that $\mathbf{s}_t$ is in fact the ancestors $\mathbf{an}_{\epsilon_{\mathbf{x}_t}}(x_{t,i}) = \{\epsilon_{x_{t,j}} \mid s_{t,j} \in \mathcal{S}\}$, which are implied in this derivation process since $\epsilon_{x_{t,j}}$ is in one-to-one correspondence with $s_{t,j}$ through indexing.

### A.5 PROOF OF THEOREM 2

Considering the mixing function $m$, and the functional relation $s_{t,j} \to x_{t,i}$, corresponding $[\mathbf{J}_m(\mathbf{s}_t)]_{i,j}$, where $i, j$ indicates the row and column index of the Jacobian matrix, respectively.

**For the elements $i \neq j$:** If there is a directed functional relationship $x_{t,j} \to x_{t,i}$, the corresponding element of the Jacobian matrix is $\frac{\partial x_{t,i}}{\partial x_{t,j}}$. If the relationship is indirect: $x_{t,j} \dashrightarrow x_{t,i}$, then for each $x_{t,k} \in \mathbf{pa}_O(x_{t,i})$, there must exist either an indirect-direct path $x_{t,j} \dashrightarrow x_{t,k} \to x_{t,i}$ or a direct-direct path $x_{t,j} \to x_{t,k} \to x_{t,i}$. In summary, through the chain rule, we obtain

$$[\mathbf{J}_m(\mathbf{s}_t)]_{i,j} = \sum_{x_{t,k} \in \mathbf{pa}_O(x_{t,i})} \frac{\partial x_{t,i}}{\partial x_{t,k}} \cdot \frac{\partial x_{t,k}}{\partial s_{t,j}}. \tag{A26}$$

For each $x_{t,k} \notin \mathbf{pa}_O(x_{t,i}), \frac{\partial x_{t,i}}{\partial x_{t,k}} = 0$, Eq. (A26) could be rewritten as

$$[\mathbf{J}_m(\mathbf{s}_t)]_{i,j} = \sum_{x_{t,k} \in \mathbf{pa}_O(x_{t,i})} \frac{\partial x_{t,i}}{\partial x_{t,k}} \cdot \frac{\partial x_{t,k}}{\partial s_{t,j}} + \sum_{x_{t,k} \notin \mathbf{pa}_O(x_{t,i})} \frac{\partial x_{t,i}}{\partial x_{t,k}} \cdot \frac{\partial x_{t,k}}{\partial s_{t,j}}$$
$$= \sum_{k=1}^{d_x} \frac{\partial x_{t,i}}{\partial x_{t,k}} \cdot \frac{\partial x_{t,k}}{\partial s_{t,j}} = \sum_{k=1}^{d_x}[\mathbf{J}_g(\mathbf{x}_t)]_{i,k} \cdot [\mathbf{J}_m(\mathbf{s}_t)]_{k,j}. \tag{A27}$$

**For the elements $i = j$:** For each $x_{t,k} \in \mathbf{pa}_O(x_{t,i})$, DAG structure ensures that $x_{t,i}$ does not appear in the set of ancestors of itself. Consequently, due to the one-to-one correspondence between $s_{t,k}$ and $x_{t,i}$, we also have that $\frac{\partial x_{t,k}}{\partial s_{t,i}} = 0$. Thus, we obtain

$$[\mathbf{J}_m(\mathbf{s}_t)]_{i,i} = \frac{\partial x_{t,i}}{\partial s_{t,i}} + 0 = \frac{\partial x_{t,i}}{\partial s_{t,i}} + \sum_{k=1}^{d_x}[\mathbf{J}_g(\mathbf{x}_t)]_{i,k} \cdot [\mathbf{J}_m(\mathbf{s}_t)]_{k,i}. \tag{A28}$$

Since for $k = i$, it holds that $[\mathbf{J}_g(\mathbf{x}_t)]_{i,k} = 0$, and for $k \neq i$, we have $[\mathbf{J}_m(\mathbf{s}_t)]_{k,i} = 0$.
Defining $\mathbf{D}_m(\mathbf{s}_t) = \text{diag}(\frac{\partial x_{t,1}}{\partial s_{t,1}}, \ldots, \frac{\partial x_{t,d_x}}{\partial s_{t,d_x}})$, we can summarize the result as

$$\mathbf{J}_g(\mathbf{x}_t)\mathbf{J}_m(\mathbf{s}_t) = \mathbf{J}_m(\mathbf{s}_t) - \mathbf{D}_m(\mathbf{s}_t). \tag{A29}$$

### A.6 PROOF OF COROLLARY 2.1

Eq. (A29) states that

$$(\mathbf{I}_{d_x} - \mathbf{J}_g(\mathbf{x}_t))\mathbf{J}_m(\mathbf{s}_t) = \mathbf{D}_m(\mathbf{s}_t). \tag{A30}$$

From the DAG structure specified in Condition 1 and the functional faithfulness assumption in Assumption 2, the Jacobian matrix $\mathbf{J}_g(\mathbf{x}_t)$ can be permuted into a lower triangular form via identical row and column permutations. Thus, the matrix $\mathbf{I}_{d_x} - \mathbf{J}_g(\mathbf{x}_t)$ is invertible for all $\mathbf{x}_t \in \mathcal{X}_t$.
Since $\mathbf{D}_m(\mathbf{s}_t)$ is obtained via multiplication with $(\mathbf{I}_{d_x} - \mathbf{J}_g(\mathbf{x}_t))$, it follows that

$$(\mathbf{I}_{d_x} - \mathbf{J}_g(\mathbf{x}_t))^{-1}\mathbf{D}_m(\mathbf{s}_t) \tag{A31}$$

is well-defined and invertible. This, in turn, implies that $\mathbf{J}_m(\mathbf{s}_t)$ is invertible.
Furthermore, we establish that

$$\text{supp}\left(\mathbf{I}_{d_x} - \mathbf{J}_g(\mathbf{x}_t)\right) = \text{supp}\left(\mathbf{J}_g(\mathbf{x}_t, \mathbf{s}_t)\right) \tag{A32}$$

since the diagonal entries of $\mathbf{J}_g(\mathbf{x}_t, \mathbf{s}_t)$ are nonzero. Given that $\mathbf{J}_g(\mathbf{x}_t, \mathbf{s}_t)$ inherits the lower triangular structure after permutation, it must also be invertible.

### A.7 PROOF OF THEOREM 3

We present some useful definitions and lemmas in our proof.

**Definition 4.** *(Ordered Component-wise Identifiability)* *Variables* $\mathbf{s}_t \in \mathbb{R}^{d_x}$ *and* $\hat{\mathbf{s}}_t \in \mathbb{R}^{d_x}$ *are identified component-wise if* $\hat{s}_{t,i} = h_{s_i}(s_{t,\pi(i)})$ *with invertible function* $h_{s_i}$ *and* $\pi(i) = i$.

**Lemma 3** (Lemma 1 in LiNGAM (Shimizu et al., 2006)). *Assume* $\mathbf{M}$ *is lower triangular and all diagonal elements are non-zero. A permutation of rows and columns of* $\mathbf{M}$ *has only non-zero entries in the diagonal if and only if the row and column permutations are equal.*

**Lemma 4** (Proposition in Lin (1997)). *Suppose that* $\hat{s}_{t,i}$ *and* $\hat{s}_{t,j}$ *are conditionally independent given* $\hat{\mathbf{z}}_t$. *Then, for all* $\hat{\mathbf{z}}_t$,

$$\frac{\partial^2 \log p(\hat{\mathbf{s}}_t \mid \hat{\mathbf{z}}_t)}{\partial \hat{s}_{t,i} \partial \hat{s}_{t,j}} = 0.$$

*Proof.* Let $(\hat{\mathbf{z}}_t, \hat{\mathbf{s}}_t, \hat{g}_m)$ be the estimations of $(\mathbf{z}_t, \mathbf{s}_t, g_m)$. By Lemma 2,

$$\mathbf{x}_t = g_m(\mathbf{z}_t, \mathbf{s}_t); \quad \hat{\mathbf{x}}_t = \hat{g}_m(\hat{\mathbf{z}}_t, \hat{\mathbf{s}}_t) \tag{A33}$$

Suppose we reconstruct observations well: $\mathbf{x}_t = \hat{\mathbf{x}}_t$. Combined with Theorem 1,

$$p(\mathbf{x}_t \mid \hat{\mathbf{z}}_t) = p(\mathbf{x}_t \mid h_z(\mathbf{z}_t)) = p(\mathbf{x}_t \mid \mathbf{z}_t) \implies p(g_m(\mathbf{s}_t, \mathbf{z}_t) \mid \mathbf{z}_t) = p(\hat{g}_m(\hat{\mathbf{s}}_t, \hat{\mathbf{z}}_t) \mid \hat{\mathbf{z}}_t). \tag{A34}$$

Corollary 2.1 has shown that $\mathbf{J}_m(\mathbf{s}_t)$ and $\mathbf{J}_{\hat{g}_m}(\hat{\mathbf{s}}_t)$ are invertible matrices, by the definition of partial Jacobian matrix: $[\mathbf{J}_m(\mathbf{s}_t)]_{i,j} = \frac{\partial x_{t,i}}{\partial s_{t,j}} = \frac{\partial g_{m_i}(\mathbf{s}_t, \mathbf{z}_t)}{\partial s_{t,j}}$,

$$\frac{1}{|\mathbf{J}_m(\mathbf{s}_t)|} p(\mathbf{s}_t \mid \mathbf{z}_t) = \frac{1}{|\mathbf{J}_{\hat{g}_m}(\hat{\mathbf{s}}_t)|} p(\hat{\mathbf{s}}_t \mid \mathbf{z}_t). \tag{A35}$$

We define $h_s := m^{-1} \circ \hat{g}_m$ for any fixed $\mathbf{z}_t$ and $\hat{\mathbf{z}}_t$, hence, $|\mathbf{J}_{h_s}(\hat{\mathbf{s}}_t)| = \frac{|\mathbf{J}_{\hat{g}_m}(\hat{\mathbf{s}}_t)|}{|\mathbf{J}_m(\mathbf{s}_t)|}$ and $\hat{\mathbf{s}}_t = h_s(\mathbf{s}_t)$. Therefore, we have

$$p(\hat{\mathbf{s}}_t \mid \mathbf{z}_t) = \frac{1}{|\mathbf{J}_{h_s}(\hat{\mathbf{s}}_t)|} p(\mathbf{s}_t \mid \mathbf{z}_t) \implies \log p(\hat{\mathbf{s}}_t \mid \hat{\mathbf{z}}_t) = \log p(\mathbf{s}_t \mid \mathbf{z}_t) - \log |\mathbf{J}_{h_s}(\hat{\mathbf{s}}_t)|. \tag{A36}$$

The second-order partial derivative of $\log p(\hat{\mathbf{s}}_t \mid \hat{\mathbf{z}}_t)$ w.r.t. $(\hat{s}_{t,i}, \hat{s}_{t,j})$ is

$$\frac{\partial \log p(\hat{\mathbf{s}}_t \mid \hat{\mathbf{z}}_t)}{\partial \hat{s}_{t,i}} = \sum_{k=1}^{n} \frac{\partial \mathbf{A}_{t,k}}{\partial s_{t,k}} \cdot \frac{\partial s_{t,k}}{\partial \hat{s}_{t,i}} - \frac{\partial \log |\mathbf{J}_{h_s}(\hat{\mathbf{s}}_t)|}{\partial \hat{s}_{t,i}} = \sum_{k=1}^{n} \frac{\partial \mathbf{A}_{t,k}}{\partial s_{t,k}} \cdot [\mathbf{J}_{h_s}(\hat{\mathbf{s}}_t)]_{k,i} - \frac{\partial \log |\mathbf{J}_{h_s}(\hat{\mathbf{s}}_t)|}{\partial \hat{s}_{t,i}},$$

$$\frac{\partial^2 \log p(\hat{\mathbf{s}}_t \mid \hat{\mathbf{z}}_t)}{\partial \hat{s}_{t,i} \partial \hat{s}_{t,j}} = \sum_{k=1}^{n} \left( \frac{\partial^2 \mathbf{A}_{t,k}}{\partial s_{t,k}^2} \cdot [\mathbf{J}_{h_s}(\hat{\mathbf{s}}_t)]_{k,i} \cdot [\mathbf{J}_{h_s}(\hat{\mathbf{s}}_t)]_{k,j} + \frac{\partial \mathbf{A}_{t,k}}{\partial s_{t,k}} \cdot \frac{\partial [\mathbf{J}_{h_s}(\hat{\mathbf{s}}_t)]_{k,i}}{\partial \hat{s}_{t,j}} \right) - \frac{\partial^2 \log |\mathbf{J}_{h_s}(\hat{\mathbf{s}}_t)|}{\partial \hat{s}_{t,i} \partial \hat{s}_{t,j}}. \tag{A37}$$

Since for any $(i, j, t) \in \mathcal{J} \times \mathcal{J} \times \mathcal{T}$, we have $s_{t,i} \perp\!\!\!\perp s_{t,j} \mid \mathbf{z}_t$, Lemma 4 tells us $\frac{\partial^2 \log p(\hat{\mathbf{s}}_t \mid \hat{\mathbf{z}}_t)}{\partial \hat{s}_{t,i} \partial \hat{s}_{t,j}} = 0$. Therefore, its partial derivative w.r.t. $z_{t,l}$ $(l \in \mathcal{J})$ is always 0:

$$\frac{\partial^3 \log p(\hat{\mathbf{s}}_t \mid \hat{\mathbf{z}}_t)}{\partial \hat{s}_{t,i} \partial \hat{s}_{t,j} \partial z_{t,l}} = \sum_{k=1}^{n} \left( \frac{\partial^3 \mathbf{A}_{t,k}}{\partial s_{t,k}^2 \partial z_{t,l}} \cdot [\mathbf{J}_{h_s}(\hat{\mathbf{s}}_t)]_{k,i} \cdot [\mathbf{J}_{h_s}(\hat{\mathbf{s}}_t)]_{k,j} + \frac{\partial^2 \mathbf{A}_{t,k}}{\partial s_{t,k} \partial z_{t,l}} \cdot \frac{\partial [\mathbf{J}_{h_s}(\hat{\mathbf{s}}_t)]_{k,i}}{\partial \hat{s}_{t,j}} \right) \equiv 0, \tag{A38}$$

since entries of $\mathbf{J}_{h_s}(\hat{\mathbf{s}}_t)$ do not depend on $z_{t,l}$. By Assumption 3, maintaining this equality requires $[\mathbf{J}_{h_s}(\hat{\mathbf{s}}_t)]_{k,i} \cdot [\mathbf{J}_{h_s}(\hat{\mathbf{s}}_t)]_{k,j} = 0$ for $i \neq j$, which implies $\mathbf{J}_{h_s}(\hat{\mathbf{s}}_t)$ is a monomial matrix.

**Eliminate the Permutation Indeterminacy.** We leverage the following properties:

1. The inverse of a lower triangular matrix remains a lower triangular matrix.

2. A matrix representing a DAG can always be permuted into a lower-triangular form using appropriate row and column permutations.

3. Corollary 2.2 states that:

$$\mathbf{J}_{g^L}(\mathbf{x}_t) = \mathbf{I}_{d_x} - \mathbf{D}_{m^L}(\mathbf{s}_t)\mathbf{J}_{m^L}^{-1}(\mathbf{s}_t); \quad \mathbf{J}_g(\mathbf{x}_t) = \mathbf{I}_{d_x} - \mathbf{D}_m(\mathbf{s}_t)\mathbf{J}_m^{-1}(\mathbf{s}_t) \tag{A39}$$

where $\mathbf{J}_{g^L}(\mathbf{x}_t)$ and $\mathbf{J}_{m^L}(\mathbf{s}_t)$ are (strictly) lower triangular matrices obtained by permuting $\mathbf{J}_g(\mathbf{x}_t)$ and $\mathbf{J}_m(\mathbf{s}_t)$, respectively. $\mathbf{D}_{m^L}(\mathbf{s}_t)$ is the diagonal matrix extracted from $\mathbf{J}_{m^L}(\mathbf{s}_t)$. Consequently, we can express the relationship between $\mathbf{J}_m(\mathbf{s}_t)$ and $\mathbf{J}_{m^L}(\mathbf{s}_t)$ as follows:

$$\mathbf{J}_{g^L}(\mathbf{x}_t) = \mathbf{P}_{d_x}\mathbf{J}_g(\mathbf{x}_t)\mathbf{P}_{d_x}^\top \implies \mathbf{J}_m(\mathbf{s}_t) = \mathbf{P}_{d_x}\mathbf{J}_{m^L}(\mathbf{s}_t)\mathbf{D}_{m^L}^{-1}(\mathbf{s}_t)\mathbf{P}_{d_x}^\top \mathbf{D}_m(\mathbf{s}_t), \tag{A40}$$

where $\mathbf{P}_{d_x}$ is the Jacobian matrix of a permutation function on the $d_x$-dimensional vector. Consequently, by $\mathbf{J}_m(\mathbf{s}_t) = \mathbf{J}_{\hat{g}_m}(\hat{\mathbf{s}}_t)\mathbf{J}_{h_s}(\mathbf{s}_t)$, we obtain

$$\mathbf{J}_{\hat{g}_m}(\hat{\mathbf{s}}_t) = \mathbf{P}_{d_x}\mathbf{J}_{m^L}(\mathbf{s}_t)\mathbf{D}_{m^L}^{-1}(\mathbf{s}_t)\mathbf{P}_{d_x}^\top \mathbf{D}_m(\mathbf{s}_t)\mathbf{J}_{h_s}^{-1}(\mathbf{s}_t), \tag{A41}$$

Using Lemma 3, we obtain $\mathbf{P}_{d_x}\mathbf{D}_{m^L}^{-1}(\mathbf{s}_t)\mathbf{P}_{d_x}^\top \mathbf{D}_m(\mathbf{s}_t)\mathbf{J}_{h_s}(\hat{\mathbf{s}}_t) = \mathbf{I}_{d_x}$, which implies $\mathbf{J}_{h_s}^{-1}(\mathbf{s}_t) = \mathbf{D}_m^{-1}(\mathbf{s}_t)\mathbf{D}_{m^L}(\mathbf{s}_t)$, a diagonal matrix. Consequently, $\mathbf{J}_{\hat{g}_m}(\hat{\mathbf{s}}_t)$ and $\mathbf{J}_m(\mathbf{s}_t)$ have the same support, meaning $\mathbf{J}_{\hat{g}}(\hat{\mathbf{x}}_t)$ and $\mathbf{J}_g(\mathbf{x}_t)$ share the same support as well, according to Corollary 2.2. Thus, by Assumption 2, causal structures over observed variables are identifiable. □

**Discussion on Assumptions.** To enhance understanding of our theoretical results, we provide some explanations of the assumptions, their connections to real-world scenarios, as well as the potential boundaries of theoretical results.

i. **Generation Variability.** Sufficient changes on generation 3 is widely used in identifiable nonlinear ICA/causal representation learning (Hyvärinen et al., 2023; Lachapelle et al., 2022; Khemakhem et al., 2020; Zhang et al., 2024; Yao et al., 2022). In practical climate science, it has been demonstrated that, within a given region, human activities ($s_{t,i}$) are strongly impacted by certain high-level climate latent variables $\mathbf{z}_t$ (Abbass et al., 2022), following a process with sufficient changes (Lucarini et al., 2014).

ii. **Functional faithfulness.** Functional faithfulness corresponds to the *edge minimality* Zhang (2013); Lemeire & Janzing (2013); Peters et al. (2017) for the Jacobian matrix $\mathbf{J}_g(\mathbf{x}_t)$ representing the nonlinear SEM $\mathbf{x}_t = g(\mathbf{x}_t, \mathbf{z}_t, \boldsymbol{\epsilon}_{\mathbf{x}_t})$, where $\frac{\partial x_{t,j}}{\partial x_{t,i}} = 0$ implies no causal edge, and $\frac{\partial x_{t,j}}{\partial x_{t,i}} \neq 0$ indicates causal relation $x_{t,i} \to x_{t,j}$. This assumption is fundamental to ensuring that the Jacobian matrix reflects the true causal graph. If our functional faithfulness is violated, the results can be misleading, but in theory (classical) faithfulness Spirtes et al. (2001) is generally possible as discussed in Lemeire & Janzing (2013) (2.3 Minimality). As a weaker version of it, edge minimality holds the same property. If needed, violations of faithfulness can be testable except in the triangle faithfulness situation Zhang (2013). As opposed to classical faithfulness Spirtes et al. (2001), for example, this is not an assumption about the underlying world, but a convention to avoid redundant descriptions.

### A.8 PROOF OF LEMMA 1

(We delay the section of this proof since it relies on previous results.) The injectivity of a operator is formally characterized by the completeness of the conditional density function $p(a \mid b)$ used in the operator, as defined below.

**Definition 5** (Completeness). *A family of conditional density functions $p_{A|B}$ is said to be complete if the only solution to $\int_A p(a)p_{a|b}(a \mid b)\,da = 0,\ \forall b \in \mathcal{B}$ is $p(a) = 0$.*

Since the transformation from $\mathbf{s}_t$ to $\mathbf{x}_t$ is invertible and deterministic, given a $\dot{\mathbf{s}}_t \in \mathcal{S}_t$, the probability density function for $\mathbf{x}_t$ can be expressed as: $p(\mathbf{x}_t) = \begin{cases} \frac{1}{|\mathbf{J}_m(\mathbf{s}_t)|}p(\mathbf{s}_t), & \mathbf{x}_t = m(\mathbf{s}_t) \\ 0, & \mathbf{x}_t \neq m(\mathbf{s}_t) \end{cases}$. Hence, the conditional probability can be represented using the Dirac delta function:

$$p(\mathbf{x}_t \mid \mathbf{s}_t) = \delta(\mathbf{x}_t - m(\mathbf{s}_t)) \implies p(\mathbf{x}_t) = L_{\mathbf{x}_t|\mathbf{S}_t} \circ p(\mathbf{s}_t) = \int_{\mathcal{S}_t} \delta(\mathbf{x}_t - m(\mathbf{s}_t))p(\mathbf{s}_t)\,d\mathbf{s}_t.$$

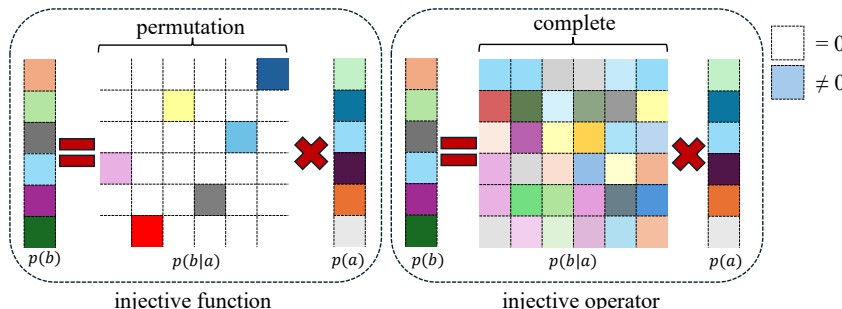

Figure A6: **Invertible Function v.s. Injective Operator.** *(Left)* Consider two variables $a$ and $b$ connected by the function $b = g(a)$, where $g$ is invertible. *(Right)* Alternatively, their relationship can be expressed as $p(b) = L_{b|a} \circ p(a)$, where $L_{b|a}$ is an injective operator. The grid represents $p(b \mid a)$, with color indicating non-zero values and white representing zero. Intuitively, in the discrete case, a full-rank matrix corresponds to this relationship.

By recalling Eq. (2), we can rewrite $p(\mathbf{x}_t)$ in terms of the operator $L_{\mathbf{x}_t|\mathbf{s}_t}$ acting on $p_{\mathbf{s}_t}$. We consider $p(\mathbf{x}_t \mid \mathbf{s}_t)$ as an infinite-dimensional vector, and the operator $L_{\mathbf{x}_t|\mathbf{s}_t}$ as an infinite-dimensional matrix where

$$L_{\mathbf{x}_t|\mathbf{s}_t} = [\delta(\mathbf{x}_t - m(\mathbf{s}_t))]^{\top}_{\mathbf{x}_t \in \mathcal{X}_t} .$$

By Corollary 2.2, since $\mathbf{J}_m(\mathbf{s}_t)$ is invertible, for any two different points $\mathbf{s}_t^{(1)}, \mathbf{s}_t^{(2)} \in \mathcal{S}_t$ ($\mathbf{s}_t^{(1)} \neq \mathbf{s}_t^{(2)}$), we have $m(\mathbf{s}_t^{(1)}) \neq m(\mathbf{s}_t^{(2)})$. This implies that the supports of $\delta(\mathbf{x}_t - m(\mathbf{s}_t^{(1)}))$ and $\delta(\mathbf{x}_t - m(\mathbf{s}_t^{(2)}))$ are disjoint. Thus, $[\delta(\mathbf{x}_t - m(\mathbf{s}_t))]^{\top}_{\mathbf{x}_t \in \mathcal{X}_t}$ preserves a one-to-one correspondence across the $\mathcal{X}_t$, ensuring:

$$\text{null } [\delta(\mathbf{x}_t - m(\mathbf{s}_t))]^{\top}_{\mathbf{x}_t \in \mathcal{X}_t} = \{0^{(\infty)}\},$$

which denotes the completeness of $L_{\mathbf{x}_t|\mathbf{s}_t}$ stated in Definition 5, indicating that $L_{\mathbf{x}_t|\mathbf{s}_t}$ is injective. The visualization in Figure A6 highlights why Assumption 1 is significantly less restrictive than the invertibility assumption adopted in most of the previous CRL literature Hyvarinen & Morioka (2016; 2017); Hyvarinen et al. (2019); Klindt et al. (2020); Zhang et al. (2024); Li et al. (2025) in a zero-measure manner.

### A.9 COMPARISON WITH EXISTING METHODS

Our method targets the joint identification of latent causal graphs and causal structures over observed variables in time series. This is essential for domains such as climate science, where latent processes govern observed dynamics. In contrast, IDOL Li et al. (2025) focuses solely on recovering latent variables and assumes a deterministic mixing function without any causal relations among observed variables. As a result, it cannot recover the causal graph on observations and fails in contexts like climate systems, where both latent and observational structures are crucial.

Prior works Monti et al. (2020); Reizinger et al. (2023) use nonlinear ICA to recover causal relations among observed variables, assuming known domain variables and non-i.i.d. data. However, (1) CaDRe does not require predefined domain variables and instead leverages contextual information to infer latent variables as conditional priors; (2) ICA-based methods are not robust under latent confounding, as spurious correlations may obscure true causal links; (3) they do not identify the latent variables or their underlying dynamics; (4) they require invertibility of $g$ and $m$, an assumption we relax in Corollary 2.1.

The FCI algorithm Spirtes & Glymour (1991) allows for latent confounding and uses conditional independence tests to infer causal relations among observed variables. However, it cannot recover the latent variables themselves or their causal influence on observations. Furthermore, the causal structure is expressed as a Partial Ancestral Graph (PAG), which represents an equivalence class and may contain ambiguous or uncertain edges. Such ambiguity is particularly problematic for applications like climate analysis, which demand interpretable and stable causal structures.

## B RELATED WORK

### B.1 CLIMATE ANALYSIS

Climate analysis is learning to address the complex, nonlinear, and high-dimensional nature of Earth system dynamics. A prominent line of work focuses on using neural networks for weather and climate forecasting, including data-driven models such as FourCastNet (Pathak & et al., 2022) and GraphCast (Lam & et al., 2022), which demonstrate remarkable predictive performance by modeling spatiotemporal dependencies. However, these methods often lack interpretability and fail to reveal the underlying causal mechanisms driving climate variability. To address this issue, recent research has integrated causal discovery into climate science. For instance, Runge et al. (2019a) introduces causal inference frameworks tailored to climate time series, incorporating techniques such as PCMCI to infer lagged and contemporaneous dependencies. Other approaches employ structural causal models (SCMs) for identifying interactions between climate variables under interventions (Reichstein & et al., 2019). Beyond shallow models, efforts have emerged to disentangle latent variables in high-dimensional climate data using variational autoencoders (Klushyn & et al., 2021). While effective, most of these methods do not guarantee identifiability or robust generalization across regimes.

More recently, hybrid models that couple dynamical systems theory with deep learning have shown promise in capturing climate processes with greater fidelity. Examples include integrating physics-based constraints into latent state-space models (Beucler & et al., 2021) and learning interpretable representations for climate variability modes such as the Madden-Julian Oscillation (Toms & Barnes, 2020). These works highlight the growing interest in combining structure learning, causal inference, and deep latent modeling to move beyond black-box predictions towards actionable scientific understanding.

### B.2 CAUSAL REPRESENTATION LEARNING

Achieving causal representations for time series data (Rajendran et al., 2024) often relies on nonlinear ICA to recover latent variables with identifiability guarantees (Yao et al., 2023; Schölkopf et al., 2021). Classical ICA methods assume a linear mixing function between latent and observed variables (Comon, 1994). To move beyond this linearity assumption, recent advances in nonlinear ICA have established identifiability under various alternative assumptions, including the use of auxiliary variables or structural sparsity (Zheng et al., 2022; Hyvärinen & Pajunen, 1999; Hyvärinen et al., 2023). One prominent line of work introduces auxiliary variables to facilitate identifiability. For instance, (Khemakhem et al., 2020) achieves identifiability by assuming latent sources follow an exponential family distribution and incorporating side information such as domain, time, or class labels (Hyvarinen & Morioka, 2016; 2017; Hyvarinen et al., 2019). To relax the exponential family requirement, (Kong et al., 2023) establishes component-wise identifiability using $2n + 1$ auxiliary variables for $n$ latent components.

Another direction pursues identifiability in a fully unsupervised setting by leveraging structural sparsity. (Lachapelle et al., 2022) propose a sparsity-based inductive bias to disentangle latent causal factors, demonstrating identifiability in multi-task learning and related settings. They further extend these results to establish identifiability up to a consistency class (Lachapelle et al., 2024), allowing partial disentanglement. Complementarily, (Zheng et al., 2022) and (Zhang et al., 2024; Li et al., 2025) exploit sparse latent structures under distributional shifts to obtain identifiability results without relying on auxiliary information.

### B.3 CAUSAL DISCOVERY

Existing causal discovery methods in climate analysis primarily build on extensions of PCMCI (Runge et al., 2019b), which effectively captures time-lagged and instantaneous linear dependencies, and its nonlinear variant (Runge, 2020). However, both approaches assume fully observed systems and neglect latent variables, limiting their applicability to complex climate dynamics. Recent causal representation learning methods motivated by climate science attempt to address this gap: (Brouillard et al., 2024) imposes strong identifiability assumptions via single-node structures, while (Yao et al., 2024) adopts an ODE-based model to study climate-zone classification, though these methods often overlook dependencies among observed variables. Beyond climate, a class of nonlinear causal discovery methods leverages Jacobian information for identifiability and acyclicity (Lachapelle et al., 2019; Rolland et al., 2022), including applications to structural equation models (Atanackovic et al., 2024), Markov structures (Zheng et al., 2023), independent mechanisms (Gresele et al., 2021), and non-i.i.d. settings (Reizinger et al., 2023). While (Dong et al., 2023) propose a general framework that accounts for hidden variables by using rank conditions on the observed covariance matrix, their

model is restricted to linear relationships and cannot recover nonlinear latent dynamics in time-series data. In contrast, our method, CaDRe, recovers latent causal structures under nonlinear dependencies, though it currently does not support cases where observed variables act as causes of latent ones—a limitation we leave to future work. Considering the nonlinear CD based on continuous optimization, we additionally provide the Table A7 for comparison.

### B.4 TIME-SERIES FORECASTING

Time series forecasting has seen rapid progress with deep learning methods that leverage various neural architectures. RNN-based models (Hochreiter & Schmidhuber, 1997; Lai et al., 2018; Salinas et al., 2020) focus on sequential dependencies, while CNN-based approaches (Bai et al., 2018; Wang et al., 2022; Wu et al., 2022) capture local temporal patterns. State-space models (Gu et al., 2021b;a; 2022) offer structured modeling of latent dynamics. Transformer-based methods (Zhou et al., 2021; Wu et al., 2021; Nie et al., 2022) further advance long-range forecasting through attention mechanisms. However, most existing methods neglect instantaneous dependencies among variables, limiting their ability to fully capture the joint dynamics of multivariate time series.

Table A7: Comparison of different methods based on their properties in function type ($f$), data, Jacobian ($J$), capability of performing CD and CRL, and whether they achieve identifiability.

| Method | $f$ | Data | $J$ | CD | CRL | Identifiability |
|---|---|---|---|---|---|---|
| LiNGAM (Shimizu et al., 2006) | Linear | Non-Gaussian | $J_{f^{-1}}$ | ✔ | ✗ | ✔ |
| GraN-DAG (Lachapelle et al., 2019) | Additive | Gaussian | $J_{f^{-1}}$ | ✔ | ✗ | ✗ |
| IMA (Gresele et al., 2021) | IMA | All | $J_f$ | ✗ | ✗ | ✔ |
| G-SCM (Zheng et al., 2023) | Sparse | All | $J_f$ | ✗ | ✗ | ✔ |
| Score-Based FCMs (Rolland et al., 2022) | Additive | Gaussian | $J_{\nabla_x \log p(x)}$ | ✔ | ✗ | ✗ |
| DynGFN (Atanackovic et al., 2024) | Cyclic (ODE) | All | $J_f$ | ✔ | ✗ | ✗ |
| JacobianCD (Reizinger et al., 2023) | All | Assums. 2, F. 1 | $J_{f^{-1}}$ | ✔ | ✗ | ✔ |
| CausalScore (Liu et al., 2024b) | Mixed | Gaussian | $J_{\nabla_x \log p(x)}$ | ✔ | ✗ | Partial |
| CaDRe (Ours) | All | All | $J_{f^{-1}}$ | ✔ | ✔ | ✔ |

## C EXPERIMENT DETAILS

This section documents the experimental protocol used to assess CaDRe on both simulated and real-world data. We begin by specifying the data-generating processes, evaluation metrics, and baseline implementations so that results can be reproduced with the same assumptions. We then describe the initialization of the observational causal graph, masking strategies informed by spatial priors, and the computation of Jacobians used for graph extraction. Finally, we summarize model architectures and training settings, followed by extended quantitative results and ablations.

### C.1 ON SIMULATION DATASET

We validate identifiability under controlled settings by varying dimension, sparsity, and temporal order.

#### C.1.1 SIMULATION DETAILS

**Data Simulation.** We generate time series data with latent variables $\mathbf{z}_t \in \mathbb{R}^{d_z}$ and observed variables $\mathbf{x}_t \in \mathbb{R}^{d_x}$, where $d_z \leq d_x$. The latent dynamics follow a leaky non-linear autoregressive model:

$$\mathbf{z}_t = \sigma\left(\sum_{\ell=1}^{L} \mathbf{W}^{(\ell)} \mathbf{z}_{t-\ell}\right) + \boldsymbol{\epsilon}_t^z, \quad \boldsymbol{\epsilon}_t^z \sim \mathcal{N}(0, \sigma_z^2 \mathbf{I}), \tag{A42}$$

where $\sigma(\cdot)$ is leaky ReLU, and $\mathbf{W}^{(\ell)}$ are lag-$\ell$ transition matrices modulated by class-specific parameters. Instantaneous causal relations among $\mathbf{x}_t$ are defined by an Erdős-Rényi DAG $\mathbf{B} \in \{0, 1\}^{d_x \times d_x}$, with time-varying edge weights:

$$\mathbf{B}_t = \alpha(t) \cdot \mathbf{B}, \quad \alpha(t) = a_1 \cos\left(\frac{2\pi t}{T}\right) + a_2. \tag{A43}$$

The observed variable $\mathbf{x}_t$ is first generated by a multilayer mixing of $(\mathbf{z}_t, \mathbf{s}_t)$, followed by additive and autoregressive noise:

$$\mathbf{x}_t = f_{\text{mix}}(\mathbf{z}_t, \mathbf{s}_t) + \mathbf{s}_t, \quad \mathbf{s}_t = \boldsymbol{\epsilon}_t^x + f_{\text{dep}}(xx_{t-1}), \tag{A44}$$

where $\epsilon_t^x \sim \mathcal{U}[0, \sigma_x]$. Then, causal effects among observed variables are injected based on $\mathbf{B}_t$ in topological order:

$$x_{t,i} \leftarrow x_{t,i} + \sum_{j \in \mathrm{pa}(i)} B_{t,j,i} \cdot x_{t,j}, \tag{A45}$$

where $\mathbf{pa}(i) = \{j \mid B_{t,j,i} \neq 0\}$ are the causal parents of variable $i$ under $\mathbf{B}_t$.

**Various Datasets.** We generate simulated time-series data using the fixed latent causal process described in Eq. (1) and illustrated in Figure 1. To comprehensively evaluate our theoretical results, we construct synthetic datasets with varying observed dimensionalities, including $d_x = 3, 6, 8, 10, 100^{*1}$ and latent dimensionalities $d_z = 2, 3, 4$, specified for each experiment. Additionally, we simulate different levels of structural sparsity in the latent process under three regimes: *Independent*, *Sparse*, and *Dense*. For evaluation, we use SHD, TPR, Precision, and Recall for causal structure recovery, and MCC and $R^2$ for assessing latent representation identifiability. As defined in Eq. (1), under the *Independent* setting for the latent temporal process and dependent noise variable $\mathbf{s}_t$, we use the generation process from (Yao et al., 2022), meaning there are no instantaneous dependencies within the $\mathbf{z}_t$. For *Sparse* and *Dense* settings, we gradually increase the graph degree after removing diagonals. Each independent noise is sampled from normal distributions.

**Implementation Details of CRL Baselines.** We employed publicly available implementations for TDRL, CaRiNG, and iCRITIS, which cover most of the baselines used in our experiments. For G-CaRL, whose official code was not released, we re-implemented the method based on the descriptions in the original paper. Furthermore, because the original iCRITIS framework was tailored for image-based inputs, we adapted it to our setting by replacing its encoder and decoder with a VAE architecture, using the same hyperparameters as in CaDRe.

**Mask by Inductive Bias.** Continuous optimization faces challenges like local minima (Ng et al., 2022; Maddison et al., 2017), making it difficult to scale to higher dimensions. However, incorporating prior knowledge on the low probability of certain dependencies (Spirtes et al., 2001; Runge et al., 2019b) enables us to compute a mask. To validate this approach using physical laws as observed DAG initialization C.2.2 in climate data, we mask 75% of the lower triangular elements in a simulation with $d_x = 100$, a ratio much lower than in real-world applications.

**Comparison with Constraint-Based Methods.** We compare our method against a series of constraint-based causal discovery algorithms, which rely on Conditional Independence (CI) tests without assuming a specific form for the SEMs. These approaches are nonparametric and model-agnostic, but they typically return equivalence classes of graphs rather than fully identifiable structures. For instance, FCI outputs Partial Ancestral Graphs (PAGs), while CD-NOD returns equivalence classes reflecting causal ambiguity under the observed CI constraints. For a fair comparison, we adopt near-optimal configurations of the most representative constraint-based methods. Specifically, we use the `Causal-learn` package (Zheng et al., 2024) to implement FCI and CD-NOD, and the `Tigramite` library (Runge et al., 2019b) for PCMCI and LPCMCI. Each method is run under recommended hyperparameter settings as reported in their respective documentation or prior studies, ensuring a reliable and balanced comparison.

i. **FCI**: We use Fisher's Z conditional independence test. For the obtained PAG, we enumerate all possible adjacency matrices and select the one closest to the ground truth by minimizing the SHD.

ii. **CD-NOD**: We concatenate the time indices $[1, 2, \ldots, T]$ of the simulated data into the observed variables and only consider the edges that exclude the time index. We use kernel-based CI test since it demonstrates superior performance here. We consider all obtained equivalence classes and select the result that minimizes SHD relative to the ground truth.

iii. **PCMCI**: We use partial correlation as the metric of the conditional independence test. We enforce no time-lagged relationships in PCMCI and run it to focus exclusively on contemporaneous (instantaneous) causal relationships. In the `Tigramite` library, this can be achieved by setting the maximum time lag $\tau_{\max}$ to zero. This effectively disables the search for lagged causal dependencies. We select contemporary relationships as the ultimate result.

iv. **LPCMCI**: Similarly to PCMCI, we use partial correlation as the metric of CI test, and select the contemporary relationships as the obtained causal graph.

---

[1]* indicates the use of a masking scheme simulated from geographical information (see Appendix C.1.1)

### C.1.2 EVALUATION METRICS.

We evaluate the recovery of latent variables and causal structures using the following metrics:

i. **Latent Space Recovery.** Following the identifiability result in Theorem 1, we measure the alignment between the estimated latent variables $\hat{\mathbf{z}}_t$ and the true latent variables $\mathbf{z}_t$ using the coefficient of determination $R^2$, where $R^2 = 1$ indicates perfect alignment. A nonlinear mapping is estimated using kernel regression with a Gaussian kernel.

ii. **Latent Component Recovery.** To evaluate component-wise identifiability as discussed in Theorem A1, we use the Spearman Mean Correlation Coefficient (MCC), which assesses the monotonic relationship between estimated and true latent components.

iii. **Latent Causal Structure.** For evaluating the recovery of latent causal graphs, both instantaneous and time-lagged, we compute the Structural Hamming Distance (SHD) between the learned and true adjacency matrices. Given the permutation indeterminacy of latent variables, we align the estimated latent causal structures $\mathbf{J}_r(\hat{\mathbf{z}}_t)$ and $\mathbf{J}_r(\hat{\mathbf{z}}_{t-1})$ with the ground truth by applying consistent permutations.

iv. **Observational Source Recovery.** As a surrogate for evaluating the causal graph over observed variables, we use MCC (Khemakhem et al., 2020) to assess the recovery of $\mathbf{s}_t$. Unlike latent variables, this metric does not allow permutations and reflects the identifiability condition stated in Theorem 3.

v. **Causal Structure Accuracy.** The recovered latent and causal DAGs over observed variables are also evaluated using SHD, normalized by the total number of possible edges to facilitate comparison across different graph sizes.

vi. **Graph-Level Metrics.** In addition to SHD, we report true positive rate (TPR), precision, and F1 score to benchmark our method against constraint-based approaches in causal graph recovery.

vii. **Wind-Induced Causal Graph (a Surrogate of Ground-Truth).** Since real-world climate datasets lack ground-truth causal structures among observations, we define a wind-induced causal graph derived from physical wind directions as a surrogate. For each grid point $i$, the dataset includes position, wind vector, and displaced position, from which the adjacency matrix $B^{\mathrm{ref}}$ is constructed.

viii. **Causal Discovery Metrics: WSHD and WTPR.** We evaluate the estimated causal graph $B$ against $B^{\mathrm{ref}}$ using two metrics:

- **WSHD:** Structural Hamming Distance between $B$ and $B^{\mathrm{ref}}$, $\mathrm{WSHD}(B, B^{\mathrm{ref}}) = \sum_{i \neq j} \mathbf{1}(B_{i,j} \neq B_{i,j}^{\mathrm{ref}})$.
- **WTPR:** Recall of wind-consistent causal edges, $\mathrm{WTPR}(B, B^{\mathrm{ref}}) = \frac{\sum_{i \neq j} \mathbf{1}(B_{i,j}=1 \land B_{i,j}^{\mathrm{ref}}=1)}{\sum_{i \neq j} \mathbf{1}(B_{i,j}^{\mathrm{ref}}=1)}$. The numerator counts correctly predicted causal edges, while the denominator counts edges inferred from wind direction.

ix. **Metric for Latent Causal Representation.** To assess the alignment between estimated latent variables $\hat{z}_t$ and known climate variables $v_t$ (e.g., precipitation, ocean currents), we use the Subset Mean Correlation Coefficient (SMCC): $\mathrm{SMCC} = \max_{\hat{v}_t \subset \hat{z}_t} \frac{1}{d-1} \sum_{j=1}^{d-1} \rho(v_t, \hat{v}_t^{(j)})$, where $\rho(v_t, \hat{v}_t^{(j)})$ is the Pearson correlation coefficient. This measures how well a subset of the latent space captures known physical signals.

### C.2 ON REAL-WORLD DATASETS

We test CaDRe on climate and weather benchmarks where ground-truth graphs are unavailable.

### C.2.1 DATASET DESCRIPTION

We include all the datasets used in our experiments below:

1. Weather [2] dataset offers 10-minute summaries from an automated rooftop station at the Max Planck Institute for Biogeochemistry in Jena, Germany.

2. CESM2 Pacific SST dataset employs monthly Sea Surface Temperature (SST) data generated from a 500-year pre-2020 control run of the CESM2 climate model. The dataset is restricted to oceanic regions, excluding all land areas, and retains its native gridded structure to preserve

---

[2]https://www.bgc-jena.mpg.de/wetter/

spatial correlations. It encompasses 6000 temporal steps, representing monthly SST values over the designated period. Spatially, the dataset comprises a grid with 186 latitude points and 151 longitude points, resulting in 28086 spatial variables, including 3337 land points where SST is undefined, and 24749 valid SST observations. To accommodate computational constraints, a downsampled version of the data, reduced to 84 grid points ($6 \times 14$), is utilized in our experiment.

3. WeatherBench (Rasp et al., 2020) is a benchmark dataset specifically tailored for data-driven weather forecasting. We specifically selected wind direction data for visualization comparisons within the same period, maintaining the original 350,640 timestamps.

4. ERSST [3] dataset is from NOAA GlobalTemp (NOAA/NCEI) official website, we use the NOAA Global Temperature Anomaly Dataset (1880–2025), which includes 2052 monthly steps and 16,020 spatial grid points per step. For time-series forecasting, we use a downscaled version with 100 dimensions, obtained by averaging over block regions.

### C.2.2 Implementation Details

**Initialization of Causal Graph over Observed Variables.** To improve the stability of continuous optimization and avoid poor local minima in estimating the causal structure matrix $\hat{\mathcal{G}}_{\mathbf{x}_t}$, we incorporate a prior based on the Spatial Autoregressive (SAR) model. The SAR model, widely applied in geography, economics, and environmental science, captures spatial dependencies through the formulation:

$$\mathbf{X} = \mathbf{Z}\beta + \lambda\mathbf{W}\mathbf{X} + \mathbf{E},$$

where $\mathbf{W}$ is the spatial weights matrix, $\beta$ is a regression coefficient, and $\mathbf{E}$ is a noise term. To simplify the model and isolate the spatial interaction component, we set $\beta = 0$ and $\lambda = 1$, resulting in the canonical SAR model:

$$\mathbf{X} = \mathbf{W}\mathbf{X} + \mathbf{E}.$$

The core assumption is that instantaneous interactions between regions are unlikely if they are separated by a substantial spatial distance. Therefore, we initialize $\mathbf{W}$ using a binary spatial adjacency matrix $\mathcal{M}_{\text{loc}}$, defined as

$$[\mathcal{M}_{\text{loc}}]_{i,j} = \mathbb{I}\left(\|s_i - s_j\|_2 \leq 50\right),$$

where $s_i$ and $s_j$ denote the spatial coordinates of regions $i$ and $j$, respectively. This constraint enforces that only regions within a Euclidean distance of 50 units are considered spatially adjacent. We then estimate $\lambda$ and update $\mathbf{W}$ by fitting the linear model $\mathbf{X} = \mathbf{W}\mathbf{X} + \mathbf{E}$ via least squares. The resulting matrix $\mathbf{W}$ is used as the initialization for the causal graph over observed variables, denoted $\mathcal{M}_{\text{init}}$.

**Compute the Causal Graph over Observed Variables.** Using a mask gradient-based approach, we compute an initial estimate of the Jacobian $\mathbf{J}_{\hat{g}}(\mathbf{x}_t)$, which encodes local sensitivities. However, these Jacobian matrices are typically dense and difficult to interpret directly. To produce a more interpretable visualization of the causal graph over observed variables, we apply a masking operation followed by elementwise thresholding:

$$\hat{\mathcal{G}}_{x_t} = \mathbb{I}\left(|\mathbf{J}_{\hat{g}}(\mathbf{x}_t) \odot \mathcal{M}_{\text{init}}| > \tau\right), \tag{A46}$$

where $\odot$ denotes the elementwise (Hadamard) product, and $\mathbb{I}(\cdot)$ is the indicator function that outputs 1 if the condition is true and 0 otherwise. We set the threshold to $\tau = 0.15$ to obtain a binary adjacency matrix. $\mathcal{M}_{\text{init}}$ is the initialization mask.
To compute the partial Jacobian $\mathbf{J}_{\hat{g}}(\mathbf{x}_t)$ with respect to $\mathbf{s}_t$ while keeping $\mathbf{z}_t$ fixed, we disable gradient tracking for $\mathbf{z}_t$ by setting `requires_grad=False`, and use `autograd.functional.jacobian` in PyTorch.

**Model Structure** We choose MICN (Wang et al., 2022) as the encoder backbone of our model on real-world datasets. Specifically, given that the MICN extracts the hidden feature, we apply a variational inference block and then an MLP-based decoder. Architecture details of the proposed method are shown in Table A8.

## D    Extended Experiment Results

We report scalability studies, ablations, higher-order dynamics, and additional forecasting comparisons.

---

[3]https://www.ncei.noaa.gov/products/extended-reconstructed-sst

Table A8: Architecture details. $T$, length of time series. $|\mathbf{x}_t|$: input dimension. $n$: latent dimension. LeakyReLU: Leaky Rectified Linear Unit. Tanh: Hyperbolic tangent function.

| Configuration | Description | Output |
|---|---|---|
| $\phi$ | `z-encoder` | |
| Input:$\mathbf{x}_{1:t}$ | Observed time series | Batch Size$\times T \times d_x$ |
| Dense | $d_x$ neurons | Batch Size$\times T \times d_x$ |
| Concat zero | concatenation | Batch Size$\times T \times d_x$ |
| Dense | $d_z$ neurons | Batch Size$\times T \times d_z$ |
| $\eta$ | `s-encoder` | |
| Input:$\mathbf{x}_{1:t}$ | Observed time series | Batch Size$\times T \times d_x$ |
| Dense | $d_x$ neurons | Batch Size$\times T \times d_x$ |
| Concat zero | concatenation | Batch Size$\times T \times d_x$ |
| Dense | $d_x$ neurons | Batch Size$\times T \times d_x$ |
| $\psi$ | `decoder` | |
| Input:$\mathbf{z}_{1:T}$ | Latent Variable | Batch Size$\times T \times (d_z + d_x)$ |
| Dense | $d_x$ neurons, Tanh | Batch Size$\times T \times d_x$ |
| $r$ | Modular Prior Networks | |
| Input: $\mathbf{z}_{1:T}$ | Latent Variable | Batch Size$\times (d_z + 1)$ |
| Dense | 128 neurons,LeakyReLU | $(d_z + 1)\times 128$ |
| Dense | 128 neurons,LeakyReLU | $128\times 128$ |
| Dense | 128 neurons,LeakyReLU | $128\times 128$ |
| Dense | 1 neuron | Batch Size$\times 1$ |
| Jacobian Compute | Compute $\log(\det(J))$ | Batch Size |

## D.1 EXPERIMENT RESULTS OF SIMULATED DATASETS

We examine how $d_x$, $d_z$, sparsity, and order affect latent recovery and structure learning.

**Study on Dimension of Observed Variables.** Table A9 reports performance on CD and CRL, evaluated across $d_x \in \{3, 5, 8, 10, 20^*, 50^*, 80^*, 100^*, 200^*\}$. The results on MCC confirm the effectiveness of our methodology under identifiability conditions. The asterisk on $d_x = 100^*$ means that we impose a physical-distance prior to eliminate 75% of the edges from the fully connected graph. The corresponding results demonstrate that our approach scales effectively to high-dimensional settings, in presence of the imposed physical prior on the learned causal graph over climate grids.

Table A9: **Results under Varying Observational Dimensionality** ($d_x$). Each setting is repeated with 5 random seeds. For evaluation, the best-converged result per seed is selected to avoid local minima.

| $d_z$ | $d_x$ | SHD ($\mathbf{J}_{\hat{g}}(\hat{\mathbf{x}}_t)$)$\downarrow$ | TPR$\uparrow$ | Precision$\uparrow$ | MCC($\mathbf{s}_t$)$\uparrow$ | MCC($\mathbf{z}_t$)$\uparrow$ | SHD ($\mathbf{J}_r(\hat{\mathbf{z}}_t)$)$\downarrow$ | SHD ($\mathbf{J}_d(\hat{\mathbf{z}}_{t-1})$)$\downarrow$ | $R^2\uparrow$ |
|---|---|---|---|---|---|---|---|---|---|
| | 3 | $0.00_{\pm 0.00}$ | $1.00_{\pm 0.00}$ | $1.00_{\pm 0.00}$ | $0.9775_{\pm 0.01}$ | $0.9721_{\pm 0.01}$ | $0.27_{\pm 0.05}$ | $0.26_{\pm 0.03}$ | $0.90_{\pm 0.05}$ |
| | 6 | $0.18_{\pm 0.06}$ | $0.83_{\pm 0.03}$ | $0.80_{\pm 0.04}$ | $0.9583_{\pm 0.02}$ | $0.9505_{\pm 0.01}$ | $0.24_{\pm 0.06}$ | $0.33_{\pm 0.09}$ | $0.92_{\pm 0.01}$ |
| | 8 | $0.29_{\pm 0.05}$ | $0.78_{\pm 0.05}$ | $0.76_{\pm 0.04}$ | $0.9020_{\pm 0.03}$ | $0.9601_{\pm 0.03}$ | $0.36_{\pm 0.11}$ | $0.31_{\pm 0.12}$ | $0.93_{\pm 0.02}$ |
| | 10 | $0.43_{\pm 0.05}$ | $0.65_{\pm 0.08}$ | $0.63_{\pm 0.14}$ | $0.8504_{\pm 0.07}$ | $0.9652_{\pm 0.02}$ | $0.29_{\pm 0.04}$ | $0.40_{\pm 0.05}$ | $0.92_{\pm 0.02}$ |
| 3 | $20^*$ | $0.09_{\pm 0.01}$ | $0.92_{\pm 0.02}$ | $0.89_{\pm 0.01}$ | $0.9573_{\pm 0.12}$ | $0.9742_{\pm 0.08}$ | $0.10_{\pm 0.01}$ | $0.18_{\pm 0.04}$ | $0.96_{\pm 0.01}$ |
| | $50^*$ | $0.13_{\pm 0.02}$ | $0.87_{\pm 0.17}$ | $0.85_{\pm 0.19}$ | $0.9318_{\pm 0.01}$ | $0.9619_{\pm 0.01}$ | $0.16_{\pm 0.02}$ | $0.22_{\pm 0.06}$ | $0.93_{\pm 0.02}$ |
| | $80^*$ | $0.15_{\pm 0.02}$ | $0.84_{\pm 0.08}$ | $0.83_{\pm 0.10}$ | $0.9223_{\pm 0.07}$ | $0.9550_{\pm 0.09}$ | $0.18_{\pm 0.02}$ | $0.25_{\pm 0.13}$ | $0.94_{\pm 0.03}$ |
| | $100^*$ | $0.17_{\pm 0.02}$ | $0.80_{\pm 0.05}$ | $0.81_{\pm 0.02}$ | $0.9131_{\pm 0.02}$ | $0.9565_{\pm 0.02}$ | $0.21_{\pm 0.01}$ | $0.29_{\pm 0.10}$ | $0.93_{\pm 0.03}$ |
| | $200^*$ | $0.16_{\pm 0.07}$ | $0.74_{\pm 0.06}$ | $0.72_{\pm 0.04}$ | $0.8950_{\pm 0.02}$ | $0.9603_{\pm 0.03}$ | $0.22_{\pm 0.02}$ | $0.35_{\pm 0.12}$ | $0.92_{\pm 0.04}$ |

**Latent Dimensionality Sensitivity.** We have conducted additional experiments on $d_z \in \{3, 5, 7, 9, 12\}$. The results are summarized in Table A10, showing that CaDRe maintains strong performance across all metrics. When $d_z$ becomes very high (*e.g.*, 12), recovery of latent causal structures [SHD($J_r(\hat{\mathbf{z}}_t)$), SHD($J_d(\hat{\mathbf{z}}_{t-1})$)] and latent variables [MCC($\mathbf{z}_t$)] slightly declines, indicating the model's limitation in high-dimensional latent spaces. Inference time remains efficient

across scales since the nonlinear ICA-based structure learning in CaDRe is equivalent to a one-step generation process.

Table A10: Performance under varying latent dimensionality $d_z$.

| $d_z$ | $d_x$ | SHD($J_{\hat{g}}(\hat{\mathbf{x}}_t)$) | TPR | Precision | MCC($\mathbf{s}_t$) | MCC($\mathbf{z}_t$) | SHD($J_r(\hat{\mathbf{z}}_t)$) | SHD($J_d(\hat{\mathbf{z}}_{t-1})$) | $R^2$ | Time (ms) |
|---|---|---|---|---|---|---|---|---|---|---|
| 3 | 10 | 0.43±0.05 | 0.65±0.08 | 0.63±0.14 | 0.85±0.07 | 0.97±0.02 | 0.29±0.04 | 0.40±0.05 | 0.92±0.02 | 0.71±0.03 |
| 5 | 10 | 0.44±0.04 | 0.63±0.09 | 0.61±0.12 | 0.82±0.06 | 0.96±0.03 | 0.27±0.05 | 0.37±0.02 | 0.90±0.02 | 0.88±0.10 |
| 7 | 10 | 0.46±0.06 | 0.60±0.02 | 0.59±0.01 | 0.89±0.05 | 0.96±0.04 | 0.27±0.03 | 0.30±0.11 | 0.96±0.03 | 1.05±0.20 |
| 9 | 10 | 0.44±0.05 | 0.67±0.06 | 0.57±0.11 | 0.81±0.04 | 0.85±0.05 | 0.32±0.04 | 0.41±0.08 | 0.88±0.03 | 1.27±0.18 |
| 12 | 10 | 0.47±0.05 | 0.61±0.07 | 0.55±0.09 | 0.70±0.06 | 0.68±0.05 | 0.50±0.13 | 0.58±0.19 | 0.80±0.03 | 1.34±0.12 |

**Study on Dimension of Latent Variables ($d_z \leq 5$).** We fix $d_x = 6$ and vary $d_z = \{2, 3, 4\}$ as shown in Table A11. The results indicate that both the Markov network and time-lagged structure are identifiable for lower dimensions. However, as the latent dimension increases, it witnesses a decline in the MCC, which is still the challenge in the continuous optimization of latent process identification (Zhang et al., 2024; Li et al., 2025). Nevertheless, the identifiability of latent space ($R^2$) remains satisfied across different settings.

Table A11: **Results on Different Latent Dimensions.** We run simulations with 5 random seeds, selected based on the best-converged results to avoid local minima.

| $d_x$ | $d_z$ | SHD ($\mathcal{G}_{x_t}$) | TPR | Precision | MCC ($\mathbf{s}_t$) | MCC ($\mathbf{z}_t$) | SHD ($\mathcal{G}_{z_t}$) | SHD ($\mathcal{M}_{lag}$) | $R^2$ |
|---|---|---|---|---|---|---|---|---|---|
| 6 | 2 | 0.12 (±0.04) | 0.86 (±0.02) | 0.85 (±0.04) | 0.9864 (±0.01) | 0.9741 (±0.03) | 0.15 (±0.03) | 0.21 (±0.05) | 0.95 (±0.01) |
| | 3 | 0.18 (±0.06) | 0.83 (±0.02) | 0.80 (±0.04) | 0.9583 (±0.02) | 0.9505 (±0.01) | 0.24 (±0.06) | 0.33 (±0.09) | 0.92 (±0.01) |
| | 4 | 0.23 (±0.02) | 0.80 (±0.06) | 0.74 (±0.01) | 0.9041 (±0.02) | 0.8931 (±0.03) | 0.33 (±0.03) | 0.48 (±0.05) | 0.91 (±0.02) |

**Correct number of Latent Variables.** To determine the correct number of latent factors, empirically, identifying the correct latent dimensionality becomes a model selection problem of finding the minimal yet sufficient number of latent variables. We treat $\hat{d}_z$ as a hyperparameter determined by predictive accuracy and reconstruction error. As shown in Table A12, when $\hat{d}_z < d_z$, reconstruction errors are high (underfitting); as $\hat{d}_z$ approaches $d_z$, errors decrease sharply, and beyond this point, additional dimensions bring negligible improvement, confirming the identifiability of $d_z$.

Table A12: Reconstruction performance under different estimated latent dimensions $\hat{d}_z$ with true $d_z = 4$.

| True $d_z$ | Estimated $\hat{d}_z$ | MSE ↓ | MAE ↓ |
|---|---|---|---|
| 4 | 2 | $0.642 \pm 0.056$ | $0.273 \pm 0.044$ |
| 4 | 3 | $0.172 \pm 0.032$ | $0.231 \pm 0.027$ |
| 4 | 4 | $\mathbf{0.115 \pm 0.018}$ | $\mathbf{0.193 \pm 0.021}$ |
| 4 | 5 | $0.133 \pm 0.019$ | $0.198 \pm 0.022$ |
| 4 | 6 | $0.109 \pm 0.021$ | $0.248 \pm 0.025$ |

Table A13: **Ablation Study on Assumption Violation.** These results verify the necessity of our assumptions in the theoretical analysis.

| Assumption | $d_z$ | $d_x$ | SHD ($J_{\hat{g}}(\hat{\mathbf{x}}_t)$) ↓ | TPR↑ | Precision↑ | MCC($\mathbf{s}_t$)↑ | MCC($\mathbf{z}_t$)↑ | SHD ($J_r(\hat{\mathbf{z}}_t)$)↓ | SHD ($J_d(\hat{\mathbf{z}}_{t-1})$)↓ | $R^2$↑ |
|---|---|---|---|---|---|---|---|---|---|---|
| No Violation | 3 | 6 | 0.18±0.06 | 0.83±0.03 | 0.80±0.04 | 0.9583±0.02 | 0.9505±0.02 | 0.24±0.19 | 0.33±0.09 | 0.92±0.01 |
| Violate A2 (Contextual Variability) | 3 | 6 | 0.26±0.07 | 0.71±0.05 | 0.68±0.05 | 0.7563±0.04 | 0.8820±0.04 | 0.36±0.07 | 0.41±0.08 | 0.67±0.03 |
| Violate A3 (Latent Drift) | 3 | 6 | 0.31±0.05 | 0.67±0.13 | 0.64±0.06 | 0.8645±0.07 | 0.8478±0.08 | 0.39±0.12 | 0.46±0.14 | 0.78±0.21 |
| Violate A5 (Generation Variability) | 3 | 6 | 0.35±0.11 | 0.65±0.12 | 0.60±0.10 | 0.7052±0.13 | 0.9325±0.03 | 0.41±0.08 | 0.47±0.10 | 0.85±0.02 |

**Ablation Study on Conditions.** We further conduct simulation studies to validate the theoretical identifiability guarantees under controlled settings with latent dimension $d_z = 3$ and observation dimension $d_x = 6$. To explicitly assess the necessity of key assumptions in our theory, we intentionally remove specific conditions, which are critical to the identifiability results. The following cases illustrate three distinct violations:

Table A14: **Performance under Higher-Order Latent Dynamics.** The results are averaged over five random seeds. Lower SHD is better, while higher TPR, Precision, MCC, and $R^2$ are better.

| $d_z$ | $d_x$ | SHD ($J_{\hat{g}}(\hat{x}_t)$) | TPR | Precision | MCC ($s_t$) | MCC ($z_t$) | SHD ($J_r(\hat{z}_t)$) | SHD ($J_d(\hat{z}_{t-1})$) | $R^2$ |
|---|---|---|---|---|---|---|---|---|---|
| 3 | 3 | $0.31 \pm 0.01$ | $0.91 \pm 0.02$ | $0.93 \pm 0.03$ | $0.9780 \pm 0.01$ | $0.9825 \pm 0.01$ | $0.26 \pm 0.06$ | $0.30 \pm 0.04$ | $0.93 \pm 0.04$ |
| 3 | 6 | $0.19 \pm 0.07$ | $0.81 \pm 0.04$ | $0.79 \pm 0.08$ | $0.9560 \pm 0.03$ | $0.9520 \pm 0.01$ | $0.25 \pm 0.05$ | $0.34 \pm 0.08$ | $0.91 \pm 0.02$ |
| 3 | 8 | $0.27 \pm 0.06$ | $0.80 \pm 0.04$ | $0.70 \pm 0.05$ | $0.9040 \pm 0.09$ | $0.9610 \pm 0.10$ | $0.34 \pm 0.09$ | $0.32 \pm 0.10$ | $0.93 \pm 0.03$ |
| 3 | 10 | $0.45 \pm 0.06$ | $0.64 \pm 0.10$ | $0.62 \pm 0.13$ | $0.8470 \pm 0.06$ | $0.9630 \pm 0.03$ | $0.30 \pm 0.06$ | $0.39 \pm 0.06$ | $0.91 \pm 0.10$ |
| 3 | 100* | $0.18 \pm 0.03$ | $0.81 \pm 0.04$ | $0.80 \pm 0.03$ | $0.9100 \pm 0.03$ | $0.9570 \pm 0.02$ | $0.22 \pm 0.02$ | $0.28 \pm 0.09$ | $0.92 \pm 0.13$ |

i. **B** (Violation of Assumption 1): To violate the injectivity of linear operators, we use a simple autoregressive process $z_t = z_{t-1} + \epsilon_{z_t}$ with $\epsilon_{z_t} \sim \text{Uniform}(0, 1)$, which fails the injectivity requirement for $L_{z_t|z_{t-1}}$ and $L_{\mathbf{x}_{t-1}|\mathbf{x}_{t+1}}$ (Mattner, 1993).

ii. **A** (Violation of Assumption 1): We enforce the generating function to be a linear orthogonal mapping: $\mathbf{x}_t = W\mathbf{z}_t + \mathbf{s}_t$, thereby violating the latent drift condition.

iii. **C** (Violation of Assumption 3): We constrain the generation variability by setting $\mathbf{s}_t = q(\mathbf{z}_t) + \epsilon_{x_t}$, where $q$ is a fixed mixing function and $\epsilon_{x_t} \sim \mathcal{N}(0, \mathbf{I}_{d_x})$. This results in a linear Gaussian model without heteroscedasticity, undermining the necessary distributional variability, as discussed in (Yao et al., 2022).

As shown in Table A13, the removal of these assumptions leads to a substantial drop in both $R^2$ and MCC for $\mathbf{s}_t$, indicating a failure to recover the latent space and the observation-level causal structure. These findings empirically substantiate the necessity of our theoretical assumptions and delineate the conditions under which identifiability breaks down.

**Experimental Verifications on the Higher-order Markov Process.** To verify our framework's capability on higher-order latent dynamics, we conduct experiments using a second-order latent Markov process. For simulating datasets, the latent process is generated using a leaky non-linear autoregressive model with $L = 2$:

$$z_t = (I - B^{-1}) \left( \sigma \left( \sum_{\ell=1}^{L} \mathbf{W}^{(\ell)} z_{t-\ell} \right) + \epsilon_t^z \right), \quad \epsilon_t^z \sim \mathcal{N}(0, \sigma_z^2 I),$$

where $\sigma(\cdot)$ is leaky ReLU, $\mathbf{W}^{(\ell)}$ are lag-$\ell$ transition matrices, and $B$ is the instantaneous latent causal adjacency matrix. Results are reported below: The table shows that CaDRe maintains high CRL quality and accurate causal discovery, confirming its effectiveness under higher-order latent dynamics. All other settings are aligned with those reported in the main paper. This setting allows us to evaluate how model performance changes under higher-order latent dynamics.

**Hyperparameter Sensitivity.** We test the hyperparameter sensitivity of CaDRe w.r.t. the sparsity and DAG penalty, as these hyperparameters have a significant influence on the performance of structure learning. In this experiment, we set $d_z = 3$ and $d_z = 6$. As shown in Table A16, the results demonstrate robustness across different settings, although the performance of structure learning is particularly sensitive to the sparsity constraint.

We include score-based causal discovery baselines, specifically the BIC method (Schwarz, 1978) and the Generalized Score Function (Huang et al., 2018). These experiments were conducted using the `causal-learn` (Zheng et al., 2024), and the results are now reported in the Table A15.

## D.2 EXPERIMENT RESULTS OF REAL-WORLD DATASETS

We provide comprehensive forecasting tables and comparisons to recent large-scale time-series models, as well as CaDRe generalizations to other domains.

### D.2.1 EXTENDED REAL-WORLD EXPERIMENTS

**Full Results of Weather Forecasting.** To provide stronger evidence of the effectiveness of our approach, we include Table A20 as a supplementary extension of Table 5. This table incorporates three additional methods, TimeMixer (Wang et al., 2024a), TimeXer (Wang et al., 2024b), and xLSTM-Mixer (Kraus et al., 2024), evaluated across three datasets. Our results remain competitive with these methods while maintaining the ability to learn meaningful causal knowledge.

Table A15: Comparison of methods under different latent dimensions.

| Method | $d_x$ | SHD↓ | Precision↑ | Recall↑ | F1↑ |
|---|---|---|---|---|---|
| **CaDRe** | 3 | 0.000 | 1.000 | 1.000 | 1.000 |
| | 6 | 0.185 | 0.803 | 0.830 | 0.816 |
| | 8 | 0.295 | 0.761 | 0.789 | 0.775 |
| | 10 | 0.432 | 0.638 | 0.656 | 0.647 |
| **FCI** | 3 | 0.186 | 0.801 | 0.760 | 0.780 |
| | 6 | 0.384 | 0.476 | 0.394 | 0.431 |
| | 8 | 0.447 | 0.398 | 0.321 | 0.356 |
| | 10 | 0.492 | 0.355 | 0.284 | 0.316 |
| **CDNOD** | 3 | 0.163 | 0.821 | 0.782 | 0.801 |
| | 6 | 0.452 | 0.432 | 0.419 | 0.426 |
| | 8 | 0.509 | 0.365 | 0.312 | 0.336 |
| | 10 | 0.546 | 0.328 | 0.276 | 0.300 |
| **PCMCI** | 3 | 0.139 | 0.843 | 0.803 | 0.823 |
| | 6 | 0.431 | 0.488 | 0.386 | 0.432 |
| | 8 | 0.501 | 0.397 | 0.308 | 0.347 |
| | 10 | 0.548 | 0.365 | 0.284 | 0.320 |
| **LPCMCI** | 3 | 0.116 | 0.864 | 0.823 | 0.843 |
| | 6 | 0.337 | 0.637 | 0.621 | 0.629 |
| | 8 | 0.441 | 0.535 | 0.486 | 0.509 |
| | 10 | 0.487 | 0.482 | 0.432 | 0.456 |
| **BIC** | 3 | 0.192 | 0.755 | 0.712 | 0.732 |
| | 6 | 0.349 | 0.602 | 0.574 | 0.588 |
| | 8 | 0.456 | 0.588 | 0.455 | 0.513 |
| | 10 | 0.493 | 0.408 | 0.406 | 0.406 |
| **Generalized Score Function** | 3 | 0.108 | 0.876 | 0.838 | 0.857 |
| | 6 | 0.298 | 0.692 | 0.659 | 0.675 |
| | 8 | 0.382 | 0.604 | 0.563 | 0.583 |
| | 10 | 0.402 | 0.512 | 0.442 | 0.474 |

Table A16: **Hyperparameter Sensitivity.** We run experiments using 5 different random seeds for data generation and estimation procedures, reporting the average performance on evaluation metrics. "/" indicates loss of explosion. Notably, an excessively large DAG penalty at the beginning of training can result in a loss explosion or the failure of convergence.

| $\alpha$ | $1 \times 10^{-5}$ | $5 \times 10^{-5}$ | $1 \times 10^{-4}$ | $5 \times 10^{-4}$ | $1 \times 10^{-3}$ | $1 \times 10^{-2}$ |
|---|---|---|---|---|---|---|
| SHD | 0.23 | 0.22 | 0.18 | 0.27 | 0.32 | 0.67 |
| $\beta$ | $1 \times 10^{-5}$ | $5 \times 10^{-5}$ | $1 \times 10^{-4}$ | $5 \times 10^{-4}$ | $1 \times 10^{-3}$ | $1 \times 10^{-2}$ |
| SHD | 0.37 | 0.18 | 0.20 | / | / | / |

**Comparisons with Large-scale Time-series Forecasting Models.** Furthermore, we evaluate a broader set of large-scale time-series forecasting models on the Weather dataset. The models considered include iTransformer Liu et al. (2023), PatchTST Nie et al. (2022), Informer Liu et al. (2024a), DLinear Zeng et al. (2023), FAN Ye et al. (2024), TimesNet Wu et al. (2022), and N-Transformer Liu et al. (2022). As shown in Table A17, this extended comparison demonstrates that our method remains competitive against state-of-the-art large-scale time-series forecasting approaches.

**Generalize CaDRe to Other Domains.** To assess the generalization ability of CaDRe beyond climate datasets, we further evaluate it on three standard long-term time-series forecasting benchmarks from diverse domains: ILI (public health), ECL (electricity consumption), and Traffic (road monitoring). As reported in Table A18, CaDRe consistently achieves competitive or superior forecasting accuracy across different horizons, often outperforming recent transformer- and CNN-based

Table A17: **Extended Results on Weather Forecasting.** Lower MSE/MAE is better. **Bold** numbers represent the best performance among the models, while underlined numbers denote the second-best.

| Dataset | Length | CaDRe | | iTransformer | | Informer | | PatchTST | | DLinear+FAN | | TimesNet | | DLinear | | N-Transformer | |
|---------|--------|-------|-------|-------|-------|-------|-------|-------|-------|-------|-------|-------|-------|-------|-------|-------|-------|
| | | MSE | MAE | MSE | MAE | MSE | MAE | MSE | MAE | MSE | MAE | MSE | MAE | MSE | MAE | MSE | MAE |
| Weather | 48 | **0.125** | **0.167** | 0.140 | 0.179 | 0.177 | 0.218 | 0.148 | 0.188 | 0.158 | 0.217 | 0.138 | 0.191 | 0.156 | 0.198 | 0.143 | 0.195 |
| | 96 | **0.157** | **0.203** | 0.168 | 0.214 | 0.225 | 0.259 | 0.187 | 0.226 | 0.199 | 0.256 | 0.180 | 0.231 | 0.186 | 0.229 | 0.199 | 0.246 |
| | 144 | 0.180 | **0.225** | **0.172** | 0.225 | 0.278 | 0.297 | 0.207 | 0.242 | 0.312 | 0.274 | 0.190 | 0.244 | 0.199 | 0.244 | 0.225 | 0.267 |
| | 192 | 0.207 | **0.248** | **0.193** | 0.241 | 0.354 | 0.348 | 0.234 | 0.265 | 0.238 | 0.298 | 0.212 | 0.265 | 0.217 | 0.261 | 2.960 | 0.315 |

baselines. These results indicate that CaDRe is not restricted to climate data but extends robustly to heterogeneous time-series domains.

Table A18: **Forecasting results on ILI, ECL, and Traffic datasets.** Lower MSE/MAE is better. **Bold** numbers represent the best performance among the models, while underlined numbers denote the second-best.

| Dataset | Length | CaDRe | | N-Transformer | | Autoformer | | MICN | | TimesNet | |
|---------|--------|-------|-------|-------|-------|-------|-------|-------|-------|-------|-------|
| | | MSE | MAE | MSE | MAE | MSE | MAE | MSE | MAE | MSE | MAE |
| ILI | 18-6 | **1.200** | **0.691** | 1.491 | 0.757 | 2.637 | 1.094 | 4.847 | 1.570 | 2.406 | 0.840 |
| | 72-24 | **1.856** | **0.833** | 2.270 | 0.988 | 2.653 | 1.116 | 4.776 | 1.556 | 2.551 | 1.039 |
| | 144-48 | **1.796** | **0.878** | 2.227 | 1.018 | 2.696 | 1.139 | 4.917 | 1.584 | 2.978 | 1.123 |
| | 216-72 | **2.010** | **0.984** | 2.595 | 1.081 | 2.960 | 1.167 | 4.804 | 1.584 | 2.696 | 1.098 |
| ECL | 18-6 | **0.114** | **0.216** | 0.128 | 0.236 | 0.136 | 0.254 | 0.250 | 0.338 | 0.134 | 0.242 |
| | 72-24 | **0.121** | **0.220** | 0.134 | 0.242 | 0.144 | 0.257 | 0.258 | 0.342 | 0.140 | 0.246 |
| | 144-48 | **0.124** | **0.225** | 0.149 | 0.256 | 0.163 | 0.275 | 0.271 | 0.353 | 0.155 | 0.260 |
| | 216-72 | **0.131** | **0.232** | 0.166 | 0.271 | 0.175 | 0.287 | 0.279 | 0.357 | 0.169 | 0.274 |
| Traffic | 18-6 | **0.475** | **0.287** | 0.797 | 0.347 | 0.554 | 0.322 | 0.487 | 0.307 | 0.781 | 0.337 |
| | 72-24 | **0.454** | **0.276** | 0.625 | 0.319 | 0.508 | 0.318 | 0.452 | 0.303 | 0.608 | 0.307 |
| | 144-48 | **0.450** | **0.275** | 0.574 | 0.314 | 0.497 | 0.319 | 0.412 | 0.282 | 0.553 | 0.296 |
| | 216-72 | **0.473** | **0.287** | 0.593 | 0.325 | 0.524 | 0.330 | 0.400 | 0.278 | 0.564 | 0.303 |

**Interpretation of Latent Variables and Causal Relations in Other Domains**   To enhance interpretability, we provide explanations of the latent variables and causal relations discovered by CADRE in different domains. These interpretations clarify how the model captures hidden factors and directional dependencies across datasets (Table A19). We further visualize the causal discovery

Table A19: Interpretation of latent variables and causal relations across datasets.

| Dataset | Meaning of Latent Variables | Meaning of Causal Relations |
|---------|-----------------------------|-----------------------------|
| **ILI (Public Health)** | Latent variables capture hidden epidemic dynamics such as *viral transmission strength, seasonality, and social mobility trends* that are not directly observed in infection counts. | Causal edges represent short-term influences between *regional infection rates* (e.g., spatial spread or temporal cross-region effects). |
| **ECL (Electricity Consumption)** | Latents encode underlying energy demand drivers—*temperature, population activity patterns, and industrial load cycles*—that evolve over time. | Causal links describe dependencies between *regional electricity stations or customer groups* (e.g., peak-demand propagation). |
| **Traffic (Road Monitoring)** | Latents represent hidden mobility factors such as *congestion propagation speed, commuting intensity, and network bottleneck dynamics*. | Causal edges correspond to directional *traffic flow influences* between neighboring sensors (e.g., upstream → downstream road segments). |

results on the **Traffic** dataset. The corresponding Jacobian matrix is:

$$\mathbf{J}_g(\mathbf{x}_t) = \begin{bmatrix} 0 & 0.2323 & 0.4551 \\ 1.4345 & 0 & 0.0369 \\ 1.9435 & 0.6632 & 0 \end{bmatrix},$$

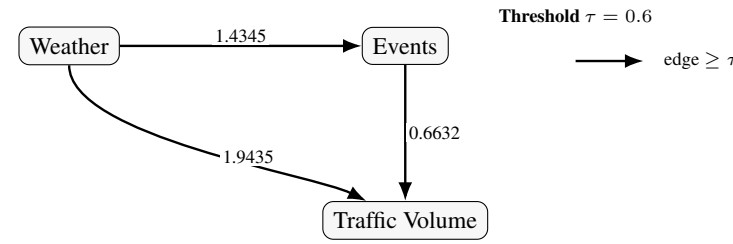

Figure A7: Causal graph for the **Traffic** dataset. Edges are labeled by entries of the Jacobian $J$ and styled by threshold $\tau = 0.6$.

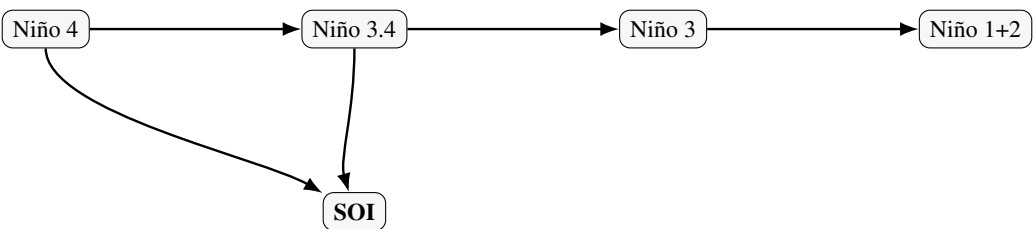

Figure A8: Causal graph for ENSO teleconnections. The chain Niño 4 $\rightarrow$ Niño 3.4 $\rightarrow$ Niño 3 $\rightarrow$ Niño 1+2 with diagonal links to **SOI**.

where we set a threshold $\tau = 0.6$ for causal visualization. We also provide the results on ENSO in Figure A8 following the similar implementations.

Since there is no physical evidence for latent variables in the traffic domain, we consider visualizing these latent components as part of future work.

**Runtime and Computational Efficiency.** We report the computational cost of the different methods considered. The comparison considers metrics including training time, memory usage, and corresponding performance MSE in the forecasting task. Note that inference time is not included in the comparison, as our work focuses on causal structure learning through continuous optimization rather than constraint-based methods. Figure A9 shows that our CaDRe method simultaneously learns the causal structure while achieving the lowest MSE, highlighting the importance of building a transparent and interpretable model. Furthermore, CaDRe exhibits similar training time and memory usage compared to mainstream time-series forecasting models in the lightweight track.

**Unrotated visualization results on climate dataset.** We show the unrotated, full visualization of causal learning on climate dataset at Figure A10.

## E MORE DISCUSSIONS

We talk about the assumptions is testable or not, and then, wextend identifiability to higher-order Markov cases and allow time-lagged effects in observations.

### E.1 ROLE OF FORECASTING AND CAUSAL IDENTIFICATION.

A clear connection exists between causal identification and forecasting performance in our framework. Better latent space recovery directly enhances predictive accuracy, as a well-approximated latent representation $\mathbf{z}_t \approx h(\hat{\mathbf{z}}_t)$ ensures that $p(\mathbf{x}_t \mid \mathbf{z}_t) = p(\mathbf{x}_t \mid h(\mathbf{z}_t))$, thereby minimizing reconstruction risk. Although graph-recovery metrics are not explicitly optimized for forecasting, accurate causal graphs introduce inductive biases that improve forecasting efficiency and reduce uncertainty by preserving physically meaningful dependencies in the data. This alignment between causal structure and predictive skill has been emphasized in recent studies on causally-informed modeling in climate science (Runge et al., 2019a). We have added this discussion to clarify the complementary roles of causal discovery and forecasting in our revised manuscript.

### E.2 TESTING ASSUMPTIONS IN CLIMATE TIME SERIES

To verify that the future climate variable $\mathbf{x}_{t+1}$ exhibits distinct statistical behavior conditioned on past states $\mathbf{x}_{t-1}$, we examined a multivariate climate time series $\mathbf{X} \in \mathbb{R}^{T \times d_x}$. For a selected variable, $\mathbf{x}_t$,

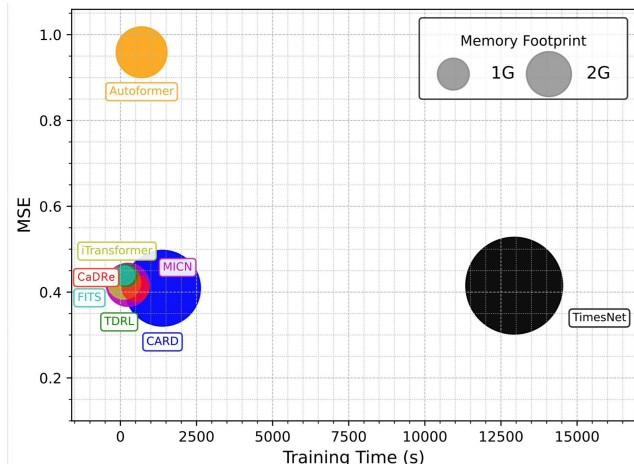

Figure A9: **Comparison of Computational Cost.** Different colors represent different methods, while the size of the circles corresponds to memory usage. The prediction length is set to 96.

the lagged variables $\mathbf{x}_{t-1}$ and $\mathbf{x}_{t+1}$ were constructed and analyzed. The range of $\mathbf{x}_{t-1}$ was divided into quantile bins, and the corresponding conditional distributions of $\mathbf{x}_{t+1}$ were visualized. The results show clear regime dependence: higher or lower $\mathbf{x}_{t-1}$ values correspond to systematically shifted or widened $\mathbf{x}_{t+1}$ distributions, confirming temporal memory in the process.

Assumptions involved with $\mathbf{z}_t$ are not immediately verified, since latent variables are inherently unobserved. But these assumption can be tested indirectly: Under these assumptions, we can first estimate the model, and then we can test whether these assumptions hold true in real-world climate data. To investigate whether the inferred series $\hat{\mathbf{z}}_t$ reflects distributional changes w.r.t. the observed climate variable $\mathbf{x}_t$, we analyze the conditional behavior $P(\hat{\mathbf{z}}_t \mid \mathbf{x}_t)$. Using quantile bins of $\mathbf{x}_t$, we visualize how the value of $\hat{\mathbf{z}}_t$ shifts across regimes of $\mathbf{x}_t$. Four diagnostics are shown: (i) conditional histograms, (ii) conditional boxplots, (iii) the joint density $(\mathbf{x}_t, \mathbf{z}_t)$, and (iv) conditional means with 95% confidence intervals. The panels reveal that higher values of $\mathbf{x}_t$ correspond to systematically broader and shifted distributions of $\hat{\mathbf{z}}_t$, confirming that $\hat{\mathbf{z}}_t$ captures regime-dependent variability in the climate data.

### E.3 IDENTIFIABILITY OF LATENT SPACE IN $n$-ORDER MARKOV PROCESS

We state conditions under which block-wise latent recovery holds for general order-$n$ dynamics as follows.

**Theorem A2.** *(**Identifiability of Latent Space in $n$-Order Markov Process**) Suppose observed variables and hidden variables follow the data-generating process in Eq. (1), and estimated observations match the true joint distribution of $\{\mathbf{x}_{t-n}, \ldots, \mathbf{x}_{t-1}, \mathbf{x}_t, \ldots, \mathbf{x}_{t+n}, \mathbf{x}_{t+n+1}, \ldots, \mathbf{x}_{t+2n}\}$. The following assumptions are imposed:*

*A1' (Computable Probability:) The joint, marginal, and conditional distributions of $(\mathbf{x}_t, \mathbf{z}_t)$ are all bounded and continuous.*

*A2' (Contextual Variability:) The operators $L_{\mathbf{x}_{t+n+1:t+2n}|\mathbf{z}_{t:t+n}}$ and $L_{\mathbf{x}_{t-n:t-1}|\mathbf{x}_{t+n+1:t+2n}}$ are injective and bounded.*

*A3' (Latent Drift:) For any $\mathbf{z}_{t:t+n}^{(1)}, \mathbf{z}_{t:t+n}^{(2)} \in \mathcal{Z}_t$ where $\mathbf{z}_{t:t+n}^{(1)} \neq \mathbf{z}_{t:t+n}^{(2)}$, we have $p(\mathbf{x}_t|\mathbf{z}_{t:t+n}^{(1)}) \neq p(\mathbf{x}_t|\mathbf{z}_{t:t+n}^{(2)})$.*

*A4' (Differentiability:) There exists a functional $F$ such that $F\left[ p_{\mathbf{x}_{t:t+n}|\mathbf{z}_{t:t+n}}(\cdot \mid \mathbf{z}_{t:t+n}) \right] = h_z(\mathbf{z}_{t:t+n})$ for all $\mathbf{z}_{t:t+n} \in \mathcal{Z}_{t:t+n}$, where $h_z$ is differentiable.*

*Then we have $\hat{\mathbf{z}}_{t:t+n} = h_z(\mathbf{z}_{t:t+n})$, where $h_z : \mathbb{R}^{d_z \times n} \to \mathbb{R}^{d_z \times n}$ is an invertible and differentiable function.*

If an $n$-order Markov process exhibits conditional independence across different time lags, block-wise identifiability of the conditioning variables can still be achieved using 3n measurements. For instance, when the lag is 2, once block-wise identifiability of the joint variables $[\mathbf{z}_t, \mathbf{z}_{t+1}]$ and $[\mathbf{z}_{t+1}, \mathbf{z}_{t+2}]$ is established, and given the known temporal direction $\mathbf{z}_t \to \mathbf{z}_{t+1}$, one can disambiguate $\mathbf{z}_t$ and $\mathbf{z}_{t+1}$

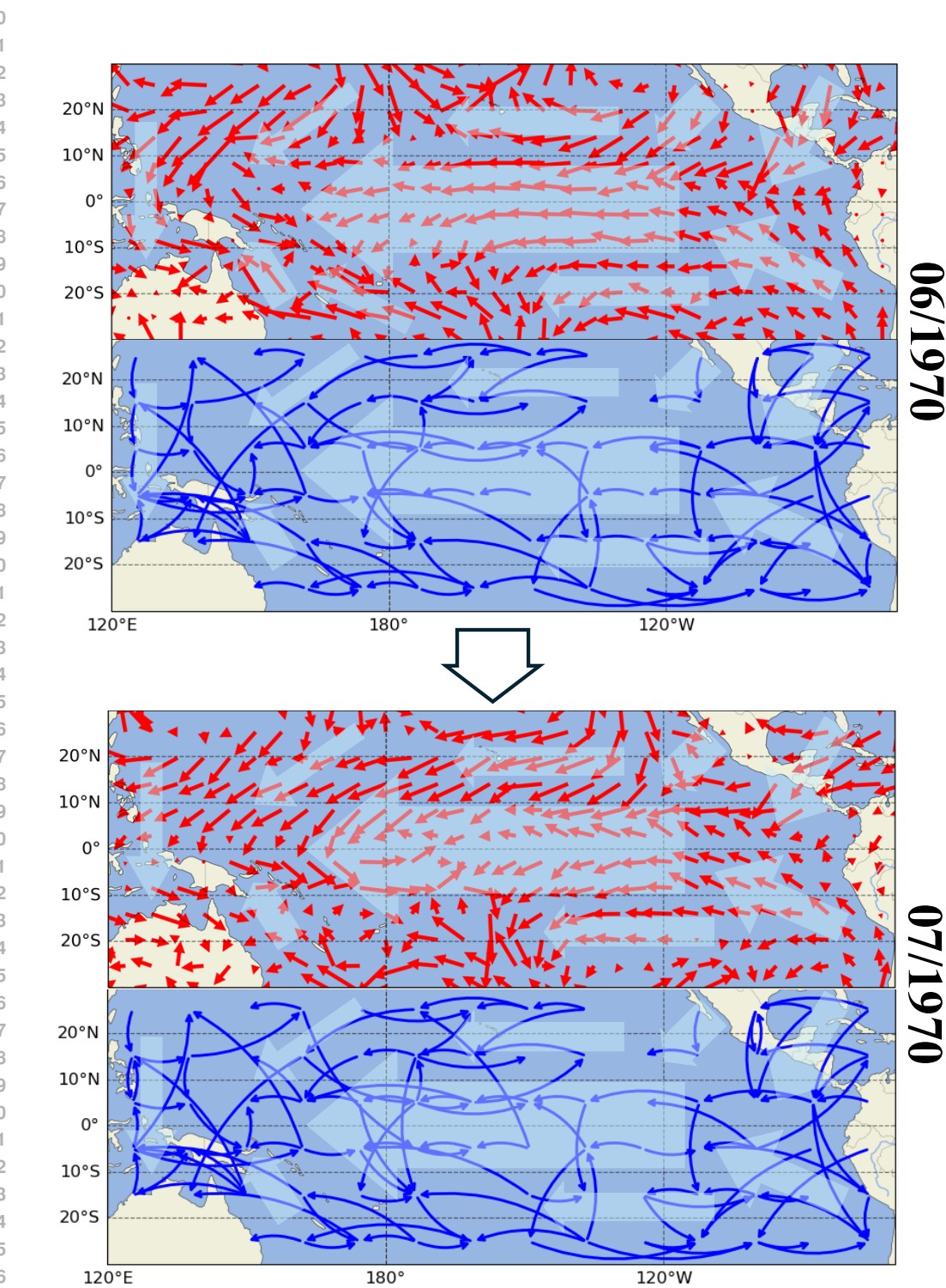

Figure A10: Unrotated visualizations on climate dataset.

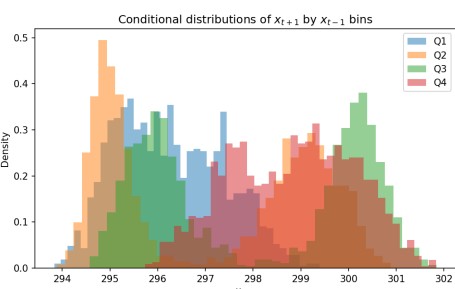

(a) Conditional distributions of $\mathbf{x}_{t+1}$ grouped by quantile bins of $\mathbf{x}_{t-1}$. Distinct shapes indicate dependence on the previous climate state.

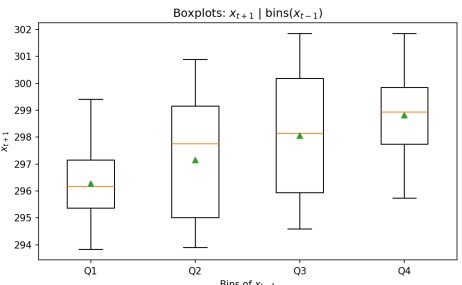

(b) Boxplots of $\mathbf{x}_{t+1}$ for different regimes of $\mathbf{x}_{t-1}$. Median and spread shift with past state, highlighting nonlinear temporal dependence.

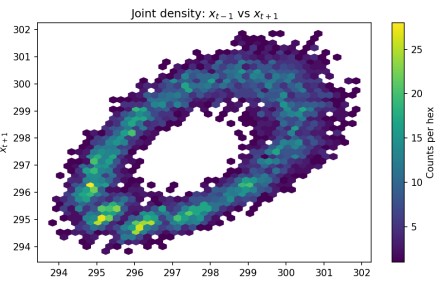

(c) Joint density of $\mathbf{x}_{t-1}$ and $\mathbf{x}_{t+1}$. The slanted density structure confirms a directional relationship between past and future values.

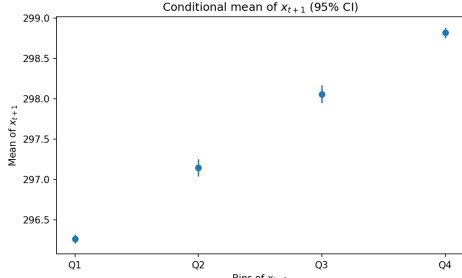

(d) Conditional means of $\mathbf{x}_{t+1}$ with 95% confidence intervals for each $\mathbf{x}_{t-1}$ bin. The monotonic trend evidences lagged dependence in the climate variable.

Figure A11: **Verification of Assumption A2 in Climate Data.** Each panel visualizes a distinct aspect of the distributional relationship between $\mathbf{x}_{t+1}$ and $\mathbf{x}_{t-1}$.

under mild variability assumptions. Subsequently, the same strategy as in Theorem A1 can be applied to achieve component-wise identifiability of $\mathbf{z}_t$ and $\mathbf{z}_{t+1}$ by leveraging conditional independencies given $\mathbf{z}_{t-2}$ and $\mathbf{z}_{t-1}$.

### E.4 Allowing Time-Lagged Causal Relationships in Observations

In this section, we demonstrate that our proposed framework is compatible with the consideration of time-lagged effects, by providing *potential solutions*.

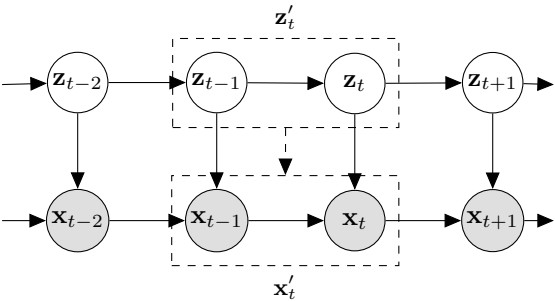

Figure A13: **4-measurement model with time-lagged effects in observed space.** $\mathbf{x}_t$ could be considered as the directed (dominating) measurement of $\mathbf{z}_t$, and $\mathbf{x}_{t-2}$, $\mathbf{x}_{t-1}$, and $\mathbf{x}_{t+1}$ provide indirect measurements of $\mathbf{z}_t$. For identifying the time-lagged causal relationships in observed space, we consider $\mathbf{z}'_t = (\mathbf{z}_{t-1}, \mathbf{z}_t)$ as the new latent variables, and $\mathbf{x}'_t = (\mathbf{x}_{t-1}, \mathbf{x}_t)$ as the new observed variables, to apply our *functional equivalence* in Theorem 2.

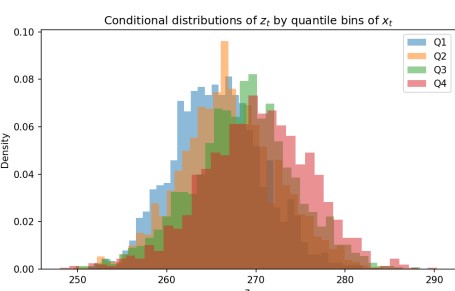

(a) Conditional distributions of $\mathbf{z}_t$ across quantile bins of $\mathbf{x}_t$, illustrating regime-dependent spread and shift.

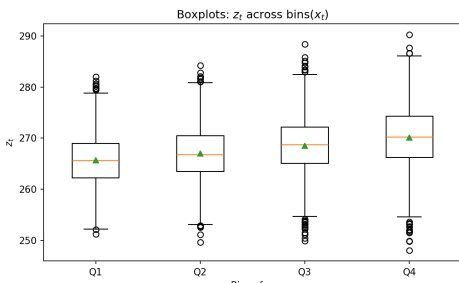

(b) Boxplots of $\mathbf{z}_t$ grouped by quantile bins of $\mathbf{x}_t$, showing distinct medians and dispersion across regimes.

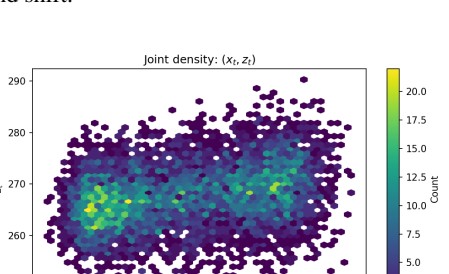

(c) Joint hexbin plot of $(\mathbf{x}_t, \mathbf{z}_t)$, highlighting contemporaneous dependence and heteroskedasticity.

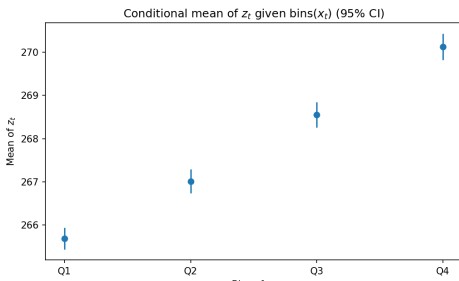

(d) Conditional means of $\mathbf{z}_t$ with 95% confidence intervals across $\mathbf{x}_t$ bins, showing monotonic increase and growing uncertainty.

Figure A12: **Verification of Assumption A3 in Climate Data.** Each panel visualizes how the series $\mathbf{z}_t$ varies across quantile regimes of the observed climate variable $\mathbf{x}_t$.

### E.4.1 PHASE I: IDENTIFYING LATENT VARIABLES FROM TIME-LAGGED CAUSALLY-RELATED OBSERVATIONS

For the identification of latent variables, we adopt the strategy outlined in (Carroll et al., 2010; Hu & Shum, 2012) to construct a spectral decomposition. We extend this approach to develop a proof strategy that establishes the identifiability of the latent space, as stated in Corollary 2.2.

We begin by defining the 4-measurement model, which includes time-series data with time-lagged effects in the observed space as a special case.

**Definition 6** (4-Measurement Model). $\mathbf{Z} = \{\mathbf{z}_{t-2}, \mathbf{z}_{t-1}, \mathbf{z}_t, \mathbf{z}_{t+1}\}$ *represents latent variables in four continuous time steps, respectively. Similarly,* $\mathbf{X} = \{\mathbf{x}_{t-2}, \mathbf{x}_{t-1}, \mathbf{x}_t, \mathbf{x}_{t+1}\}$ *are observed variables that directly measure* $\mathbf{z}_{t-2}, \mathbf{z}_{t-1}, \mathbf{z}_t, \mathbf{z}_{t+1}$ *using the same generating functions g. The model is defined by the following properties:*

- *The transformation within* $\mathbf{z}_{t-2}, \mathbf{z}_{t-1}, \mathbf{z}_t, \mathbf{z}_{t+1}$ *is not measure-preserving.*

- *Joint density of* $\mathbf{x}_{t-2}, \mathbf{x}_{t-1}, \mathbf{x}_t, \mathbf{x}_{t+1}, \mathbf{z}_t$ *is a product measure w.r.t. the Lebesgue measure on* $\mathcal{X}_{t-2} \times \mathcal{X}_{t-1} \times \mathcal{X}_t \times \mathcal{X}_{t+1} \times \mathcal{Z}_t$ *and a dominating measure $\mu$ is defined on $\mathcal{Z}_t$.*

- ***Limited feedback**:* $p(\mathbf{x}_t \mid \mathbf{x}_{t-1}, \mathbf{z}_t, \mathbf{z}_{t-1}) = p(\mathbf{x}_t \mid \mathbf{x}_{t-1}, \mathbf{z}_t)$.

- *The distribution over* $(\mathbf{X}, \mathbf{Z})$ *is Markov and faithful to a DAG.*

Limited feedback explicitly assumes that future events do not cause past events and excludes instantaneous effects from $\mathbf{x}_t$ to $\mathbf{z}_t$. As illustrated in Figure A13, $\mathbf{x}_{t-2}, \mathbf{x}_{t-1}, \mathbf{x}_t, \mathbf{x}_{t+1}$ are defined as different measurements of $\mathbf{z}_t$, forming a temporal structure characteristic of a typical 4-measurement model. Under the data-generating process depicted in Figure A13, and based on the assumption of

limited feedback, we propose the following framework:

$$
\begin{aligned}
p(\mathbf{x}_{t-1}, \mathbf{x}_t, \mathbf{x}_{t+1}, \mathbf{x}_{t+2}) &= \int_{\mathcal{Z}_t} p(\mathbf{x}_{t+1} \mid \mathbf{x}_t, \mathbf{z}_t) p(\mathbf{x}_t | \mathbf{x}_{t-1}, \mathbf{z}_t) p(\mathbf{x}_{t-1}, \mathbf{x}_{t-2}, \mathbf{z}_t) dz_t \\
&= \int_{\mathcal{Z}_t} p(\mathbf{x}_{t+1} \mid \mathbf{x}_t, \mathbf{z}_t) p(\mathbf{x}_t, \mathbf{x}_{t-1}, \mathbf{z}_t) p(\mathbf{x}_{t-2} \mid \mathbf{z}_t, \mathbf{x}_{t-1}) dz_t.
\end{aligned}
\tag{A47}
$$

**Discussion of achieving the identifiability of latent space.**   Comparing Eq. (A47) with Eq. (A3), which represents the foundational result for proving the identifiability of latent space under the 3-measurement model, we extend the identification strategy from (Carroll et al., 2010; Hu & Shum, 2012) to the 4-measurement model. This forms the critical step in our identification process. We adopt assumptions analogous to those in (Carroll et al., 2010; Hu & Shum, 2012) and Theorem 1, and suppose the followings:

  i. The joint distribution of $(\mathbf{X}, \mathbf{Z})$ and all their marginal and conditional densities are bounded and continuous.

  ii. The linear operators $L_{\mathbf{x}_{t+1}|x_t, z_t}$ and $L_{x_{t-2}, \mathbf{x}_{t-1}, x_t, \mathbf{x}_{t+1}, z_t}$ are injective for bounded function space.

  iii. For all $\mathbf{z}_t, \mathbf{z}_t' \in \mathcal{Z}_t$ $(\mathbf{z}_t \neq \mathbf{z}_t')$, the set $\{\mathbf{x}_t : p(\mathbf{x}_t|\mathbf{z}_t) \neq p(\mathbf{x}_t|\mathbf{z}_t')\}$ has positive probability.

hold true. Similar to the proof of our identifiability of latent space in Section A.2, except for the conditional independence introduced by the temporal structure, the key assumptions include an injective linear operator to enable the recovery of the density function of latent variables and distinctive eigenvalues to prevent eigenvalue degeneracy. The primary difference is the property *limited feedback*, where we can adopt the strategy in (Carroll et al., 2010) to construct a unique spectral decomposition, where $(\mathbf{x}_{t-2}, \mathbf{x}_{t-1}, \mathbf{x}_t, \mathbf{x}_{t+1}, \mathbf{z}_t)$ correspond to $(X, S, Z, Y, X^*)$, respectively. Following this, we apply the key steps of our identification process as detailed in the Appendix A.2. Ultimately, we can establish that the block $(\mathbf{z}_t, \mathbf{x}_t)$ is identifiable up to an invertible transformation:

$$
(\hat{\mathbf{z}}_t, \hat{\mathbf{x}}_t) = h_{x,z}(\mathbf{z}_t, \mathbf{x}_t).
\tag{A48}
$$

where $h_{x,z} : \mathbb{R}^{d_x + d_z} \to \mathbb{R}^{d_x + d_z}$ is an invertible function. Since the observation $\mathbf{x}_t$ is known and suppose $\hat{\mathbf{x}}_t = \mathbf{x}_t$, this relationship indeed represents an invertible transformation between $\hat{\mathbf{z}}_t$ and $\mathbf{z}_t$ as

$$
\hat{\mathbf{z}}_t = h_z(\mathbf{z}_t).
\tag{A49}
$$

With an additional assumption of a sparse latent Markov network, we achieve component-wise identifiability of the latent variables, as stated in Theorem A1 in appendix, leveraging the proof strategies of (Zhang et al., 2024; Li et al., 2025). These results are stronger than those in (Carroll et al., 2010).

### E.4.2   PHASE II: IDENTIFYING TIME-LAGGED OBSERVATION CAUSAL GRAPH

**Unified Modeling across Neighboring Time Points.**   In the presence of time-lagged effects in the observed space, such as $\mathbf{x}_{t-1} \to \mathbf{x}_t$, alongside the causal DAG within $\mathbf{x}_t$, as depicted in Figure A13, we show that by introducing an expanded set of latent variables $\mathbf{z}_t' = (\mathbf{z}_{t-1}, \mathbf{z}_t)$ and an expanded set of observed variables $\mathbf{x}_t' = (\mathbf{x}_{t-1}, \mathbf{x}_t)$, the property of functional equivalence is preserved. Moreover, identifiability continues to hold, and, broadly speaking, it becomes more accessible due to the incorporation of Granger causality principles in time-series data (Freeman, 1983), if we assume that future events cannot influence or cause past events.

**Functional Equivalence in Presence of Time-Lagged Effects.** As shown in Figure A14, we show that, if we consider the time-lagged causal relationship in observed space, it still can be processed with the technique as in our paper proposed, through considering time-lagged causal relationships as a part of the causal graph over observed variables, by reformulating $\mathbf{z}_t' = (\mathbf{z}_{t-1}, \mathbf{z}_t)$, $\mathbf{x}_t' = (\mathbf{x}_{t-1}, \mathbf{x}_t)$ and $\mathbf{s}_t' = (\mathbf{s}_{t-1}, \mathbf{s}_t)$, to apply the Corollary 2.2. Specifically, the time-lagged effects from $\mathbf{x}_{t-2}$ can be considered as side information, which does not make difference to causal relationships from $\mathbf{x}_{t-1}$ to $\mathbf{x}_t$ and its corresponding ICA form.

### E.4.3   ESTIMATION METHODOLOGY

**Slided Window.**   Building on the analysis above, we aggregate two adjacent time-indexed observations into a single new observation. By employing a sliding window with a step size of 1, we obtain $T - 1$ new observations along with their corresponding latent variables, thereby aligning with the estimation methodology described in Section 4.

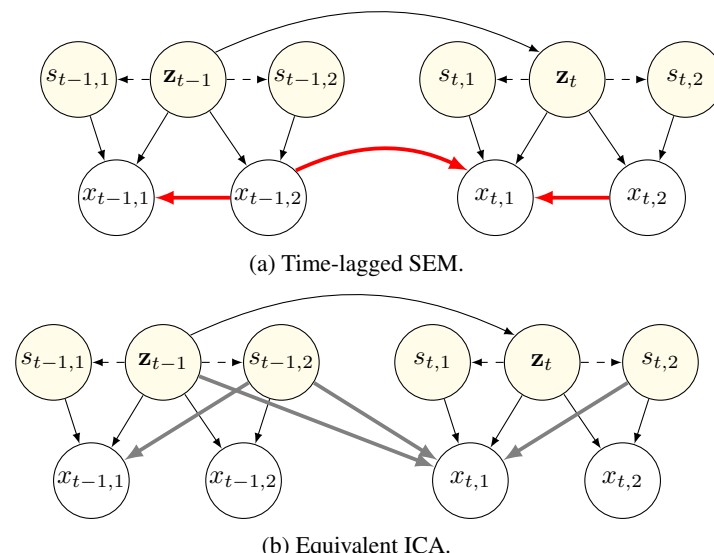

(a) Time-lagged SEM.

(b) Equivalent ICA.

Figure A14: Equivalent time-lagged SEM and ICA in the case with time-lagged causal relationships in observed space. The red lines in Figure A14a indicate that information are transmitted by the instantaneous and the time-lagged causal graphs over observed variables, while the gray lines in Figure A14b represent that the information transitions are equivalent to originating from contemporary $\mathbf{s}_t$ and previous $(\mathbf{z}_t, s_{t-1,2})$ within the mixing structure.

**Structure Prunning.** For structure learning, given the assumption that future climate cannot cause past climate, we can mask $\frac{1}{4}$ of elements in the causal adjacency matrix during implementation, as depicted in Figure A15. Compared with the original implementation, the masking simplifies the difficulty of optimization by reducing the degrees of freedom in the graph.

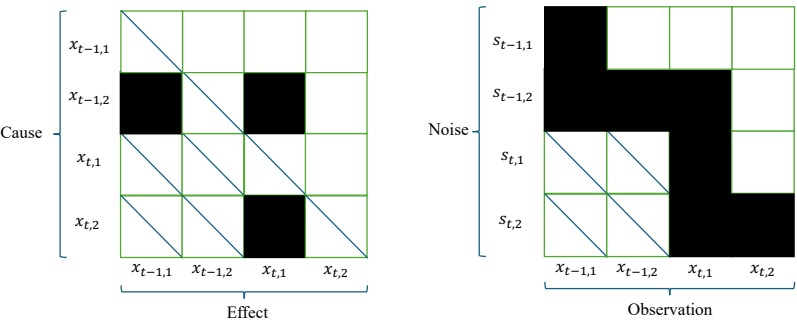

(a) Causal adjacency matrix of SEM.  (b) Mixing matrix of equivalent ICA.

Figure A15: Interpreting Figure A14 with causal adjacency matrix of the SEM and the mixing matrix of the equivalent ICA. The diagonal lines indicate masked elements, as future events cannot cause past events, and self-loops are not permitted. Black blocks represent the presence of a causal relationship or functional dependency in the generating function $g_m$, while white blocks indicate the absence of such a relationship.

## F    BROADER IMPACTS

The proposed CaDRe framework offers a substantial advancement in climate science by enabling the joint identification of latent dynamic processes and causal structures over observed variables from purely observational data. Understanding these structures is critical for interpreting complex atmospheric phenomena, improving forecasting accuracy, and informing climate-related decision-making. By providing identifiability guarantees without relying on restrictive assumptions, CaDRe addresses fundamental limitations in existing climate modeling approaches, particularly in the presence of latent confounders and observational noise.

The ability to recover interpretable latent drivers and causal graphs directly from climate data enhances scientific understanding and supports more transparent and robust climate models. This is especially valuable for anticipating and responding to climate variability and extreme events. Moreover, the theoretical framework underlying CaDRe extends to other scientific domains involving spatiotemporal processes, but its primary impact lies in improving the causal interpretability and empirical grounding of climate analyses. As such, CaDRe represents a step toward causally principled climate modeling, with the potential to inform both scientific inquiry and policy development.

## G    DISCLOSURE OF LLM USAGE

In accordance with the ICLR 2026 policy on the use of Large Language Models (LLMs), we disclose that LLMs were used solely to correct grammatical issues in this paper. No part of the research ideation, experimental design, implementation, or analysis relied on LLMs. The authors take full responsibility for the content of the paper.

Table A20: **Full Results on Temperature Forecasting across Different Datasets.** Lower MSE/MAE is better. **Bold** numbers represent the best performance among the models, while underlined numbers denote the second-best.

| Dataset | Length | CaDRe | | TDRL | | CARD | | FITS | | MICN | | iTransformer | | TimesNet | | Autoformer | | Timer-XL | | TimeMixer | | TimeXer | | xLSTM-Mixer | |
|---|---|---|---|---|---|---|---|---|---|---|---|---|---|---|---|---|---|---|---|---|---|---|---|---|---|
| | | MSE | MAE | MSE | MAE | MSE | MAE | MSE | MAE | MSE | MAE | MSE | MAE | MSE | MAE | MSE | MAE | MSE | MAE | MSE | MAE | MSE | MAE | MSE | MAE |
| CESM2 | 96 | 0.410 | 0.483 | 0.439 | 0.507 | 0.409 | 0.484 | 0.439 | 0.508 | 0.417 | 0.486 | 0.422 | 0.491 | 0.415 | 0.486 | 0.959 | 0.735 | 0.433 | 0.497 | 0.412 | 0.439 | 0.485 | 0.465 | 0.367 | 0.452 |
| CESM2 | 192 | 0.412 | 0.487 | 0.440 | 0.508 | 0.422 | 0.493 | 0.447 | 0.515 | 1.559 | 0.984 | 0.425 | 0.495 | 0.417 | 0.497 | 1.574 | 0.972 | 0.454 | 0.524 | 0.445 | 0.424 | 0.493 | 0.505 | 0.434 | 0.498 |
| CESM2 | 336 | 0.413 | 0.485 | 0.441 | 0.505 | 0.421 | 0.497 | 0.482 | 0.536 | 2.091 | 1.173 | 0.426 | 0.494 | 0.423 | 0.499 | 1.845 | 1.078 | 0.527 | 0.565 | 0.541 | 0.421 | 0.499 | 0.508 | 0.448 | 0.471 |
| Weather | 96 | 0.157 | 0.203 | 0.442 | 0.511 | 0.423 | 0.497 | 0.172 | 0.221 | 0.199 | 0.256 | 0.168 | 0.214 | 0.180 | 0.231 | 0.225 | 0.338 | 0.167 | 0.219 | 0.163 | 0.219 | 0.163 | 0.220 | 0.160 | 0.196 |
| Weather | 192 | 0.207 | 0.248 | 0.492 | 0.545 | 0.482 | 0.544 | 0.216 | 0.260 | 0.238 | 0.298 | 0.193 | 0.241 | 0.212 | 0.265 | 0.304 | 0.368 | 0.236 | 0.268 | 0.214 | 0.261 | 0.226 | 0.249 | 0.194 | 0.238 |
| Weather | 336 | 0.270 | 0.314 | 0.536 | 0.612 | 0.525 | 0.596 | 0.386 | 0.439 | 0.316 | 0.496 | 0.426 | 0.494 | 0.423 | 0.499 | 0.354 | 0.408 | 0.274 | 0.312 | 0.270 | 0.321 | 0.275 | 0.313 | 0.275 | 0.316 |
| ERSST | 96 | 0.145 | 0.268 | 0.187 | 0.268 | 0.197 | 0.273 | 0.539 | 0.297 | 0.726 | 0.765 | 0.247 | 0.264 | 0.432 | 0.508 | 0.953 | 0.272 | 0.163 | 0.259 | 0.172 | 0.272 | 0.365 | 0.344 | 0.345 | 0.255 |
| ERSST | 192 | 0.208 | 0.307 | 0.214 | 0.293 | 0.233 | 0.375 | 0.226 | 0.752 | 1.263 | 0.892 | 0.251 | 0.535 | 0.452 | 0.585 | 1.024 | 0.908 | 0.210 | 0.294 | 0.214 | 0.302 | 0.372 | 0.367 | 0.371 | 0.297 |
| ERSST | 336 | 0.305 | 0.361 | 0.462 | 0.388 | 0.487 | 0.484 | 0.439 | 0.535 | 1.173 | 1.172 | 0.305 | 0.659 | 0.581 | 0.607 | 1.387 | 1.353 | 0.352 | 0.337 | 0.439 | 0.394 | 0.429 | 0.448 | 0.476 | 0.357 |

