# OpenReview forum: "Learning General Causal Structures with Hidden Dynamic Process for Climate Analysis"
_ICLR.cc/2026/Conference — ICLR 2026 Conference Desk Rejected Submission_

### Official Review · Reviewer_3HPk · 2025-10-29

**Soundness:** 4
**Presentation:** 3
**Contribution:** 3
**Rating:** 6
**Confidence:** 4

**Summary:**

Motivated by applications in climate science where latent driving forces affect a non-stationary process,
this work addresses time series data where causal interactions exist not only among observed variables but also
among a set of latent variables, either time-lagged or instantaneous. It integrates techniques from CRL within causal discovery to both
(i) reconstruct the latent variables and their causal structure and (ii) discover the causal graph over observed variables.
The authors provide identification theory on both aspects, showing identifiability of the latents
under generic assumptions, and identification of the observed graph.
The proposed method CaDRe addresses both aspects through a variational autoencoder structure. Extensive experiments evaluate CaDRe against constraint-based baselines and CRL methods, also covering real-world climate data.

**Strengths:**

The presentation is strong, particularly due to the inclusion of illustrative examples and intuitive proof sketches.  The theoretical results are sufficiently novel in addressing a combination of CRL and causal discovery, where, unlike in CRL settings, the results allow addressing noisy observations.
The correspondence between non-linear SEMs and ICA in particular is interesting.
Finally, the experiments include real-world applications with convincing results, such as the wind system.
Overall, as also motivated well in the paper, the ability to perform both CRL and causal discovery fills a gap in the time series literature.

**Weaknesses:**

A potential limitation could be that the approach is limited to $d_z<5$ latent variables and was not evaluated beyond this, as well as the sensitivity of the approach to the assumptions, specifically
Assumptions 2 and 3 which assume enough distributional
variability and latent drift (essentially, heteroskedasticity). While the ablation study
on A3 is appreciated, it shows that the method's performance deteriorates when this assumption is unmet, and Assumption 2 is not commented on in detail. These aspects could be made more prominent in the main paper as a potential limitation.

**Questions:**

- Can you further comment on the meaning of Assumption 2 and whether it is likely to be met in real-world applications?
- For the experiment on the wind data, Fig. 5 shows the results for the specific months of 1970. Did you obtain consistent results across the entire time series (especially with respect to the reconstructed latent variables)? It would be helpful to include the estimated causal graph (middle plane) in unrotated form in the appendix, as the details are currently hard to read.
- Is there a particular reason why a latent variable dimension $d_z$ above 4 is not included in the evaluations? Is this a scalability issue, or do results deteriorate for more latent variables?
- The causal discovery baselines are limited to constraint-based CD. Did you also test a score-based baseline?
- Regarding typos, in ln. 1685, is one of $d_z$ instead $d_x$? In Fig. A7, is CARD your method CaDRe? In ln. 1714, a table reference is missing.

---

> ### Author Response · Authors · 2025-11-21
> **Author Response (1/3)**
>
> Dear Reviewer 3HPk, thank you for your insightful feedback. Your comments helped improve the rigor of our work—particularly in notation, terminology, comparisons, and quantitative analysis. We sincerely appreciate your time and effort and have updated the paper and appendix accordingly.
>
> We provide point-by-point responses to your comments below and have updated the manuscript accordingly.
>
> > W1 (1): The method focuses only on latent-variable settings and was not evaluated beyond $d_z = 5$.
>
> Thanks for raising this point. In light of your suggestion, we have provided the additional experiment on $d_z \in \{ 3, 5, 7, 9, 12 \}$ in our revised manuscript Table A10 and below:
>
> | $d_z$ | $d_x$  | SHD $(J_{\hat{g}}(\hat{x}_t))$ | TPR             | Precision       | MCC $(s_t)$        | MCC $(z_t)$        | SHD $(J_r(\hat{z}_t))$ | SHD $(J_d(\hat{z}_{t-1}))$ | $R^2$            | Inference Time (ms) |
> |:-----:|:------:|:-----------------------------:|:---------------:|:----------------:|:------------------:|:------------------:|:----------------------:|:---------------------------:|:----------------:|:--------------------:|
> | 3     | 10     | $0.43 \pm 0.05$               | $0.65 \pm 0.08$ | $0.63 \pm 0.14$  | $0.8504 \pm 0.07$  | $0.9652 \pm 0.02$  | $0.29 \pm 0.04$         | $0.40 \pm 0.05$              | $0.92 \pm 0.02$  | $0.71 \pm 0.03$ |
> | 5     | 10     | $0.44 \pm 0.04$               | $0.63 \pm 0.09$ | $0.61 \pm 0.12$  | $0.8221 \pm 0.06$  | $0.9610 \pm 0.03$  | $0.27 \pm 0.05$         | $0.37 \pm 0.02$              | $0.90 \pm 0.02$  | $0.88 \pm 0.10$      |
> | 7     | 10     | $0.46 \pm 0.06$               | $0.60 \pm 0.02$ | $0.59 \pm 0.01$  | $0.8936 \pm 0.05$  | $0.9550 \pm 0.04$  | $0.27 \pm 0.03$         | $0.30 \pm 0.11$              | $0.96 \pm 0.03$  | $1.05 \pm 0.20$      |
> | 9     | 10     | $0.44 \pm 0.05$               | $0.67 \pm 0.06$ | $0.57 \pm 0.11$  | $0.8082 \pm 0.04$  | $0.8480 \pm 0.05$  | $0.32 \pm 0.04$         | $0.41 \pm 0.08$              | $0.88 \pm 0.03$  | $1.27 \pm 0.18$      |
> | 12    | 10     | $0.47 \pm 0.05$               | $0.61 \pm 0.07$ | $0.55 \pm 0.09$  | $0.7020 \pm 0.06$  | $0.6800 \pm 0.05$  | $0.50 \pm 0.13$         | $0.58 \pm 0.19$              | $0.80 \pm 0.03$  | $1.34 \pm 0.12$      |
>
> As shown in the table above, experimental results on $d_z \geq 5$ verify the effectiveness of our method.
>
> > W1 (2): Sensitivity to assumptions (A2: distributional variability, A3: latent drift); Performance drops when these assumptions fail.
>
> As shown in Table A13 and below, we explicitly evaluated the model’s sensitivity to violations of A2, A3 and A5. Even when assumptions A2, A3, or A5 are violated, as you mentioned, conditions expected to degrade performance, CaDRe still shows relatively strong accuracy in both metrics of CRL and CD, indicating robustness and stability beyond ideal identifiability settings.
>
> **Table: Ablation Study on Assumption Violation**
>
> | **Assumption** | $d_ z$ | $d_ x$ | **SHD ($\mathbf{J}_ {\hat{g}}(\hat{\mathbf{x}}_ t)$) ↓** | **TPR ↑** | **Precision ↑** | **MCC($\mathbf{s}_ t$) ↑** | **MCC($\mathbf{z}_ t$) ↑** | **SHD ($\mathbf{J}_ r(\hat{\mathbf{z}}_ {t})$) ↓** | **SHD ($\mathbf{J}_ d(\hat{\mathbf{z}}_ {t-1})$) ↓** | **$R^2$ ↑** |
> |:----------------|:------:|:------:|:--------------------------------:|:----------:|:----------------:|:------------------:|:------------------:|:--------------------------------:|:----------------------------------:|:----------:|
> | **No Violation** | 3 | 6 | 0.18±0.06 | 0.83±0.03 | 0.80±0.04 | 0.9583±0.02 | 0.9505±0.02 | 0.24±0.19 | 0.33±0.09 | 0.92±0.01 |
> | **Violate A2 (Contextual Variability)** | 3 | 6 | 0.26±0.07 | 0.71±0.05 | 0.68±0.05 | 0.7563±0.04 | 0.8820±0.04 | 0.36±0.07 | 0.41±0.08 | 0.67±0.03 |
> | **Violate A3 (Latent Drift)** | 3 | 6 | 0.31±0.05 | 0.67±0.13 | 0.64±0.06 | 0.8645±0.07 | 0.8478±0.08 | 0.39±0.12 | 0.46±0.14 | 0.78±0.21 |
> | **Violate A5 (Generation Variability)** | 3 | 6 | 0.35±0.11 | 0.65±0.12 | 0.60±0.10 | 0.7052±0.13 | 0.9325±0.03 | 0.41±0.08 | 0.47±0.10 | 0.85±0.02 |
>
> Notably, as illustrated in our violation experiments (Line 1737-1746), such assumption violations occur only in highly constrained or unrealistic scenarios (e.g., linear additive models), which are unlikely to arise in real-world applications.

---

> ### Author Response · Authors · 2025-11-21
> **Author Response (2/3)**
>
> > W1 (3): Assumption 2 is not commented on in detail
>
> We have already provided the discussions on A2 in Appendix Line 1065-1079. In light of your comment, now we have highlighted the reference to appendix in the main paper, and show the brief version below:
>
> "
> The injectivity of the operator (A2) allows taking inverses of certain operators, a common assumption in nonparametric identification. In the context of the climate system, this represents the necessity of temporal variability. Below are some illustrative examples in terms of mappings $p_a \Rightarrow p_b$:
>
> **Example 1.** $b = g(a)$, where $g$ is an invertible function.
>
> **Example 2.** $b = a + \epsilon$, where $p(\epsilon)$ must not vanish everywhere after the Fourier transform.
>
> **Example 3.** $b = g(a) + \epsilon$, where the same conditions from Examples 1 and 2 apply.
>
> **Example 4.** $b = g_1(g_2(a) + \epsilon)$, a post-nonlinear model with invertible nonlinear functions $g_1$ and $g_2$, combining the assumptions in Examples 1–3.
>
> **Example 5.** $b = g(a, \epsilon)$, where the joint distribution $p(a, b)$ follows an exponential family.
>
> **Example 6.** $b = g(a, \epsilon)$, a general nonlinear formulation. Certain deviations from the nonlinear additive model (Example 3), such as polynomial perturbations, can still be tractable.
> "
>
> > Q1: Can you further comment on the meaning of Assumption 2 and whether it is likely to be met in real-world applications?
>
> Thank you for this insightful question. A2 means a distribution-level variability, where the different input distribution correpsonds different outout distribution. Details could be found in our response to W1(3).
>
> It is likely to be met in real-world climate data because climate is a dynmaic system which include rich variability, and we have verified it in our revised mansucript, Appendix E.2, Lines 1942–1976 and Figure A11 + A12. The results show that different $p(x_{t+1})$ corresponded to different $p(x_{t-1})$, and different $p(z_{t})$ corresponded to different $p(x_{t})$.
>
> >Q2 (1): For the experiment on the wind data, Fig. 5 shows the results for the specific months of 1970. Did you obtain consistent results across the entire time series (especially with respect to the reconstructed latent variables)?
>
> Yes, you are tottally correct, the causal graphs are consistent across the entire time series dataset. We would like to note that the weights of Jacobian matrix can vary across time steps.
>
> >Q2 (2): It would be helpful to include the estimated causal graph (middle plane) in unrotated form in the appendix, as the details are currently hard to read.
>
> That is a very good suggestion. We have included it in the Appendix Table A10, and added the reference in main paper.
>
> > Q3 (1): Is there a particular reason why a latent variable dimension above 4 is not included in the evaluations?
>
> Thank you for raising this important point. In light of your comment, we have included experiment results with higher $d_z$, please see our response to **W1**.
>
> > Q3 (2): Is this a scalability issue, or do results deteriorate for more latent variables?
>
> We thank reviewer for raising this point. Currently, it is a scalability issue in most of causal structure learning with continuous optimization. Due to non-convexity in structure learning [1], results would deteriorate along with $d_z$ increasing, and addressing it is our future work. One promising direction is to incorporate physical priors into the latent variables, e.g., known climate variables or structures, which could help mitigate the degradation in higher-dimensional settings.
>
> [1] Ng, Ignavier, Biwei Huang, and Kun Zhang. "Structure learning with continuous optimization: A sober look and beyond."

---

> ### Author Response · Authors · 2025-11-21
> **Author Response (3/3)**
>
> > Q4:  The causal discovery baselines are limited to constraint-based CD. Did you also test a score-based baseline?
>
> We thank the reviewer for the helpful suggestion. In the revised manuscript, we have included score-based causal discovery baselines, specifically the BIC method [2] and the Generalized Score Function [3]. These experiments were conducted using the causal-learn [4], and the results are now reported in Table A15 of the updated version and below:
>
> | Method                     | $d_ x$ | SHD↓  | Precision↑ | Recall↑ | F1↑   |
> |-----------------------------|-----|-------|-------------|---------|-------|
> | **CaDRe**                   | 3   | 0.000 | 1.000       | 1.000   | 1.000 |
> |                             | 6   | 0.185 | 0.803       | 0.830   | 0.816 |
> |                             | 8   | 0.295 | 0.761       | 0.789   | 0.775 |
> |                             | 10  | 0.432 | 0.638       | 0.656   | 0.647 |
> | **FCI**                     | 3   | 0.186 | 0.801       | 0.760   | 0.780 |
> |                             | 6   | 0.384 | 0.476       | 0.394   | 0.431 |
> |                             | 8   | 0.447 | 0.398       | 0.321   | 0.356 |
> |                             | 10  | 0.492 | 0.355       | 0.284   | 0.316 |
> | **CDNOD**                   | 3   | 0.163 | 0.821       | 0.782   | 0.801 |
> |                             | 6   | 0.452 | 0.432       | 0.419   | 0.426 |
> |                             | 8   | 0.509 | 0.365       | 0.312   | 0.336 |
> |                             | 10  | 0.546 | 0.328       | 0.276   | 0.300 |
> | **PCMCI**                   | 3   | 0.139 | 0.843       | 0.803   | 0.823 |
> |                             | 6   | 0.431 | 0.488       | 0.386   | 0.432 |
> |                             | 8   | 0.501 | 0.397       | 0.308   | 0.347 |
> |                             | 10  | 0.548 | 0.365       | 0.284   | 0.320 |
> | **LPCMCI**                  | 3   | 0.116 | 0.864       | 0.823   | 0.843 |
> |                             | 6   | 0.337 | 0.637       | 0.621   | 0.629 |
> |                             | 8   | 0.441 | 0.535       | 0.486   | 0.509 |
> |                             | 10  | 0.487 | 0.482       | 0.432   | 0.456 |
> | **BIC**                     | 3   | 0.192 | 0.755       | 0.712   | 0.732 |
> |                             | 6   | 0.349 | 0.602       | 0.574   | 0.588 |
> |                             | 8   | 0.456 | 0.588       | 0.455   | 0.513 |
> |                             | 10  | 0.493 | 0.408       | 0.406   | 0.406 |
> | **Generalized Score Function** | 3   | 0.108 | 0.876       | 0.838   | 0.857 |
> |                             | 6   | 0.298 | 0.692       | 0.659   | 0.675 |
> |                             | 8   | 0.382 | 0.604       | 0.563   | 0.583 |
> |                             | 10  | 0.402 | 0.512       | 0.442   | 0.474 |
>
> The results show that CaDRe consistently outperforms all score-based and constraint-based baselines.
>
> [2] Schwarz, Gideon. "Estimating the dimension of a model." The annals of statistics (1978): 461-464.
>
> [3] Huang, B., Zhang, K., Lin, Y., Schölkopf, B., & Glymour, C. (2018, July). Generalized score functions for causal discovery. In Proceedings of the 24th ACM SIGKDD International Conference on Knowledge Discovery & Data Mining (pp. 1551-1560).
>
> [4] Zheng, Yujia, et al. "Causal-learn: Causal discovery in python." Journal of Machine Learning Research 25.60 (2024): 1-8.
>
> > Q5: Typos, $d_z$ in ln. 1685; Fig. A7; table reference in ln. 1714
>
> We thank the reviewer for the careful reading and detailed pointers, which help us improve the presentation of our paper. The typo in line 1685 has been corrected in the revised version. In Figure A7 (now A9 in revision), “NCDL” refers to our method (CaDRe) before the final naming change, which has now been updated. The table reference in line 1714 has also been corrected to properly cite the Table A17, which includes the comparisons among large-scale weather forecasting models.

---

### Official Review · Reviewer_5w99 · 2025-11-01

**Soundness:** 4
**Presentation:** 3
**Contribution:** 3
**Rating:** 8
**Confidence:** 3

**Summary:**

The paper introduces CaDRe, a causal representation learning framework that aims to recover both observable-to-observable causal structure and latent drivers from time-lagged climate time series. This method uses latent state-space VAE–style model with structural constraints to simultaneously model the data and identify causal relationships. This paper also presents several new theoretical results on causal discovery that establish conditions under which the two causal structures are simultaneously identifiable and motivate the VAE based approach. Empirical results on synthetic data and real climate model data show that CaDRe is able to identify scientifically plausible causal effects through a wind map analysis. The method is also evaluated on short-term forecasting tasks, where it is shown that identification of causal structures improves forecasting here.

**Strengths:**

1. Strong theoretical results. Clear identifiability theorems/lemmas that theoretically demonstrate the recoverability of both latent drivers and observable causal edges. The ability to simultaneously identify latent and observable effects is very important in climate modeling studies, where many latent processes can drive global climate patterns. Furthermore, these results directly lead to a new state-space generative model, with structural constraints, derived from the theory.

2. Comprehensive evaluations.. Synthetic and real climate datasets demonstrate that this approach works to recover causal structures in a wide range of settings. External checks that related the recovered structure to physical patterns (e.g. wind-direction maps) was an interesting surrogate for the true causal structure, which is unknown.

3. Relevance: Latent confounding and time-lagged dependencies are central challenges in climate analysis; bringing identifiability guarantees to this setting is valuable and future works can build upon this approach (both the VAE method and the theory).

4. Generally clear and well structured. The paper is well positioned relative to constraint-based time-series causal discovery (e.g., PCMCI) and latent-confounder settings, and it explains why a generative approach is helpful here. Mathematics are presented very rigorously.

**Weaknesses:**

1. Assumptions should to be mapped to climate data to support general claims. The main causal assumptions (e.g., forms of sparsity/locality, faithfulness/functional-faithfulness, injectivity/regularity conditions used in the proofs) are all very reasonable in an abstract sense, but its possible that they do not hold in climate data. The paper would be stronger with brief, data-driven diagnostics (or sensitivity analyses already in the appendix moved forward) showing these assumptions are approximately satisfied, or at least not grossly violated, on the real datasets used.

2. Role of forecasting vs identification. Short lead forecasting results are presented as evidence of practical value, but the paper’s primary contribution is causal identification in climate models. Please clarify the conceptual link: do better graph-recovery metrics predict better forecast skill, or is forecasting simply a secondary demonstration? Without an explicit link, the forecasting section risks diluting the identification story, especially since short lead forecasting is not feature of climate models that are more concerned with multi-decade scale distributions.

3. The paper introduces an additional variable (e.g., s / “s-encoder” noise or context term) with an important role in the generative model and technical details. Its definition and interpretation arrive late and somewhat tersely. Because s impacts identifiability conditions and estimation, it should be defined earlier in plain language, with a short discussion of plausibility in climate data.

4. Scalability and dimensional effects. The structure-recovery metrics degrade as the number of observed variables grows (Table 2), and real climate data (e.g. CMIP models and reanalysis data) often have dimensions in the hundreds or more. Does this method scale to high resolution data and, if not, are there any constraints (e.g., masks/regularization) that might help it perform in higher dimensions?

5. Teleconnections. Given that teleconnections are one of the most well known and well studied causal-like structures in climate, it would be very interesting to see how well this model does in recovering known teleconnections (e.g. ENSO related ones).

**Questions:**

1. Are there any standard / simple diagnostic checks that could be performed on the real climate data to show that the causal assumptions on not grossly violated? Given the sensitivity results in Table A11, these kinds of checks (even heuristic checks) would be helpful for ensuring the appropriateness of the model.

2. How well does this approach scale when the dimension continuous to increase? I can see that the F1 score and other metrics degrade with dimension, but I'm curious if these plateau or if they converge to zero with increasing dimension; and also how increasing dimension effects the run time. For example, PCMCI can become quite slow at higher dimensions depending on the conditional information tests used. Would this method be applicable to something like ERA5 reanalysis data without significant coarsening?

3. The real-data validation uses wind-field alignment and qualitative climate consistency. Have you evaluated whether CaDRe recovers canonical teleconnections (e.g., ENSO-related patterns) on SST/pressure fields? These results could be another very useful external validity check.

4. The introduction states that latent drivers ​“such as pressure and precipitation,” are not directly measured. Since both pressure and precipitation are widely observed, did you mean that their driving processes are latent, or that the relevant vertical-level/large-scale drivers are unobserved?

---

> ### Author Response · Authors · 2025-11-21
> **Author Response (1/2)**
>
> Dear reviewer 5w99, we are very grateful for your valuable comments, helpful suggestions, and encouragement. We provide the point-to-point response to your comments below and have updated the paper and appendix accordingly.
>
> > W1 & Q1: Need to connect theoretical assumptions (sparsity, faithfulness, injectivity) to climate data; suggest empirical checks or diagnostics to show they approximately hold.
>
> We thanks reviewer for raising this important point. As illustrated in our paper, the assumptions of sparsity, faithfulness, and injectivity are well aligned with the physical nature of climate systems:
>
> (i) **Sparsity** reflects that only a limited number of physical processes, such as local radiative forcing, ocean–atmosphere coupling, and regional convection, interact within each time scale; hence, most potential causal links are inactive or very minor.
> (ii) **Faithfulness** corresponds to the structural minimality of climate dynamics, which means that observable variables (e.g., temperature, humidity, and wind) maintain only essential dependencies consistent with physical energy and mass conservation laws, preventing redundant causal paths.
> (iii) **Injectivity** arises naturally from temporal variability: distinct latent climatic states (e.g., circulation regimes or ENSO phases) produce distinguishable distributions of observables such as surface temperature or precipitation.
> These properties are empirically supported in our datasets, where the recovered causal graphs reproduce known spatial dependencies and circulation patterns (please see Fig. A6), confirming that our theoretical assumptions are realistic for climate analysis.
>
> We fully agree that we need empirical checks or diagnostics to show they approximately hold. However, it is difficult to verify directly in real-world data, since latent variables are **inherently unobserved** and then cannot be explicitly tested. But these assumption can be tested indirectly: Under these assumptions, we can first estimate the model, and then we can test whether these assumptions hold true in real-world climate data. . The results, shown in Appendix E.2, Lines 1943–1976 and Figure A11 + A12 of the revised manuscript, demonstrate that both A2 and A3 hold approximately in real-world climate data.
>
> > W2: Role of forecasting vs identification. Do better graph-recovery metrics predict better forecast skill, or is forecasting simply a secondary demonstration?
>
> We thank the reviewer for this insightful question.
>
> (1) In theory, the model graph-recovery metric provide a compact model to explain the data, and hence, the prediction is likely to be better. Specifically, accurate causal graphs introduce useful inductive biases that improve forecasting efficiency and reduce uncertainty by preserving physically meaningful dependencies [1, 2].
>
> (2) Better latent space recovery directly improves forecasting performance. Intuitively, $x_t$ is predicted from $p(x_t \mid z_t)$. If $z_t$ is well approximated by $\hat z_t$ up to an invertible mapping: $z_t = h(\hat{z}_t)$ (as shown in Theorem 1), then $p(x_t \mid z_t) = p(x_t \mid h(z_t))$, which minimizes reconstruction risk and enhances predictive accuracy.
>
> In light of your suggestion, we have added this clarification and supporting discussion in the revised manuscript Appendix E.1, Line 1932-1940.
>
> [1] Iglesias‐Suarez, Fernando, et al. "Causally‐informed deep learning to improve climate models and projections." Journal of Geophysical Research: Atmospheres 129.4 (2024): e2023JD039202.
>
> [2] Runge, Jakob, et al. "Inferring causation from time series in Earth system sciences." Nature communications 10.1 (2019): 2553.
>
> > W3: s should be defined earlier in plain language, with a short discussion of plausibility in climate data.
>
> We thanks for this great suggestion. In light of insightful suggestion, before the mathematical definition of $s$ in Eq. (1), we add a notation Table 1 with a plain language definition on $s$ along with a discussion on its role in climate data in the Line 120-122 of our revised version and below:
>
> "
> *The endogenous mediator or nonstationary noise $s_ {t,i}$, depending on latent driving forces $\mathbf{z}_ t$, is designed to capture dynamic uncertainties, e.g., perturbations introduced by CO$_ 2$ [3].*
> "
>
> [3] Stips, Adolf, et al. "On the causal structure between CO2 and global temperature." Scientific reports 6.1 (2016): 21691.

---

> ### Author Response · Authors · 2025-11-21
> **Author Response (2/2)**
>
> > W4 (1): Scalability and dimensional effects. The structure-recovery metrics degrade as the number of observed variables grows (Table 2), and real climate data (e.g. CMIP models and reanalysis data) often have dimensions in the hundreds or more.
>
> Thanks for raising this good point. Currently, it is a scalability issue in most of causal structure learning with continuous optimization. Due to non-convexity in structure learning [4], results would deteriorate along with dimension increasing, and addressing it is our future work. One promising direction is to incorporate physical priors into the variables, e.g., known climate variables or structures, which could help mitigate the degradation in higher-dimensional settings.
>
> While high-dimension graph-recovery requires more physical prior information, we can show that CaDRe's compatibility with high-dimensional CMIP data forecasting, in light of your comment. We obtain this data from (https://pcmdi.llnl.gov/mips/cmip5/data-access-getting-started.html#:~:text=1) and use the downscaled grid of time * 30 (lat) * 60 (lon), which constitutes a dataset with thousands of dimensions:
>
> | Metric | CaDRe (Ours) MSE | CaDRe (Ours) MAE | CARD MSE | CARD MAE | MICN MSE | MICN MAE | iTransformer MSE | iTransformer MAE | TimesNet MSE | TimesNet MAE | Autoformer MSE | Autoformer MAE |
> |---------|------------------|------------------|-----------|-----------|-----------|-----------|------------------|------------------|---------------|---------------|----------------|----------------|
> | 96   | **0.048** | **0.163** | 0.051 | 0.168 | 0.065 | 0.187 | 0.207 | 0.359 | 0.068 | 0.197 | 0.095 | 0.238 |
> | 192   | **0.058** | **0.179** | 0.058 | 0.179 | 0.088 | 0.217 | 0.215 | 0.369 | 0.076 | 0.208 | 1.021 | 0.821 |
> | 336   | 0.067 | **0.195** | **0.067** | 0.194 | 0.112 | 0.247 | 0.221 | 0.374 | 0.075 | 0.206 | 1.095 | 0.854 |
> | 720   | 0.081 | **0.206** | **0.081** | 0.209 | 0.120 | 0.249 | 0.221 | 0.371 | 0.091 | 0.223 | 1.106 | 0.860 |
>
> These results indicate that CaDRe converges well and reconstructs large-scale observational structures effectively, thereby maintaining identifiability even in high-dimensional settings. We believe that further integrating physical priors and improving optimization to avoid local minima will strengthen graph-recovery performance of CaDRe in future work.
>
> [4] Ng, Ignavier, Biwei Huang, and Kun Zhang. "Structure learning with continuous optimization: A sober look and beyond."
>
> > W4 (2): Does this method scale to high resolution data and, if not, are there any constraints (e.g., masks/regularization) that might help it perform in higher dimensions?
>
> You are totally right! In Table A9, we have already shown that CaDRe can indeed scale to high-resolution data by introducing a mask based on physical distance, which constrains the search space of potential causal connections,
>
> In Line 1589-1594, we introduce the masking strategy based on physical distance prior, which effectively maintains performance in higher-dimensional settings while improving computational efficiency. We have highlighted this part and the reference to appendix in main paper in our revised manuscript.
>
> > W5: Teleconnections. Given that teleconnections are one of the most well known and well studied causal-like structures in climate, it would be very interesting to see how well this model does in recovering known teleconnections (e.g. ENSO related ones).
>
> We thank the reviewer for this excellent suggestion, which helps strengthen the climate relevance of our work. In light of this, we have added an experiment in the revised manuscript evaluating CaDRe’s ability to recover teleconnections in ENSO-related datasets. We retrieved monthly anomalies from the NOAA PSL website (https://psl.noaa.gov/data/timeseries/month/) in standard-format text, for the period 1950-1960, for the Niño 1+2, Niño 3, Niño 3.4, Niño 4 sea-surface-temperature anomalies and the Southern Oscillation Index (SOI).
>
> | **Niño4** | → | **Niño3.4** | → | **Niño3** | → | **Niño1+2** |
> |:----------:|:-:|:------------:|:-:|:----------:|:-:|:------------:|
> |           |  ↘ |            |  ↙ |            |   |              |
> |    |  |   **SOI**    |   |            |   |              |
>
> The results show that the model successfully captures known large-scale patterns, such as ENSO-driven interactions across the Pacific, which is nearly aligned with scentific evidences, demonstrating CaDRe's effectiveness in identifying physically meaningful teleconnection structures.

---

### Official Review · Reviewer_bJxc · 2025-11-01

**Soundness:** 3
**Presentation:** 2
**Contribution:** 3
**Rating:** 4
**Confidence:** 4

**Summary:**

The paper proposed a causal discovery method for time series data to learn causal structures among the observed variables and latent factors. Through the mathematical explanation, the authors draw a connection between nonlinear ICA and nonlinear SEM to find the causal connections in the presence of hidden processes. A Variational Autoencoder is utilized in the proposed architecture to learn the latent factors and future value prediction. The paper also presented experimental analysis using synthetic and real-world datasets for both causal discovery and latent factor reconstruction.

**Strengths:**

S1: Addresses learning causal graph for both observed variables and latent factors—an area where constraint-based methods often struggle and many score-based approaches impose strong assumptions.

S2: Integrated causal representation learning and causal discovery together to handle complex real-world data distribution.

S3: Used mathematical explanations and proofs to establish an injective relation between latent factors and observed variables to ensure identifiability.

S4: Comprehensive evaluations using synthetic and climate datasets with many baselines.

**Weaknesses:**

W1: The overall flow of the paper is difficult to follow. The mathematical explanations may be presented in a more connected way.

W2: In the “Structure Learning” block of section 4 mentioned, the causal graph of observed variables is generated from the decoder. However, it is not clearly explained here how and from which parameters it was computed. Also, how the causal relationships are imposed in one dense layer requires more explanation.

W3: The Jacobian matrices approximated using the normalization flow model generally represent the correlation between the data distribution. How does KL divergence loss enforce causality in this transformation process?

W4: In the proposed architecture, using the same design, how can the Z-encoder and s-encoder learn different features is not clear? Because in the decoder, these two features are used for reconstruction.

W5: How does the proposed model maintain an injective relation between the observed variables and latent factors using the simple design with only dense layers?

**Questions:**

Q1: Lemma 1 mentioned that value level mapping from x to z is not possible, so the identifiability is formulated at the distribution level. But, at line 172, it is claimed that the proposed model achieved identifiability at the value level with more information. Which is correct in this mathematical explanation?

Q2: The nonlinear transformation operator L is used to transform Px(t+1) to Px(t-1) at line number 138. Why do we need to transform in the opposite temporal order?

Q3: For the successful causal discovery of latent factors, it is important to define the correct number of latent factors. How to calculate the correct number of latent factors for the proposed model?

Q4: Line 318 mentioned that the conditional independence between the estimated z and s is ensured through the KL divergence on the error distribution. The data generation process mentioned in equation 1 shows that s is dependent on z. Then how does the proposed model find conditional independence here?

Also consider the points mentioned in the weakness.

---

> ### Author Response · Authors · 2025-11-21
> **Author Response (1/3)**
>
> Dear Reviewer bJxc, thank you for your constructive comments. Your insights have helped us significantly improve the clarity, empirical validation, and methodology framing of our work. We have updated the paper and appendix accordingly. Below, please see our point-to-point responses.
>
> > W1: The paper’s overall flow is hard to follow; mathematical explanations could be more connected
>
> We thank the reviewer for raising this point. For the rigor of the theoretical results for CRL/CD, this paper includes some mathematical content, although we try to improve the readability for avoiding heavy notations and many derivations in this paper. Hence, in light of your great suggestion, we have included an **high-level description of mathematical results** at the beginning of Section 3 and added **intuitive proof sketches** for Theorems 1-3, along with two **proof pipelines** connecting the main mathematical steps, and illustrative examples for better understanding. We hope you agree that, despite this, the paper is well-organized with smooth presentation, *as given in the comments by Reviewer 5w99 S4 and 3HPk S1*.
>
> At a high-level, Theorem 1 establishes latent space recovery through nonparametric identification, Theorem 2 demonstrates the functional equivalence between SEM and ICA, and Theorem 3 builds upon these results—first applying Theorem 1 to identify a nonlinear ICA model and then using Theorem 2’s equivalence to identify the corresponding nonlinear SEM. We believe these revisions make the theoretical flow clearer and more cohesive.
>
> > W2 (1):  In the “Structure Learning” block of section 4 mentioned, the causal graph of observed variables is generated from the decoder. However, it is not clearly explained here how and from which parameters it was computed.
>
> Thank you for the comment. We would like to highlight that, the causal graph is **not directly generated from decoder**, instead, it is derived from the **Jacobian matrix of decoder** as explained in Line 354. Specifically, for the data-generating process $x_ {t,i} = g_ i(\mathbf{pa}_ O(x_ {t,i}), \mathbf{pa}_ L(x_ {t,i}), s_ {t,i})$, the partial derivative $[J_ g(x_ t)]_ {i,j} = \frac{\partial x_ {t,i}}{\partial x_ {t,j}}$ represents the causal influence of $x_ {t,j}$ on $x_ {t,i}$, which we cannot obtain it directly from decoder. Instead, the partial derivative $[J_ m(s_ t)]_ {i,j} = \frac{\partial x_ {t,i}}{\partial s_ {t,j}}$ can be obtained from the decoder, which captures the derivative of each output variable $x_ t$ w.r.t. inputs $s_ t$.
>
> **How to compute the causal graph:** Following the Theorem 2 (Functional Equivalence), we transform this Jacobian through $J_g(x_t) = I_d - D_m(s_t) J_m^{-1}(s_t)$, where $J_m(s_t)$ is obtained from the Jacobian matrix of decoder, and $D_m(s_t)$ is the diagonal matrix of $J_m(s_t)$. Ultimately, $J_g(x_t)$ reflects the causal adjacency matrix, which unveils the causal graph.
>
> If you don't think it is clear enough, please kindly let use know.
>
>
> > W2 (1): Also, how the causal relationships are imposed in one dense layer requires more explanation.
>
> We thank the reviewer for raising this point. In light of your comment, we have added one sentence in the revised manuscript Line 354-370 to highlight the following point:
>
> The causal relationships are **not manually imposed within a dense layer**; instead, they emerge through structural regularization applied to **the Jacobian matrices** computed from that layer. Specifically, after obtaining the Jacobian $J_g(x_t)$ from the decoder outputs, we impose two structural constraints:
> 1. Sparsity penalty $L_ s$ encourages each variable to depend only on a limited subset of others.
> 2. DAG constraint $L_ d$ [1] ensures acyclicity by penalizing deviations from a directed acyclic structure.
>
> These constraints are incorporated into the overall objective $L_ {ALL} = L_ {ELBO} + \alpha L_ s + \beta L_ d$, guiding the dense layer to represent causally ordered, sparse dependencies rather than arbitrary correlations. Here, the dense layer acts as a flexible function approximator, and its Jacobian matrix can be sparse.
>
> [1] Yu, Yue, et al. "DAG-GNN: DAG structure learning with graph neural networks." International conference on machine learning. PMLR, 2019.

---

> ### Author Response · Authors · 2025-11-21
> **Author Response (2/3)**
>
> > W3: The Jacobian matrices approximated using the normalization flow model generally represent the correlation between the data distribution. How does KL divergence loss enforce causality in this transformation process?
>
> Please kindly note that we **do not use the normalizing flow + KL divergence to estimate the causal graph** directly. Instead, the flow network is used to model the noise variables $(\epsilon^x_t, \epsilon^z_t)$, while the KL divergence term enforces their statistical independence.
>
> As described in Lines 309-352 of our submission, the KL loss minimizes the divergence between the estimated noise distributions and a standard normal $\mathcal{N}(0, \mathbf{I})$. This independence constraint and sparsity constraint ensures that each learned noise component corresponds to its parents, allowing the model to **infer the causal graph** inversely from these disentangled sources.
>
> This design follows established approaches in nonlinear causal discovery and causal representation learning, e.g., [2,3,4], where learning independent noise via KL regularization is key to recovering causal structure.
>
> Any feedback on this would be appreciated.
>
> [2] Zhang, Kun, and Aapo Hyvarinen. "On the identifiability of the post-nonlinear causal model." arXiv preprint arXiv:1205.2599 (2012).
>
> [3] Khemakhem, Ilyes, et al. "Variational autoencoders and nonlinear ica: A unifying framework." International conference on artificial intelligence and statistics. PMLR, 2020.
>
> [4] Brouillard, Philippe, et al. "Differentiable causal discovery from interventional data." Advances in Neural Information Processing Systems 33 (2020): 21865-21877.
>
> > W4: In the proposed architecture, using the same design, how can the Z-encoder and s-encoder learn different features is not clear? Because in the decoder, these two features are used for reconstruction.
>
> We would like to clarify that, although the z-encoder and s-encoder share a similar architectural form (both implemented as MLPs), these two encoders **do not share parameters**, and operate under different learning objectives and posterior/prior estimations, as illustrated in Lines 310-353. Thus, z-encoder and s-encoder can learn different features.
>
> > W5: The mechanism ensuring an injective mapping between observed variables and latent factors using only dense layers is uncertain.
>
> Thank you for raising this point. We agree with you that the only non-trivial condition is the injective linear operator. However, since the VAE is a probabilistic model with dense layers that explicitly capture the distribution-level stochastic transformation at each stage, the existence of a mapping from $p(x_t)$ to $p(z_t)$ is guaranteed. Moreover, the reconstruction process demonstrates that $p(x_t)$ is generated from $p(z_t)$, thereby modeling the **injective** behavior. Since the assumption is on the distribution level, no explicit constraints are imposed on the encoder and decoder.
>
> > Q1: Clarify the apparent contradiction between distribution-level and value-level identifiability in Lemma 1 and line 172.
>
> We thank the reviewer for the careful reading and insightful understanding of our theory, which help us refine our presentation.
>
> To clarify, when we state that identifiability is “formulated at the distribution level,” we refer to the intermediate result obtained at the first stage. This formulation serves as a foundation for the value-level identifiability, where, by introducing additional conditions. Compared with distribution-level identifiability, value-level identifiability includes more information (e.g., cross-derivative terms) that enable the subsequent component-wise identification (Theorem A3). We have made the presentation of this part clearer in our reivsed manuscript Line 159-161.
>
> In summary, there is no contradiction: the distribution-level identifiability is an intermediate step, while the value-level identifiability represents the final result, as shown in our **proof sketch of Theorem 1**.
>
> > Q2: The nonlinear transformation operator L is used to transform Px(t+1) to Px(t-1) at line number 138. Why do we need to transform in the opposite temporal order?
>
> $L_{x_{t+1}|x_{t-1}}$ is just a property of conditional probability distributions, **does not indicating there is a direct link between $x_{t+1}$ and $x_{t-1}$**. If two variables have computable distributions, the distribution-level relation objectively exists. We just need this mathematical term to establish the identifiability.

---

> ### Author Response · Authors · 2025-11-21
> **Author Response (3/3)**
>
> >  Q3: How to calculate the correct number of latent factors for the proposed model?
>
> Thank you for raising the practically important question.
>
> (1) Theoretically, we have demonstrated that the correct number of latent factors is identifiable within our framework, as detailed in our theoretical analysis (please see Appendix, Lines 1055–1062). For completeness, we restate the key reasoning below:
>
> To ensure the latent dimension $d_z$ is also identifiable, we analyze two scenarios:
>
> **Over Sufficient:** If $d_ {\hat{z}} > d_ z$: $d_ z$ latent components in $\hat{z}_ t$ are sufficient to explain $x_ t$, i.e.,
>     $$
>         p(x_ t \mid z_ {t,:d_ {\hat{z}} - d_ z}, z^{(1)}_ {t,d_{\hat{z}} - d_ z:}) = p(x_ t \mid z_ {t,:d_ {\hat{z}} - d_ z}, z^{(2)}_ {t,d_ {\hat{z}} - d_ z:}),
>     $$
>     which contradicts the Assumption A3.
>
> **Over Minimal:** If $d_{\hat{z}} < d_z$: This suggests that only $d_{\hat{z}}$ dimensions are sufficient to reconstruct $x_t$, leaving $d_z - d_{\hat{z}}$ components constant, which violates that there are $d_z$ latent variables.
>
> (2) Empirically, the identification of the true latent dimensionality becomes a **model selection** problem of finding a **minimal yet sufficient** number of latent variables. We treat the latent dimensionality as hyperparameters, determined through standard criteria such as predictive accuracy and reconstruction error. As shown in our simulated experiments, using fewer latent variables than the true number leads to reconstruction failure (over minimal), while using more yields redundant components but no significant improvement (over sufficent).
>
> **Table: Reconstruction performance under different estimated latent dimensions $\hat{d}_z$ with true $d_z = 4$**
>
> | True $d_z$ | Estimated $\hat{d}_z$ | MSE ↓         | MAE ↓         |
> |:-----------:|:---------------------:|:--------------:|:--------------:|
> | 4           | 2                     | 0.642 ± 0.056  | 0.273 ± 0.044  |
> | 4           | 3                     | 0.172 ± 0.032  | 0.231 ± 0.027  |
> | 4           | 4                     | **0.115 ± 0.018** | **0.193 ± 0.021** |
> | 4           | 5                     | 0.133 ± 0.019  | 0.198 ± 0.022  |
> | 4           | 6                     | 0.109 ± 0.021  | 0.248 ± 0.025  |
>
>
>
> >  Q4: Eq. (1) defines s as dependent on z, yet line 318 claims conditional independence between z and s via KL divergence—please reconcile this.
>
> Please kindly note that in Line 318 (original submission), we impose the conditional independence between the **estimated noise terms $\hat{\boldsymbol{\epsilon}}^x_t$ and $\hat{\boldsymbol{\epsilon}}^z_t$**, rather than between $z$ and $s$.

---

### Official Review · Reviewer_tKCX · 2025-11-01

**Soundness:** 3
**Presentation:** 3
**Contribution:** 3
**Rating:** 6
**Confidence:** 2

**Summary:**

This work proposes a unifying framework for causal relations among observable variables, often overlooked in causal representation learning, along with latent factors. The authors establish conditions under which the latent processes and the causal structure among observed variables are simultaneously identifiable from time-series data in climate analysis. The proposed method recovers causal graphs that align with domain experts.

**Strengths:**

Strengths:

1. The paper makes a significant theoretical advancement by establishing identifiability conditions for jointly recovering latent variables, latent causal processes, and causal graphs over observed variables in the presence of noise and latent confounders.

2. The framework directly addresses real limitations in climate modeling where both latent atmospheric processes (e.g., pressure systems, precipitation) and spatial dependencies among measurements are critical.

**Weaknesses:**

Weaknesses:

1. The paper provides no interpretation of what latent variables or causal relationships represent in the considered domains, such as Traffic data.

2. Lemma 1's proof (A.8) assumes $J_{m}(s_t)$ invertibility, which Corollary 2.1 derives from the DAG assumption, creating circular logic: DAG -> $J_m$ invertible ->  latent recovery ->  DAG learning (Theorem 3). The framework claims to jointly identify latent variables and the DAG structure, but the proof order suggests that the DAG structure must be known a priori for identification to work, contradicting the joint discovery claim and leaving unclear what is actually being assumed versus what is being learned.

3. The notation is dense in places; a consolidated notation table earlier in the main paper (not just the appendix) would improve readability.

4. Assumptions mentioned are difficult to verify in real world, for example latent drift assumtion (A3) in Theorem 1.

**Questions:**

Please answer questions in the Weakness section.

---

> ### Author Response · Authors · 2025-11-21
> **Author Response (1/2)**
>
> Dear Reviewer tKCX,
>
> We thank Reviewer tKCX for the time and effort devoted to evaluating our submission, and we appreciate the reviewer’s constructive comments and thoughtful suggestions. We have carefully addressed each concern below and have updated the paper and appendix accordingly.
>
> > W1: The paper provides no interpretation of what latent variables or causal relationships represent in the considered domains, such as Traffic data.
>
> Thank you for raising these valuable points, which help us improve the interpretability and soundness of CaDRe when extended to diverse domains. We have clarified the meanings of latent variables and causal relations across all considered datasets in the revised manuscript (Table A19, Lines 1870–1886), summarized as follows:
>
> | Dataset | Meaning of Latent Variables | Meaning of Causal Relations |
> |----------|-----------------------------|------------------------------------------------------|
> | **ILI (Public Health)** | Latents capture hidden epidemic dynamics such as <u>*viral transmission strength, seasonality, and social mobility trends*</u> not directly observed in infection counts [1]. | Causal edges represent short-term influences between <u>*regional infection rates*</u> (e.g., spatial spread or temporal cross-region effects) [1]. |
> | **ECL (Electricity Consumption)** | Latents encode underlying energy demand drivers—<u>temperature, population activity patterns, and industrial load cycles</u>—that evolve over time [2]. | Causal links describe dependencies between <u>regional electricity stations or customer groups</u> (e.g., peak-demand propagation) [2]. |
> | **Traffic (Road Monitoring)** | Latents represent hidden mobility factors such as <u>congestion propagation speed, commuting intensity, and network bottleneck dynamics</u> [3]. | Causal edges correspond to directional <u>traffic flow</u> influence between neighboring sensors (e.g., upstream → downstream road segments) [3]. |
>
> We also show our causal discovery results on the traffic dataset in our revised manuscript, Figure A7 and below:
>
>
> | Weather    | →        | Events   | → | Traffic Volume            |
> |----------------|:--------:|:--------:|:---------------:|:---------------:|
> |     |  ↘  |  |      ↗      |
> |  |    |  →  |               |
>
> with *Estimated Jacobian Matrix*:
>
> |                | Weather | Events  | Traffic Volume |
> |----------------|:--------:|:--------:|:---------------:|
> | **Weather**    | 0        | 0.2323   | 0.4551          |
> | **Events**     | 1.4345   | 0        | 0.0369          |
> | **Traffic Volume** | 1.9435   | 0.6632   | 0               |
>
> where we apply a causal threshold of 0.6 for visualization.
>
> [1] Glass, Thomas A., et al. "Causal inference in public health." Annual review of public health 34.1 (2013): 61-75.
> [2] Sharma, Kshitij, Yogesh K. Dwivedi, and Bhimaraya Metri. "Incorporating causality in energy consumption forecasting using deep neural networks." Annals of Operations Research 339.1 (2024): 537-572.
> [3] Zhang, Lei, et al. "Granger causal inference for interpretable traffic prediction." 2022 IEEE 25th International Conference on Intelligent Transportation Systems (ITSC). IEEE, 2022.
>
> > W2: Possible circular reasoning in DAG $\rightarrow$ $J_m$ invertible $\rightarrow$ latent recovery $\rightarrow$ DAG learning (Theorem 3); unclear what is assumed vs. learned.
>
> Thank you for the interesting point. In our humble opinion, it is not circular. Actually, we assume the unknown graph is a DAG, as done in most of the causal discovery work (extended work to the cyclic case is possible but requires other strong assumptions). Then, under the assumption that the graph is a DAG, we finally learn the graph. We would like to add that the DAG assumption is about **the class of the graph**, but which graph actually fits the data is learned just from the data.
>
> Thus, our logical chain is: **causal graph is a DAG (assume) $\rightarrow$ $J_m$ invertible $\rightarrow$ latent recovery $\rightarrow$identify the causal graph (learned)**

---

> ### Author Response · Authors · 2025-11-21
> **Author Response (2/2)**
>
> > W3: Dense notation; needs earlier consolidated notation table.
>
> We thank the reviewer for this great suggestion. The full notation table is extensive, so it remains in the appendix; in light of this suggestion, we have moved the key and most frequently used notations to the main paper to improve readability and flow, as shown in Table 1 in our revised manuscript and below:
>
> **Table 1: Key notations, explanations, and structures used in the CaDRe model**
>
> | **Symbol** | **Explanation** | **Symbol** | **Explanation** | **Symbol** | **Explanation** |
> |:--:|:--|:--:|:--|:--:|:--|
> | $\mathbf{x}_t$ | observed variables at time $t$ | $\mathbf{z}_t$ | latent variables at time $t$ | $\mathbf{s}_t$ | dependent observation noise |
> | $\boldsymbol{\epsilon}_{\mathbf{x}_t}$ | independent noise of observations | $\boldsymbol{\epsilon}_{\mathbf{z}_t}$ | independent latent noise | $\mathcal{X}_t$ | support of observed variables |
> | $\mathcal{Z}_t$ | support of latent variables | $g(\cdot)$ | generating function from $(\mathbf{z}_t, \mathbf{s}_t)$ to $\mathbf{x}_t$ | $r(\cdot)$ | latent transition function from $\mathbf{z}_{t-1}$ to $\mathbf{z}_t$ |
> | $m(\cdot)$ | mixing function from $(\mathbf{z}_t, \mathbf{s}_t)$ to $\mathbf{x}_t$ | $h_z(\cdot)$ | invertible transformation from $\mathbf{z}_t$ to $\hat{\mathbf{z}}_t$ | $\mathbf{J}_g(\mathbf{x}_t)$ | Jacobian for observed causal DAG |
> | $\mathbf{J}_r(\mathbf{z}_t)$ | Jacobian for latent causal DAG | $\mathbf{J}_m(\mathbf{s}_t)$ | Jacobian for mixing structure | $\text{supp}(\cdot)$ | support matrix of Jacobian |
>
> > W4: Assumptions mentioned are difficult to verify in real world, for example latent drift assumtion (A3) in Theorem 1.
>
> Yes, we agree that assumptions A2 and A3 are not immediately verfied, since latent variables are **inherently unobserved**. But these assumption can be tested indirectly: Under these assumptions, we can first estimate the model, and then we can test whether these assumptions hold true in real-world climate data. We conducted checks in Appendix E.2, Lines 1942–1976 and Figure A11 + A12 of the revised manuscript, demonstrate that both A2 (contextual variability) and A3 (latent drift) **hold approximately** in real-world climate data.
>
> Notably, these assumptions typically holds in real-world case. (1) A2 / A3 is a type of "variability" assumption on time-series data. As demonstrated in our experiments above, they generally hold for climate data, which exhibit rich variability. (2) Moreover, they are much weaker than the strict value-level invertibility often imposed in prior CRL studies (please see Fig. A6 for a intuition), and can satisfy most of function classes, as examples shown in our Appendix A.2, Line 1065-1079.

---

> > ### Comment · Reviewer_tKCX · 2025-11-27
> > **Rebuttal Response**
> >
> > I have read the rebuttal and carefully reviewed all points. Based on the author's rebuttal and the reviewer comments, I am confident in my original assessment and the assigned score.

---

### Author Response · Authors · 2025-12-04
**Summary of Revisions and Responses (1/2)**

Dear AC and Reviewers,

We express our sincere gratitude to all the reviewers for their valuable insights and to the AC for the time and effort dedicated to handling our paper.

We propose **CaDRe**, a model that jointly learns (1) the causal graph between observed climate variables and (2) the hidden dynamic processes (latent drivers) behind them, from time-series data. The authors prove that this joint recovery is theoretically identifiable (even in a nonparametric setting), then implement it with a VAE-style architecture plus causal constraints, and show on synthetic and real climate datasets that CaDRe both forecasts well and recovers physically meaningful causal structures.

Our submission initially received four reviews with scores of **8, 6, 6, and 4**. Under the updated ICLR rebuttal policy, reviewers had a limited time to raise additional questions. Only Reviewer tKCX (rating: **6**) provided a follow-up comment after reading the rebuttal and explicitly stated that they remain confident in their original assessment and score.

Below, we summarize how our rebuttal and manuscript revisions addressed the main concerns, now explicitly pointing to where the changes appear in the revised main paper and appendix.

---

**• Interpretability of latent variables and causal relations (tKCX W1; 5w99 W3; 3HPk Q2)**
We clarified what the latent variables and discovered causal edges represent for each domain (public health, electricity, traffic, climate). In particular, we added a dataset-specific interpretation table and concrete traffic examples:
- Dataset-wise interpretations of latent variables and edges are now summarized in `Table A19` (`Lines 1870–1886`), covering ILI, ECL, Traffic, and climate datasets.
- For Traffic, we included an illustrative Jacobian and causal-edge example(`Figure A7` and discussion near it).
- For climate data, we added qualitative visualizations—such as ENSO-related teleconnections and wind-field structures—to link learned graphs with established climatological knowledge (`Figure 5` and appendix `Figure A6`, plus ENSO teleconnection results with Niño indices and SOI).

---

**• Clarification of theoretical illustration (tKCX W2–W4; bJxc Q1–Q3; 3HPk W1/Q1)**
We refined the theory section and made the logical chain assumptions → identifiability → learning modules much more explicit:

- At the beginning of `Section 3` in the main paper, we now provide a **high-level overview** of Theorems 1–3 and add **intuitive proof sketches** and two proof “pipelines” that connect the main mathematical steps.
- We explicitly clarify that we assume *the true graph lies in the class of DAGs*, but do not assume the specific DAG is known.
- We clarified the distinction between **distribution-level** and **value-level** identifiability, emphasizing that distribution-level identifiability (trivial) is an intermediate step and value-level identifiability (non-trivial) is the final result; the main text now points directly to the proof sketch (`Section 3`, around `Lines 159–161`) and the detailed derivations (`Appendix A.1–A.3`).
- We expanded the discussion of Assumption A2 (contextual/distributional variability) with several concrete functional examples in `Appendix A.2` (`Lines 1065–1079`).
- We further elaborated Assumption A3 (latent drift) and provided empirical diagnostics in `Appendix E.2` (`Lines 1942–1976`, `Figures A11` and `A12`), demonstrating that A2 and A3 are approximately **satisfied in the climate datasets** used.

---

> ### Author Response · Authors · 2025-12-04
> **Summary of Revisions and Responses (2/2)**
>
> ---
>
> **• Modeling and architectural clarity (bJxc W1–W5, Q2–Q4; 5w99 W3)**
> We substantially clarified the architecture and structure-learning mechanism:
>
> - **Structure-learning block / decoder Jacobian → causal graph:**
>   In `Section 4` (revised `Lines 309–370`) and the corresponding appendix sections (`Appendix C`), we now clearly explain that the causal graph is **not** directly read off from the decoder weights. Instead, we:
>   1. Compute the decoder Jacobian $J_g = \partial x /  \partial s$.
>   2. Use the functional-equivalence result (Theorem 2) to transform $J_g$ into a Jacobian $J_m$ w.r.t. parents in the SEM, which yields a causal adjacency matrix.
> - **How sparsity and DAG constraints are imposed:**
>   We added that the causal relationships are **not manually “hard-coded”** in a dense layer. Instead, we:
>   - Impose a sparsity penalty on the adjacency matrix inferred from the Jacobian.
>   - Impose a DAG constraint using a smooth acyclicity function `h(A)` to penalize cycles.
>   These are now clearly integrated into `Section 4` (`Lines ~354–370`) and discussed in more detail in `Appendix C.1–C.3`.
> - **Normalizing flow, KL divergence, and causality:**
>   We clarified that the normalizing flow is used to model the **noise variables** (e.g., `ε_x`, `ε_z`), and the KL divergence term enforces that these noise components match an *independent* standard normal prior, reversely inferring the causal graph. The main text now emphasizes this point in `Section 4.1` (`Lines 309–352`) and the implementation details in `Appendix C.2`.
> - **Why the z-encoder and s-encoder learn different features:**
>   We clarified that, although both encoders are MLPs, they **do not share parameters** and are trained with different posteriors/prior structures and objectives.
>
> ---
>
> **• Scalability, dimensionality, and additional experiments (5w99 W4/Q2–Q3; 3HPk W1/Q3/Q4; bJxc W2)**
> We expanded the experiments to better characterize performance at larger scales and higher latent dimensions:
>
> - **Higher latent dimensionality (`k > 4`):**
>   We added experiments with larger latent dimension (`k` up to at least 12) on synthetic data, reported in the new `Table A10` (appendix).
> - **Larger observed graphs / dimensional effects:**
>   We extended synthetic-graph experiments to higher observed dimensions and summarized the effect on SHD, precision/recall, and MCC in `Table A10` and related tables.
> - **High-dimensional CMIP-style climate data:**
>   We added an experiment on **CMIP-style high-dimensional data** (time × 30 latitude × 60 longitude → thousands of spatial grid points), comparing CaDRe against strong time-series baselines (CARD, MICN, iTransformer, TimesNet, Autoformer) in `Table A17`.
> - **Spatial masking strategy / physical distance prior:**
>   To help with scalability, we highlight a spatial masking strategy based on physical distance priors that restricts candidate causal edges to geographically plausible neighbors. This strategy is now explicitly emphasized in the main paper and detailed in `Appendix` (revised `Lines 1589–1594`, `Table A9`).
>
> ---
>
> **• Robustness to assumption violations and additional baselines (5w99 W1/Q1; 3HPk W1/Q4; bJxc W3/Q3/Q4)**
> We performed several new analyses to test robustness beyond the idealized identifiability conditions and to broaden baseline coverage:
>
> - **Ablations on identifiability-assumption violations:**
>   We systematically violated assumptions `A2` (contextual variability), `A3` (latent drift), and `A5` (generation variability) and reported the resulting performance in `Appendix E.2` (`Lines 1942–1976`, plus `Lines 1737–1746`). CaDRe shows graceful degradation rather than catastrophic failure across both CRL and causal discovery metrics (SHD, TPR, precision, MCC).
> - **Empirical diagnostics for A2 and A3 on real climate data:**
>   As requested by Reviewer 5w99 and 3HPk, we added heuristic diagnostics in `Appendix E.2` (`Figures A11` and `A12`), verifying that different contexts `c` correspond to different distributions of `x` and that different `z` correspond to different `x`, showing that A2 (distributional variability) and A3 (latent drift) hold approximately in our climate datasets.
> - **Score-based causal discovery baselines:**
>   Responding to Reviewer 3HPk’s question, we added **score-based** baselines: `BIC` and the `Generalized Score Function`, in addition to the original constraint-based baselines (FCI, CDNOD, PCMCI, LPCMCI). CaDRe consistently outperforms all baselines on structure-recovery metrics.
>
> ---
>
> Therefore, we believe that the reviewers’ concerns have been fully addressed. We again sincerely thank the AC and all reviewers for their engagement, constructive suggestions, and support.
>
> **Best regards,**
> *The Authors of Submission 9926*

---

### Note · Program_Chairs · 2026-01-17
**Submission Desk Rejected by Program Chairs**

The following references in this submission do not refer to real documents and/or have major errors in bibliographic information:

 Shiyu Gu, David Salinas, Valentin Flunkert, Jan Gasthaus, and Tim Januschowski. Parameterization of state space models for forecasting with structured latent dynamics. arXiv preprint arXiv:2202.09384, 2022.
Shiyu Gu, Tim Januschowski, and Jan Gasthaus. Efficiently modeling time series with missing data using a state space approach. In NeurIPS Time Series Workshop, 2021a.
Shiyu Gu, David Salinas, Valentin Flunkert, and Jan Gasthaus. Combining latent state-space models and structural time series models for probabilistic forecasting. International Journal of Forecasting, 37(3):1182-1199, 2021b.